# Constraining the trend in the ocean $CO_2$ sink during 2000–2022

Nicolas Mayot [1] ✉, Erik T. Buitenhuis[1], Rebecca M. Wright [1], Judith Hauck [2,3], Dorothee C. E. Bakker [1] & Corinne Le Quéré [1]

The ocean will ultimately store most of the $CO_2$ emitted to the atmosphere by human activities. Despite its importance, estimates of the 2000–2022 trend in the ocean $CO_2$ sink differ by a factor of two between observation-based products and process-based models. Here we address this discrepancy using a hybrid approach that preserves the consistency of known processes but constrains the outcome using observations. We show that the hybrid approach reproduces the stagnation of the ocean $CO_2$ sink in the 1990s and its reinvigoration in the 2000s suggested by observation-based products and matches their amplitude. It suggests that process-based models underestimate the amplitude of the decadal variability in the ocean $CO_2$ sink, but that observation-based products on average overestimate the decadal trend in the 2010s. The hybrid approach constrains the 2000–2022 trend in the ocean $CO_2$ sink to $0.42 \pm 0.06$ Pg C yr$^{-1}$ decade$^{-1}$, and by inference the total land $CO_2$ sink to $0.28 \pm 0.13$ Pg C yr$^{-1}$ decade$^{-1}$.

The ocean plays a crucial role in the global carbon cycle and climate. Every year, it absorbs, on average, 25% of the anthropogenic $CO_2$ emitted to the atmosphere from burning fossil fuels and land-use change[1,2]. The ocean will be the main storage for most of the $CO_2$ emitted to the atmosphere from human activities, absorbing more than half of the cumulative emissions on a timescale of 1000 years, and between 60% and 85% on timescales of 10,000 years or longer[3]. The ocean $CO_2$ sink responds mainly to the increase in atmospheric $CO_2$ and is the result of processes taking place on different timescales: the dissolution of anthropogenic $CO_2$ at the surface, its buffering by carbonate chemistry, and its transport to the intermediate and deep ocean by ocean circulation[4]. This uptake is modulated by the variable growth rate in atmospheric $CO_2$[5]. Furthermore, the ocean carbon reservoir is also sensitive to climate variability and climate change[6,7], which modulates the response of the ocean $CO_2$ sink to anthropogenic $CO_2$ emissions, generates variability, and can alter both the decadal rate of change and long-term storage of carbon in the ocean. An accurate estimate of the trends and variability in the ocean $CO_2$ sink is needed to better understand how the ocean carbon cycle responds to the various drivers of change and to predict its evolution and its long-term fate. An accurate assessment of the ocean $CO_2$ sink and its

variability also provides an independent constraint on the other terms of the global carbon budget, in particular the trend and variability of the $CO_2$ sink by the terrestrial biosphere[8].

Over the last two decades, considerable progress has been made in improving our quantitative assessment of the ocean $CO_2$ sink and its trend and variability. The IPCC Fourth Assessment Report[9] assessed a mean ocean $CO_2$ sink for the 1990s based on observations of $2.2 \pm 0.4$ Pg C yr$^{-1}$, which has stood the test of time[1,3]. The first estimates of the ocean $CO_2$ sink were based on ocean general circulation models[10], followed by estimates inferred from $O_2/N_2$ observations[11,12] and broader geochemical data[10,13]. Later, the development of global ocean biogeochemical models (GOBMs) provided the first estimates of the ocean $CO_2$ sink variability in response to climate variations and long-term increases in atmospheric $CO_2$ concentration[14,15]. These models simulate the processes that regulate the full carbon cycle in the ocean (both natural and anthropogenic) and respond to changes in climate, weather, and variations in $CO_2$ levels in the atmosphere. Results from such process-based ocean models have suggested that the ocean $CO_2$ sink is sensitive to climate change and variability but that this variability is much smaller than the trend induced by the rising atmospheric $CO_2$[16].

[1]School of Environmental Sciences, University of East Anglia, Norwich, UK. [2]Alfred-Wegener-Institut, Helmholtz-Zentrum für Polar- und Meeresforschung, Bremerhaven, Germany. [3]Universität Bremen, Bremen, Germany. ✉e-mail: n.mayot@uea.ac.uk

More recently, observation-based estimates of the ocean $CO_2$ sink have become available from multiple data products. This has been made possible by the annual release of quality-controlled observations of $CO_2$ fugacity ($fCO_2$) at the sea surface—analogous to the partial pressure of $CO_2$—compiled within the Surface Ocean $CO_2$ Atlas (SOCAT) since 2015[17,18], and the emergence of machine learning and other advanced numerical approaches to extrapolate these relatively sparse space-time observations of $fCO_2$. Estimates from these observation-based products ($fCO_2$-products), which all used the SOCAT database as a starting point, confirmed some aspects of the ocean $CO_2$ sink simulated by the GOBMs, including the mean ocean $CO_2$ sink within observational uncertainties, the presence of a hiatus in its growth rate in the 1990s[19,20], and of variability much smaller than its long-term trend[2]. It should be noted that these $fCO_2$-products only assess the air-sea $CO_2$ flux, and not where anthropogenic $CO_2$ is ultimately stored in the ocean, which would require additional measurements of carbon in the water column, as well as more assumptions[21–24].

Despite recent progress, the two types of estimates used within the global carbon budget annual update by the carbon research community diverge over the last two decades (2000–2022)[22,23,25], with the $fCO_2$-product ensemble suggesting a decadal rate of growth of the ocean $CO_2$ sink almost twice as high as that simulated by the GOBMs ensemble (Table 1)[2,25]. The systematic nature of this discrepancy suggests a structural error in one or both methodologies used. In addition, the $fCO_2$-products suggest a higher amplitude of decadal variability on average than that simulated by the GOBMs over the period 1990–2022 (Table 1, Supplementary Fig. S1)[2]. Results from models using data assimilation[26,27] also suggest an underestimated decadal variability in GOBMs air-sea $CO_2$ flux[28,29]. This lack of consistency between the $fCO_2$-products and the GOBMs ensemble originates in the mid- and high-latitude regions of both hemispheres (poleward of 30°N and 30°S)[2].

We focus here on understanding the factor-of-two inconsistency in estimates of the 2000–2022 trend in the ocean $CO_2$ sink between $fCO_2$-products and GOBMs, which occurs despite the increasing number of $fCO_2$ observations (i.e., from ~4500 gridded observational data points a year in the 1990s, to 10,000 in the 2000s, and 15,000 in the 2010s). We introduce and use a hybrid approach that uses, as a starting point, the NEMO-PlankTOM12.1 GOBM forced with meteorological and climate reanalysis data from the National Centres for Environmental Prediction (NCEP[30]) and observed atmospheric $CO_2$ concentration. The hybrid approach then constrains the model outcome on a yearly basis using SOCAT observations to provide an annual estimate of the ocean $CO_2$ sink (see methods). NEMO-PlankTOM12.1 is the latest update of an established GOBM that was used from the onset in the annual updates of the global carbon budget analysis and which was designed for the study of the ocean $CO_2$ sink variability[31]. Its overall performance in simulating ocean physics and biogeochemistry is comparable to that of other GOBMs in the global carbon budget analysis (see methods, Table 1, Supplementary Figs. S2 and S3)[2,19].

The hybrid approach preserves the coherence of the physical and biogeochemical processes represented in NEMO-PlankTOM12.1 and goes beyond the traditional model evaluation by constraining the model output fields of $fCO_2$ against the observed $fCO_2$ data provided by SOCAT. A similar hybrid approach was recently published[32], but with a machine learning algorithm used to derive the factors influencing the $fCO_2$ variability. Here, the mechanism as represented in the NEMO-PlankTOM12.1 model, including the mixed-layer dynamics and the large-scale circulation, the carbonate chemistry, and the organic carbon transfer to depth resulting from biological processes (see methods for a description of the model) remained unchanged and thus also constrained the results. The hybrid approach used here thus provides an estimate of the ocean $CO_2$ sink that is based on both observations and on current theoretical understanding of the response of the ocean $CO_2$ sink to climate change and variability. Note that the hybrid approach used here, which optimises a target variable using multiple model simulations and a cost function, has been used in previous studies to constrain global ocean primary production[33] and air-sea fluxes of $N_2O$[34] and $CCl_4$[35].

## Results

### Constraints on the decadal variability of the global ocean $CO_2$ sink

In order to use observations to constrain the annual global ocean $CO_2$ sink simulated by NEMO-PlankTOM12.1 standard model simulation, perturbed simulations were deliberately produced. This was obtained by perturbing model parameters. Perturbed simulations provided a range of possible values for the ocean $CO_2$ sink around the estimate from the standard model simulation, and covered the expected range suggested by the global carbon budget analysis. We show here results obtained with the perturbation of the half-saturation constant of bacterial remineralization, which is more homogeneous and, therefore, more robust (see methods and the Supplementary information for details of the sensitivity analyses). Then, for each year, the optimal $CO_2$ sink was found within this range of possibilities by optimising the calculated mean square error (MSE) between the simulated $fCO_2$ and the SOCAT observations. The hybrid approach also provides a quantitative estimate of uncertainty (see methods).

The hybrid approach increases the decadal variability of the ocean $CO_2$ sink simulated by the NEMO-PlankTOM12.1 process-based ocean model (see methods for the definition of decadal variability). Originally, over the period 1990–2022, NEMO-PlankTOM12.1 simulated amplitudes of decadal variability for the ocean $CO_2$ sink of 0.11 Pg C $yr^{-1}$. This value is at the high end of the decadal variability simulated by the other GOBMs used in the global carbon budget analysis (Table 1). The hybrid approach further increases this simulated decadal variability by 18% to $0.13 \pm 0.02$ Pg C $yr^{-1}$, to a value close to the decadal variability estimated by the $fCO_2$-products ($0.14 \pm 0.06$ Pg C $yr^{-1}$).

We tested the robustness of the decadal variability produced by the hybrid approach with respect to (i) the choice in the selected model's configuration and parameter perturbed, and (ii) the annual availability and distribution of the SOCAT data. To do this, we first

**Table 1 | Temporal variability of the ocean $CO_2$ sink estimated using different methods**

| Methods | Amplitude of decadal variability (Pg C $yr^{-1}$) | Decadal trends (Pg C $yr^{-1}$ $decade^{-1}$) | | | |
|---|---|---|---|---|---|
| | 1990–2022 | 1990s | 2000s | 2010s | 2000–2022 |
| **Process-based models** | | | | | |
| GCB's GOBMs | $0.08 \pm 0.02$ | $0.11 \pm 0.16$ | $0.40 \pm 0.10$ | $0.34 \pm 0.08$ | $0.28 \pm 0.05$ |
| NEMO-PlankTOM12.1 | 0.11 | 0.02 | 0.27 | 0.53 | 0.33 |
| **Observation-based products** | | | | | |
| GCB's $fCO_2$-products | $0.14 \pm 0.06$ | $-0.12 \pm 0.35$ | $0.71 \pm 0.38$ | $0.68 \pm 0.19$ | $0.54 \pm 0.13$ |
| **Hybrid approach** | | | | | |
| This study | $0.13 \pm 0.02$ | $-0.19 \pm 0.17$ | $0.80 \pm 0.21$ | $0.44 \pm 0.15$ | $0.42 \pm 0.06$ |

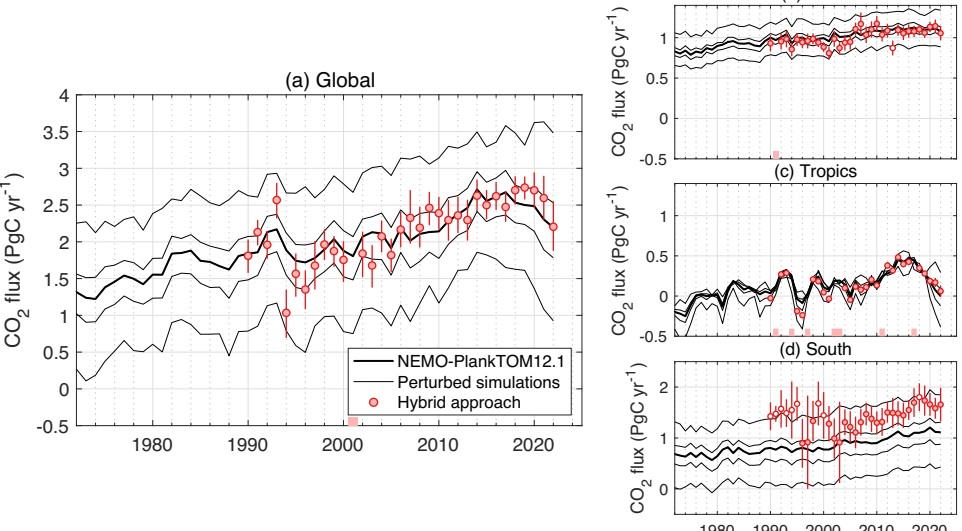

**Fig. 1 | Ocean CO$_2$ sink constrained at global scale and by latitude bands between 1990 and 2022.** Positive values denote a sink for CO$_2$. **a** The global ocean is divided into three latitudinal bands: **b** North (>30°N), **c** Tropics (30°N–30°S), and **d** South (<30°S). The thick black line represents NEMO-PlankTOM12.1 standard simulation, the thin black lines represent the perturbed simulations, and the red dots represent the estimate using the hybrid approach with ±1σ (68%) confidence interval. On the *x* axis, the years highlighted in red have an unconstrained ocean CO$_2$ sink. Empty red dots are years with an ocean CO$_2$ sink value constrained but outside the bounds of the perturbation experiments (i.e., uncertain values, see methods). The perturbed simulations are produced by varying the half-saturation constant of bacterial remineralisation (from $5 \times 10^{-6}$ mol L$^{-1}$ to $18 \times 10^{-6}$ mol L$^{-1}$).

applied the hybrid approach to a total of six different model setups (see methods). This first sensitivity analysis suggested a comparable increase in decadal variability ($0.16 \pm 0.05$ Pg C yr$^{-1}$, Supplementary Table 3). Secondly, the hybrid approach was applied by considering observations from three consecutive years (see methods). This second sensitivity analysis also suggested a comparable increase in the decadal variability (to $0.14 \pm 0.02$ Pg C yr$^{-1}$, Supplementary Fig. S1). Overall, these two sensitivity analyses confirmed the robustness of the amplitude of the decadal variability suggested by the hybrid approach (Table 1). In contrast, the amplitude of the year-to-year variability was less robust because of insufficient data to constrain the hybrid approach on a yearly basis, especially in the 1990s (see Supplementary information).

## Constraints on the regional ocean CO$_2$ sink

The regional ocean CO$_2$ sinks were constrained by applying the same hybrid approach but separately for three latitude bands, using only the observations in the North (>30°N), Tropics (30°S to 30°N), or South (<30°S) to constrain the regional ocean CO$_2$ sink simulated by NEMO-PlankTOM12.1 (Fig. 1b–d). The hybrid approach substantially modified the simulated ocean CO$_2$ sink in the mid- and high-latitude regions, particularly in the South, but with the Tropics remaining similar to the original NEMO-PlankTOM12.1 results. The hybrid approach increased the decadal variability simulated by NEMO-PlankTOM12.1 in the North and South regions (Supplementary Table 2).

In the North, where there are more SOCAT observations, the hybrid approach barely modified the mean ocean CO$_2$ sink simulated by NEMO-PlankTOM12.1. However, in the South, where there are fewer observations, the hybrid approach increased the mean ocean CO$_2$ sink simulated by NEMO-PlankTOM12.1 by 44% in the period 2000–2022. The hybrid approach applied at every 5° of latitude to constrain the mean climatological ocean CO$_2$ sink during 2000–2022 (rather than its annual values) shows that the mismatch between the model and the fCO$_2$ observations in the 40–60°S band could be the main cause of the underestimation of the ocean CO$_2$ sink in the South (Fig. 2). Note that it is the Southern region that has the most influence on the global ocean CO$_2$ sink.

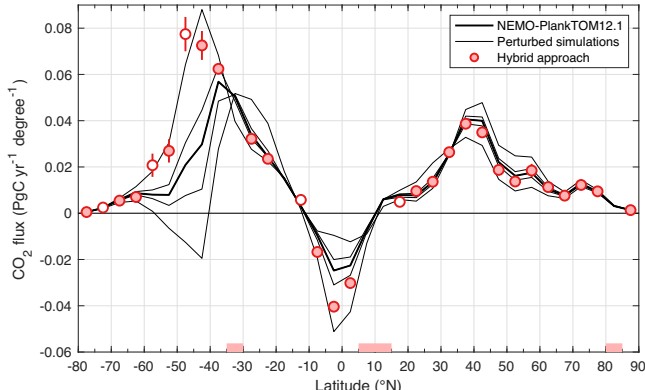

**Fig. 2 | Latitudinal mean ocean CO$_2$ sink averaged between 2000 and 2022.** Positive values denote a sink for CO$_2$. The thick black line represents NEMO-PlankTOM12.1 standard simulation, the thin black lines represent the perturbed simulations, and the red dots represent the hybrid approach with ±1σ (68%) confidence interval. On the *x* axis, latitudes highlighted in red have an unconstrained ocean CO$_2$ sink. Empty red dots are latitudes with an uncertain constrained ocean CO$_2$ sink (see methods).

## Decadal trends of the global ocean CO$_2$ sink

In the 1990s and 2000s, the hybrid approach enhanced the decadal trends simulated by NEMO-PlankTOM12.1, bringing the trends closer to those suggested by the fCO$_2$-products. In the 1990s, the hybrid approach decreased the NEMO-PlankTOM12.1 simulated trend from 0.02 to $-0.19 \pm 0.17$ Pg C yr$^{-1}$ decade$^{-1}$, to a value with the same sign and within the range of the fCO$_2$-product ensemble trend of $-0.12 \pm 0.35$ Pg C yr$^{-1}$ decade$^{-1}$. In the 2000s, the hybrid approach increased the simulated trend from 0.27 to $0.80 \pm 0.21$ Pg C yr$^{-1}$ decade$^{-1}$, to a value also closer to and within the range of the fCO$_2$-product ensemble of $0.71 \pm 0.38$ Pg C yr$^{-1}$ decade$^{-1}$ (Fig. 3, Table 1).

In the 2010s, the hybrid approach decreased the simulated trend by NEMO-PlankTOM12.1 from 0.53 to $0.44 \pm 0.15$ Pg C yr$^{-1}$ decade$^{-1}$, which is below the strong decadal trend of $0.68 \pm 0.19$ Pg C yr$^{-1}$

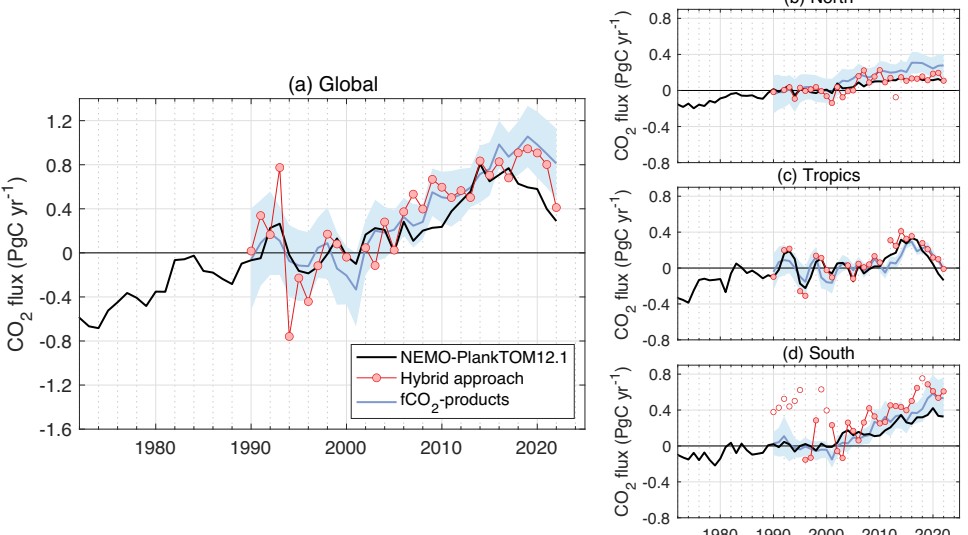

**Fig. 3 | Anomalies of the ocean $CO_2$ sink constrained at global scale and by latitude bands between 1990 and 2022. a** The global ocean is divided into three latitudinal bands: **b** North (>30°N), **c** Tropics (30°N–30°S), and **d** South (<30°S). The black line represents NEMO-PlankTOM12.1 standard simulation, the red line with the dots represents the hybrid approach, and the blue line represents the fCO₂-products. The fCO₂-product estimate is shown with ± 1σ. For each time series, the long-term mean between 1990 and 1999 was removed to focus on the variability of the ocean $CO_2$ sink. fCO₂-product data credit: Global Carbon Budget 2023, licensed under Creative Commons Attribution 4.0 International (https://creativecommons.org/licenses/by/4.0/deed.en), no changes were made.

decade⁻¹ suggested by the fCO₂-product ensemble, but above the decadal trend of $0.34 \pm 0.08$ Pg C yr⁻¹ decade⁻¹ suggested by the GOBMs ensemble. As a consequence, the trend between 2000 and 2022 from the hybrid approach ($0.42 \pm 0.06$ Pg C yr⁻¹ decade⁻¹) lies between the trend simulated by NEMO-PlankTOM12.1 and the GOBM ensemble ($0.28 \pm 0.05$ Pg C yr⁻¹ decade⁻¹), and that suggested by the fCO₂-product ensemble ($0.54 \pm 0.13$ Pg C yr⁻¹ decade⁻¹). However, unlike the GOBM and fCO₂-product ensembles where the growth is similar between the two decades, the trend in the hybrid approach is made of a decade of strong growth (in the 2000s) followed by a decade of weak growth (the 2010s). The distinct decadal trend variations between the 2000s and the 2010s suggested by the hybrid approach are robust to different configurations of the original model and to the choice of perturbed parameters (see methods). In comparison, only two out of seven fCO₂-products suggest that the trend in the 2000s should be higher than the trend in the 2010s over the global ocean.

### Origin of the discrepancy among estimates

The differences in the decadal trends for the 2010s among the hybrid approach, the NEMO-PlankTOM12.1 and the fCO₂-products were mostly associated with the mid- and high-latitude regions (Fig. 3). Further investigation to determine the origin of the discrepancy was conducted, first by scrutinising estimates in the North where the density of SOCAT observations was the highest. This region encompasses 17% of the global ocean area. Because of the higher density of observations in the North compared with other latitudes, it is possible in this region to compare the trends in ocean $CO_2$ flux in areas that are generally well-sampled to those in areas that are poorly sampled. A similar analysis cannot be done for the South because of the insufficient data coverage. In the North, the hybrid approach and average GOBMs produce lower trends compared to the average of the fCO₂-products (Fig. 3b).

The SOCAT observations, by themselves, do not confirm the existence of a strong decadal trend in the 2010s in the North. The strong decadal trend in the ocean $CO_2$ flux estimated by the fCO₂-products is primarily driven by diverging trends between the $CO_2$ fugacity at the surface of the ocean compared to that in the atmosphere ($\Delta fCO_2$)[36]. The SOCAT observations converted into $\Delta fCO_2$ in

the North region between 2000 and 2019 show a positive and higher trend in the 2000s compared to the 2010s (Fig. 4). Similar temporal patterns were visible in the $\Delta fCO_2$ data from the fCO₂-products subsampled to SOCAT sampling points, with a decadal trend in $\Delta fCO_2$ in the 2000s significantly higher than in the 2010s (Kruskal–Wallis test, $p$ value < 0.01), as expected. However, when not subsampled, the fCO₂-products suggested a decadal trend in $\Delta fCO_2$ in the 2000s that is not significantly higher than in the 2010s (Kruskal–Wallis test, $p$ value = 0.14), with an overlap in the estimated uncertainties in the two decades, explaining the small differences in the $CO_2$ sink trend between the 2000s and 2010s. This is induced by the fact that three of the seven fCO₂-products suggested a greater trend in the 2010s compared to the 2000s, and a fourth fCO₂-product suggested a strong trend in both decades. In comparison, when subsampled or not subsampled, the GOBMs suggested a decadal trend in $\Delta fCO_2$ in the 2000s significantly higher than in the 2010s (Kruskal–Wallis tests, $p$ value < 0.001). In addition, the decadal trend in $\Delta fCO_2$ in the North during the 2010s is significantly higher in the fCO₂-product ensemble than in the GOBMs ensemble (Kruskal–Wallis test, $p$ value < 0.01). The differences between the subsampled and not subsampled results suggest that different extrapolation methods outside of data-rich regions could account for the higher decadal trend in ocean $CO_2$ sink in the North over the 2010s in the fCO₂-products ensemble compared to the GOBMs ensemble and the hybrid approach (Fig. 3b, Supplementary Table 2).

Within the four fCO₂-products that suggested a strong positive decadal trend in $\Delta fCO_2$ in the 2010s in the North, the ocean area associated with the highest trend values overlaps the northern Pacific Ocean (Fig. 5a) which was undersampled in the 2010s (white areas in Fig. 5b). In quantitative terms, these four fCO₂-products suggested that the Pacific Ocean contributed 0.04 Pg C yr⁻¹ decade⁻¹ of the 2010s trend in the North, while the other three fCO₂-products (below the fCO₂-product average) suggested that the Pacific Ocean had a negative trend in the 2010s (−0.01 Pg C yr⁻¹ decade⁻¹), as simulated by the GOBM average.

In addition to the extrapolation problems in the undersampled northern regions mentioned above, on a global scale, the estimate of the positive trend in the 2010s from the fCO₂-product ensemble has

been revised downwards in successive publications of the global carbon budget analysis between 2021 and 2023, while their trends for the 1990s and 2000s have remained relatively similar (Fig. 6). For each of the global carbon budget analyses published between 2021 and 2023, the fCO$_2$-product ensemble average has always been produced from seven estimates. However, two of the seven fCO$_2$-products were introduced, replacing previously submitted products that were not updated, and five were slightly updated. Among these five fCO$_2$-products, on average, the 2010 trend between the 2021 and 2023 publications decreased by −0.05 Pg C yr$^{-1}$ decade$^{-1}$. Thus, it was mainly the turnover in the last two fCO$_2$ products between 2021 and 2023 that led to a visible decrease in the ensemble average of −0.25 Pg C yr$^{-1}$ decade$^{-1}$ for the 2010 trend. Consequently, the downward revision observed for the observation-based estimate was mainly due to a change in two fCO$_2$-product methodologies and, to a lesser extent, to the annual updates of the SOCAT database and fCO$_2$-product

methods, suggesting that the trend of the 2010s estimated with the fCO$_2$-products is not robust at this stage.

## Discussion

Over the 1990s and 2000s, the ocean CO$_2$ sink experienced a well-documented stagnation in the 1990s[14,19,20,31] and a reinvigoration trend in the 2000s[37], but the amplitude of these decadal trends is uncertain. Several studies have shown that the trends were present in many GOBMs but with amplitude much lower than in the fCO$_2$-products[19,20]. Results from the hybrid approach presented here confirm that the amplitude of decadal trends in the ocean CO$_2$ sink is underestimated by NEMO-PlankTOM12.1 and the other GOBMs and suggest decadal trend values close to those estimated using fCO$_2$-products for these two decades (Table 1).

For the 2010 decade, the hybrid approach suggests a lower decadal trend in the ocean CO$_2$ sink compared to the 2000s, with a value closer to that simulated by most GOBMs and lower than that suggested by the fCO$_2$-products ensemble. However, both GOBMs and fCO$_2$-products ensembles suggest similar trends between 2000s and 2010s, while the hybrid approach (including the sensitivity analyses, Supplementary Table 3) consistently produced a higher trend in the 2000s compared to the 2010s.

Our results suggest that the estimate of the ocean CO$_2$ sink trend in the 2010s by the fCO$_2$-product ensemble is overestimated and sensitive to the availability and distribution of fCO$_2$ observations. Over the last three annual updates of global carbon budgets[2,38,39], the 2010s trend estimated from the ensemble of available products has decreased by 14% each year (Fig. 6). This supports our finding of an overestimated trend in the 2010s ocean CO$_2$ sink from the fCO$_2$-products ensemble, which is adjusted downwards as new data become available. In addition, the replacement of two members of the fCO$_2$-product ensemble by a hybrid approach along the same lines as presented here[32,40] and by a revised fCO$_2$-product aimed at improving the retrieval of the ocean CO$_2$ sink trend[41] have led to this downward revision of the 2010s trend in the latest ensemble.

The ±1σ uncertainty provided for the hybrid results reflects the capacity of the hybrid approach to constrain the annual ocean CO$_2$ sink given the availability and distribution of the fCO$_2$ observations. The annual uncertainty is then propagated to the decadal trend. The trend for the period 2000–2022 is better constrained than the individual ten-year trends, since the longer period naturally filters out short-term variability. Nevertheless, sensitivity tests suggest that additional uncertainty to the model set up influences the exact value of the trends, but not the overall patterns, and in particular the trend in the 2010s which is systematically lower than the trend in the 2000s in all sensitivity tests performed, and also systematically lower than the

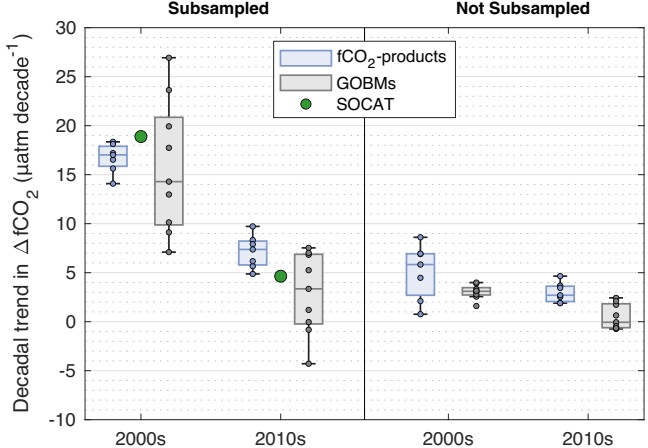

**Fig. 4 | Decadal trends in ΔfCO$_2$ in the North (>30°N, excluding the Arctic Ocean).** Comparison of the measured decadal trend in ΔfCO$_2$ from SOCAT (green dots) with that of the fCO$_2$-product ensemble (blue box-plots and dots) and of the GOBM ensemble (grey box-plots and dots). The box-plots display: the median, the lower and upper quartiles, and the minimum and maximum values that are not outliers. The trends are calculated from median annual values for the North region. On the left, fCO$_2$-products and GOBMs were subsampled at SOCAT locations, while on the right, they were not subsampled, and the value for the North region as a whole is shown. fCO$_2$-product and GOBM data credit: Global Carbon Budget 2023, licensed under Creative Commons Attribution 4.0 International (https://creativecommons.org/licenses/by/4.0/deed.en), no changes were made.

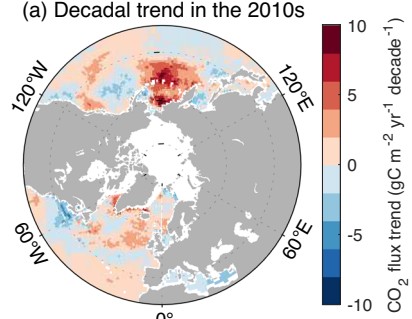

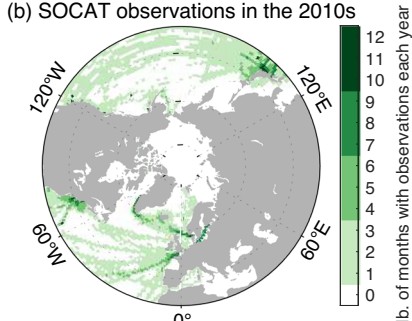

**Fig. 5 | Spatial distribution of anomalies in decadal trends of the ocean CO$_2$ sink estimated by four fCO$_2$-products in the North, and location of observations, for the period 2010–2019. a** The anomalies represent the difference between the average estimate from the four fCO$_2$-products with the highest trends during this decade, and the average of all seven fCO$_2$-products. **b** Median number of months with SOCAT observations available each year between 2010 and 2019. fCO$_2$-product data credit: Global Carbon Budget 2023, licensed under Creative Commons Attribution 4.0 International (https://creativecommons.org/licenses/by/4.0/deed.en), no changes were made.

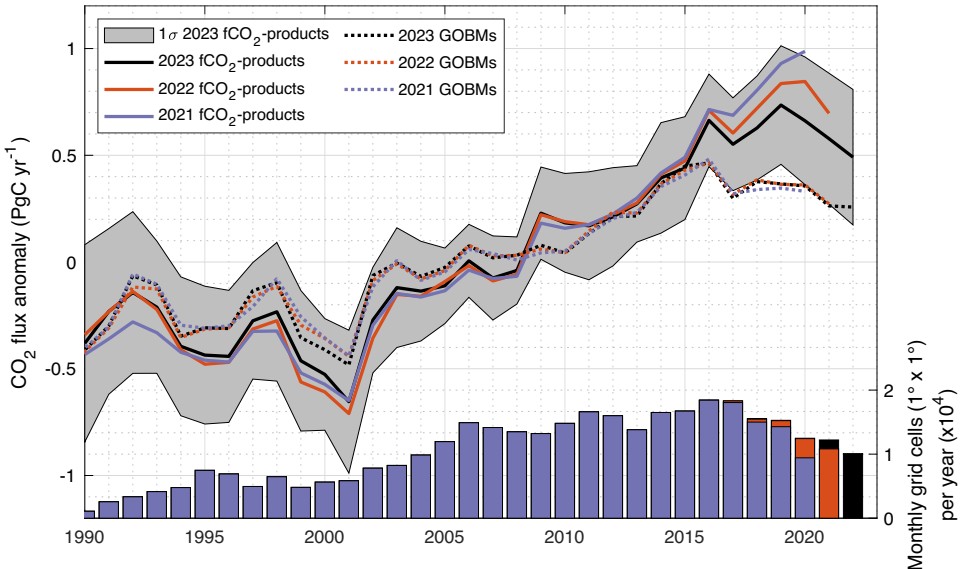

**Fig. 6 | Changes in fCO$_2$-product estimates over the last three updates of the global carbon budget analysis.** The last update of the fCO$_2$-product estimate was in 2023 (black line and grey shade). The previous two estimates were in 2022 (orange) and 2021 (blue). The left $y$ axis represents the global CO$_2$ flux anomaly (for each time series the long-term mean between 1990 and 2020 was removed). GOBM estimates for the same three global carbon budget updates are also shown in dotted lines. The bar chart at the bottom represents the number of annual observations in the SOCAT database, for each annual version of SOCAT (v2021, v2022, and v2023; using the same colour code as for the line). fCO$_2$-product and GOBM data credit: Global Carbon Budget 2023, licensed under Creative Commons Attribution 4.0 International (https://creativecommons.org/licenses/by/4.0/deed.en), no changes were made.

fCO$_2$-products ensemble for that decade. Our analysis demonstrates the importance of regular updates and efforts to collect fCO$_2$ observations as part of SOCAT[17], as well as regular evaluations of data products, including fCO$_2$-products and new hybrid methodologies[42,43].

Differences between NEMO-PlankTOM12.1, the hybrid approach, and the fCO$_2$-products ensemble for the 2010s decadal trend are mostly visible in the mid- and high-latitude regions of both hemispheres (Fig. 3). Our results suggest that some fCO$_2$-products overestimate the decadal trend of the 2010s in northern regions where there are few observations. In the northern latitudes, where the availability of measurements is highest, the fCO$_2$-product ensemble gives a decadal trend in the 2000s not significantly different from that of the 2010s (Fig. 4). Four fCO$_2$-products suggest a growing or strong trend during the 2010s, contrary to the fCO$_2$ observations alone, which is explained here by their strong trends in areas that were undersampled during the 2010s (Fig. 5). Hence, we hypothesise that methodological issues in some fCO$_2$-products could lead to an unrealistic amplification of the ocean CO$_2$ sink trend in the 2010s. In addition, the ocean CO$_2$ sink in the northern region is also more influenced by coastal processes than the southern region, which despite their importance remain uncertain[44–48]. Consequently, in the northern region, the way in which coastal fCO$_2$ observations are considered by the various fCO$_2$-products could induce some of the discrepancies among fCO$_2$-products. This would partly explain the lack of coherence between the GOBMs and the fCO$_2$-products over this recent decade[2,6,22,23].

In the Southern Ocean, our hybrid approach suggests that existing fCO$_2$ measurements could corroborate a strong and positive decadal trend in this region in the 2010s, and more generally between 2000–2022. However, the paucity of fCO$_2$ measurements in the Southern Ocean impedes our ability to evaluate the decadal trend in this region using observations only[42]. Nevertheless, our estimate of the decadal trend of the Southern Ocean CO$_2$ sink in the period 2000–2022 is within the range of the fCO$_2$-product ensemble (Supplementary Table 2). But the uncertainties associated with our hybrid approach are the largest in the Southern Ocean. Moreover, recent studies showed that undersampling could be responsible for strong biases in fCO$_2$ products in that region[43].

The hybrid approach presented here suggests that GOBMs underestimate the amplitude of the decadal variability of the ocean CO$_2$ sink over the past three decades by about $38 \pm 8\%$, and that the value should be within the range suggested by fCO$_2$-products (Table 1). Although our hybrid approach always suggests an underestimation of the decadal variability by GOBMs, the exact value is sensitive to the specific model configuration (Supplementary Table 3). The underestimation of the decadal variability by GOBMs was mostly reported in the North and South regions[2,20]. These deficiencies in the mid- and high-latitude regions have been related to the coarse resolution of the ocean circulation models[2], the generally poor representation of the seasonality of fCO$_2$ in these regions[49–51], and / or insufficient variability in simulated mode-water formation[52]. A possible overestimation of the decadal variability by fCO$_2$-products by 30% has been postulated before[42] because of undersampling of surface ocean fCO$_2$, mostly in the Southern Ocean[43,53,54]. Results from our hybrid approach also show there are remaining issues in CO$_2$ flux estimates of undersampled regions by fCO$_2$-products, but suggest that the amplitude of decadal variability is nevertheless greater than that estimated by GOBMs.

Finally, for the mean ocean CO$_2$ sink, the hybrid approach returns a higher mean CO$_2$ sink than NEMO-PlankTOM12.1 in the Southern Ocean because it corrects a consistent bias of overestimation of the surface ocean fCO$_2$ (Supplementary Fig. S2). Studies based on emergent constraint properties[55–57], and thorough assessments of the ability of GOBMs to simulate the Southern Ocean CO$_2$ sink[58], have also suggested that the current generation of models underestimates the global ocean CO$_2$ sink due to a deficient representation of ocean circulation in the Southern Ocean. The CO$_2$ outgassing from river fluxes, which is not included in NEMO-PlankTOM12.1 and the other GOBM simulations, could also explain the bias. River outgassing in the Southern Ocean has been estimated at 0.32 Pg C yr$^{-1}$ but with a high degree of uncertainty[2,59]. This bias would not affect the estimates of variability and trends, which are mainly driven by perturbation of surface ocean dynamics and atmospheric CO$_2$ concentration resulting from climate trends and variability (particularly winds) and by the evolution of atmospheric CO$_2$[6,16,60]. However, this means that our hybrid approach is less robust in estimating the mean ocean CO$_2$ sink

than the variability and trend of this sink, because the mean ocean $CO_2$ sink also depends on mixing between the surface ocean and the deep ocean[10], a process that is weakly constrained when only using surface $fCO_2$ observations, as is the case in the hybrid approach. Further work, in particular the use and/or assimilation of ocean interior carbon data[24,28,29], would be better suited to constrain the mean ocean $CO_2$ sink.

Our hybrid approach has shown that estimates of the temporal evolution of the ocean $CO_2$ sink can be reconciled, providing a well-constrained estimate of a significant growth in the ocean $CO_2$ sink between 2000 and 2022 of $0.42 \pm 0.06$ Pg C yr$^{-1}$ decade$^{-1}$, corresponding to a growth of 1.0 Pg C yr$^{-1}$ over those 23 years. Results from our hybrid approach show similarities and discrepancies with the $CO_2$ sink estimates from both GOBMs and $fCO_2$-products. Therefore, our analysis validates the global carbon budget approach to evaluate the ocean $CO_2$ sink by combining the data-based and process-model estimates[2]. Moreover, it confirms the importance of high-density $fCO_2$ observations, which are notably lacking in the Southern Ocean, for informing the $fCO_2$-products and our hybrid approach. It suggests that $fCO_2$-products could be further improved by scrutinizing the extrapolation of observations in the 2010s, which are evolving over the different versions released, in order to understand differences among $fCO_2$-products and then help improve the products.

Within the limits of the hybrid approach, a trend of $0.28 \pm 0.13$ Pg C yr$^{-1}$ decade$^{-1}$ in the total land $CO_2$ sink (including natural fluxes and emissions from land-use changes) can be inferred based on our estimate of the trend in the ocean $CO_2$ sink for 2000–2022, corresponding to a growth of 0.6 Pg C yr$^{-1}$ over those 23 years. This result was obtained by adding to and subtracting from our estimate of the ocean $CO_2$ sink, global carbon budget estimates for the growth rate of atmospheric $CO_2$ and $CO_2$ emissions from fossil fuels (taking into account cement carbonation, detailed in the Supplementary information)[2]. Our estimated trend in the total land $CO_2$ sink lies between the $0.43 \pm 0.20$ Pg C yr$^{-1}$ decade$^{-1}$ trend estimated by the global carbon budget analysis[2] and the trend of $0.07 \pm 0.14$ Pg C yr$^{-1}$ decade$^{-1}$ that would be obtained with the ocean $CO_2$ sink estimate from the $fCO_2$-products alone. Therefore, the land trend inferred from $fCO_2$ observations suggests either an overestimation of the increasing trend in the simulated land $CO_2$ sink by Dynamic Global Vegetation Models, an overestimation of the decreasing trend in $CO_2$ emissions from land-use changes by bookkeeping approaches used in the global carbon budget analysis (which might be increasing), or both[2]. The latter includes the possibility that emissions from land-use changes were stable or even increased during 2000–2022, which is plausible given the large uncertainty in land-cover change data and in management processes[2]. Our results demonstrate that ocean observations can constrain trends in land $CO_2$ fluxes, and that results are limited by the availability of $fCO_2$ observations.

## Methods
### Amplitude of the decadal variability and decadal trend estimates
The amplitude of the decadal variability was estimated as the standard deviation of the decadal component of the annual ocean $CO_2$ sink time series. The decadal components were extracted from the time series using the following signal decomposition methodology[52]: (i) the linear trend and long-term mean were removed to isolate the temporal variability, (ii) the decadal component of this detrended time series was obtained by filtering this time series with a 10-year Hanning window. The Hanning window is a filtering function with a 'bell-shaped' curve used to smooth the signal by emphasizing the feature near the center of the window. All uncertainty ranges are reported here with a $\pm 1\sigma$ (68%) confidence interval, as in global carbon budget analysis[2].

The decadal trends of ocean $CO_2$ sink and $\Delta fCO_2$ were estimated with the Theil-Sen slope estimator[61,62]. To calculate $\Delta fCO_2$, the

atmospheric $CO_2$ mole fraction ($xCO_2$) used in the global carbon budget analysis (data from the U.S. National Oceanic and Atmospheric Administration, Global Monitoring Laboratory[63]) is first converted to $pCO_2$ taking into account the atmospheric surface pressure corrected for the vapour pressure of seawater ($P_{atm}$), and then to $fCO_2$ with a fugacity coefficient estimated as follows[64],

$$fCO_2 = pCO_2 \cdot \exp\left(P_{atm} \cdot \frac{\left(B + (1 - xCO_2)^2 \cdot 2\delta\right)}{(R \cdot T)}\right) \qquad (1)$$

where $T$ is the sea surface temperature (in Kelvin, from OISST1.2[65]), $B$ and $\delta$ are the virial coefficients for carbon dioxide[66] and $R$ is the gas constant[64]. The necessary sea surface salinity data come from EN4 (EN.4.2.2.g10[67]), and the surface atmospheric pressure data come from ERA5[68].

For the GOBM and $fCO_2$-product ensembles (from the global carbon budget analysis), the amplitude of the decadal variability, and the decadal trends were calculated for each member of the ensemble. The mean ensemble values are reported with their standard deviation. For the hybrid approach, the amplitude of the decadal variability, and decadal trends were calculated with the annual constrained values. An estimate of the standard deviation around these values was obtained by re-calculating 10,000 times these estimates with annual ocean $CO_2$ sink values that had been randomly selected (from a uniform distribution) within the estimated confidence intervals of the annual constrained value.

### Description and evaluation of the NEMO-PlankTOM12.1 model
The NEMO-PlankTOM12.1 model consists of a global ocean general circulation model, NEMO v3.6, with an embedded biogeochemical model, PlankTOM12.1, forced by atmospheric meteorological data from the NCEP reanalysis product[30].

NEMO-PlankTOM12.1 used the NEMO model[69] in its global configuration on the ORCA tripolar grid, with a longitudinal resolution of 2° and an average latitudinal resolution of 1.5°, the latter being enhanced up to 0.3° in the tropics and at high latitudes, and a temporal resolution of 96 min. This physical ocean model comprises a total of 31 vertical z levels with a vertical resolution of 10 m for the first 100 m, decreasing progressively to a resolution of 500 m at a depth of 5 km. The NEMO model is based on the Navier-Stokes equations and a nonlinear equation of state. It explicitly calculates vertical mixing using a turbulent closure model. Subgrid-scale eddy-induced mixing is represented with a parameterisation[70]. NEMO is coupled to the Louvain-La-Neuve sea ice model (LIM[71]).

The PlankTOM12.1 biogeochemical model simulates the full marine cycles of carbon, oxygen, phosphorus and silicon, and simplified cycles for iron and nitrogen. This biogeochemical model was obtained by merging two versions of the PlankTOM model series, which had been developed in parallel, one focused on the role of jellyfish[72], and one focused on the role of pteropods[73]. This version has been used in the global carbon budget analysis 2022[39] and 2023[2]. Its ecosystem component is based on the representation of 12 Plankton Functional Types (PFTs), including six phytoplankton, five zooplankton and one bacteria. Spatiotemporal variations in PFT concentrations are induced by the simulated response of each PFT to environmental conditions, including temperature, nutrient availability, light, and interactions between PFTs. PlankTOM12.1 explicitly represents dissolved organic carbon and two size classes of particulate organic carbon, one small and one large. These components are influenced by the particle aggregation process, and the large particles are also influenced by the effect of mineral ballasting. Simulated dissolved inorganic carbon and alkalinity are influenced by air-sea exchanges of $CO_2$, calcification (production and dissolution), primary production, and remineralisation of organic matter (grazing by zooplankton and

remineralisation by bacteria). The alkalinity is also influenced by denitrification. A full description of PlankTOM12.1 biogeochemistry has been published[34]. Model simulations are too short to fully represent the input of river fluxes and their subsequent outgassing of $CO_2$ in the open ocean. Instead, constant river fluxes of dissolved and organic carbon and nutrients are prescribed as input at the location of river mouth, and corresponding fluxes are removed from the bottom sediments to conserve mass. The version of the NEMO-PlankTOM12.1 code used here is the same as that used in the latest global carbon budget analysis, forced with NCEP reanalysis. For this, the model was spun up first from 1750 to 1947 with a 30 years (1948–1977) climatological annual cycle of atmospheric forcing from the NCEP reanalysis product, followed by the use of annual forcing from 1948 until 2022.

The validation of this model version was first carried out by ensuring that the simulated surface chlorophyll-*a* concentration, primary production, and nutrient distributions were reasonably simulated, as in previous model versions[74]. Second, we examined the RMSE relative to the SOCAT gridded $fCO_2$ observations and the temporal variability of the ocean $CO_2$ sink between 1990–2022. These two variables are used to evaluate GOBMs in the global carbon budget analysis. The RMSE value associated with NEMO-PlankTOM12.1 (38.5 µatm) is within the range of GOBMs used in the global carbon budget analysis (31.3 µatm–45.0 µatm). The interannual and decadal variabilities of the ocean $CO_2$ sink from NEMO-PlankTOM12.1 are also comparable to the other GOBMs of the global carbon budget (Supplementary Table 1), and the simulated mean ocean $CO_2$ sink in the 1990s (1.91 Pg C yr$^{-1}$) falls within the observational range (1.5 to 2.9 Pg C yr$^{-1}$)[3].

Finally, the performance of NEMO-PlankTOM12.1 was evaluated with the metrics adopted by the global carbon budget in 2023: the simulated Atlantic Meridional Overturning Circulation, Southern Ocean sea surface salinity, the Southern Ocean stratification index, and surface ocean Revelle factor. The values simulated by NEMO-PlankTOM12.1 are within the range of the values simulated by the other GOBMs and are close to the observed values, with the exception of the Southern Ocean stratification index for which NEMO-PlankTOM12.1 has the lowest value but remains within comparable range (Supplementary Fig. S3).

## Hybrid approach

A hybrid approach is developed to constrain the annual ocean $CO_2$ sink simulated by the NEMO-PlankTOM12.1 model on the basis of the model-observation mismatch for surface $fCO_2$. This approach is not implemented to significantly improve the model-observation mismatch, but to correct for annual biases in the simulated ocean $CO_2$ sink, after the standard simulation is done, thus providing an adjusted annual estimate with uncertainty. This methodology has been previously used to constrain the climatological ocean primary production[33] and air-sea fluxes of $N_2O$[34] and $CCl_4$[35]. Because there are more observations available for surface $fCO_2$ observations than for $N_2O$ or $CCl_4$, this hybrid approach can be performed annually.

The surface $fCO_2$ observations used here are the ones compiled within the SOCAT v2023 database[75]. This database is a gridded product (1° × 1°) with a monthly temporal resolution. All monthly model outputs used here were regridded to the same spatial resolution as SOCAT.

First, to perform this hybrid approach, four perturbed simulations of NEMO-PlankTOM12.1 are produced with higher average MSEs (between the model and SOCAT observations, with equal weight given to each gridded observational data) over the simulated period, and lower or higher annual ocean $CO_2$ sink. These perturbed simulations range from 1.3 Pg C yr$^{-1}$ to 3.2 Pg C yr$^{-1}$ on average during 2000–2022, and span the expected range suggested by the global carbon budget analysis (i.e., 2.6 ± 0.4 Pg C yr$^{-1}$ on average, with

individual years ranging from 1.8 Pg C yr$^{-1}$ to 3.0 Pg C yr$^{-1}$). They are obtained by perturbing model parameters, changing the half-saturation constant for bacteria remineralisation of organic carbon from $5 \times 10^{-6}$ mol L$^{-1}$ to $18 \times 10^{-6}$ mol L$^{-1}$. This parameter was chosen because of its strong influence on the ocean $CO_2$ sink that is relatively uniform over the entire ocean. Second, for each year, a plot of the annual MSE values (on the $y$ axis) against the annual ocean $CO_2$ sink ($x$ axis) associated with the four model simulations (the optimal simulation and the four perturbed simulations) is produced and a cubic function is fitted through these data points. Third, the constrained annual ocean $CO_2$ sink is estimated by finding the local minimum (turning point) associated with the concave upward section of the fitted cubic function. This local minimum corresponds to a theoretical model simulation with an annual ocean $CO_2$ sink that presents the smallest MSE ($MSE_{min}$). Note that if the fit did not have a local minimum the ocean $CO_2$ sink from this year is not constrained. Years with a constrained ocean $CO_2$ sink not within the range of the ocean $CO_2$ sink from the perturbed simulations are kept but considered uncertain values.

Finally, the ± 1σ (68%) confidence interval associated with the determined $MSE_{min}$ value is estimated with this formula[33],

$$\frac{MSE_{68\%}}{MSE_{min}} = 0.468 \times \frac{n}{(n-2)} \times \sqrt{\left(\frac{2(2n-2)}{n(n-4)}\right)} + \frac{n}{(n-2)} \qquad (2)$$

where $n$ is the number of gridded observational data points, and $MSE_{68\%}$ corresponds to the MSE value for theoretical model simulations located at the borders of the confidence interval, and their associated annual ocean $CO_2$ sink is estimated by using the cubic function (i.e., determined where $f(x) = MSE_{68\%}$ with $x$ being the annual ocean $CO_2$ sink). See Supplementary Fig. S4 for a graphical interpretation of this hybrid approach. The global performance of the hybrid approach is evaluated with the two standard metrics used by the global carbon budget annual analysis[2]—i.e., the RMSE relative to the SOCAT $fCO_2$ observations and the estimated temporal variability of the ocean $CO_2$ sink (results in Supplementary Table 1).

The hybrid approach reduced the RMSE between NEMO-PlankTOM12.1 and SOCAT observations (the RMSE values associated with the perturbed simulations, PlankTOM12.1 and the hybrid approach are 39.9 µatm, 38.5 µatm and 38.0 µatm, respectively). For comparison, the GOBMs and $fCO_2$-products listed in the global carbon budget analysis in 2023 had mean RMSE values of 39.0 µatm and 20.3 µatm, respectively. Note that $fCO_2$-products were based on SOCAT observations and are therefore not independent, which explains their lower RMSE values.

This hybrid approach is also used to constrain regional ocean $CO_2$ sink values at three latitude bands, and at every 5° of latitude. For this regional analysis, the number of available SOCAT observations is lower, therefore only the quadratic function is used to provide wider confidence intervals and reflect the lower confidence in the constrained values.

We performed sensitivity analyses to test the robustness of our results to the choice of perturbed model parameters and model configurations. The perturbed simulations were repeated with parameters of phytoplankton respiration, and with a combination of both bacterial half-saturation and phytoplankton respiration. The model configuration was changed by using ERA5 reanalysis as weather-forcing data. In total, we thus have applied the hybrid approach to six different set ups, with three choices of perturbation parameters and two choices of forcing configurations (Supplementary Fig. S5). Regardless of the parameter and configuration used, the results consistently produced the lowest trend in the 1990s, and a higher trend in the 2000s than in the 2010s, although the exact trends within each decade varied (Supplementary Fig. S6 and Supplementary Table 3). We show here results of the model forced with NCEP, which has a lower RMSE (38.5 µatm)

compared to the configuration forced with ERA5 (40.0 µatm), and the perturbation of the half-saturation constant of bacterial remineralisation, which produces changes in $fCO_2$ that are more uniform across the ocean.

Finally, we also carried out an analysis to assess the influence of the annual application of the hybrid approach by considering, instead, three consecutive years. Therefore, rather than using a series of plots of the annual MSE value against the annual ocean $CO_2$ sink (e.g., Supplementary Fig. S4), we used a series of plots of the 3-year MSE value against the 3-year average ocean $CO_2$ sink.

## Data availability
Data from NEMO-PlankTOM12.1 and hybrid approach generated in this study have been deposited: https://osf.io/2kzps/?view_only=6ad809f1887342a0a19907e40a33e7cf. All other data is publicly available and instructions on how to access it are published on the following GitHub repository: https://github.com/nmayot/hybrid_approach.

## Code availability
The code used to perform the analysis is publicly available on a GitHub repository: https://github.com/nmayot/hybrid_approach.

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

## Acknowledgements

N.M. and C.L.Q. acknowledge the funding from the European Commission through the H2020 project 4 C (grant no. 821003). C.L.Q. was

funded by the UK Royal Society (grant no. RP\R1\191063). N.M. and R.M.W. were funded by UK's Natural Environment Research Council (SONATA: grant no. NE/P021417/1). The research presented in this paper was carried out on the High-Performance Computing Cluster supported by the Research and Specialist Computing Support service at the University of East Anglia. Funding to J.H. was provided by the Initiative and Networking Fund of the Helmholtz Association (Helmholtz Young Investigator Group Marine Carbon and Ecosystem Feedbacks in the Earth System [MarESys], Grant VH-NG-1301), by the ERC-2022-STG OceanPeak (Grant 101077209) and by the European Union's Horizon Europe research and innovation programme under Grant 101083922 (OceanICU Improving Carbon Understanding). The work reflects only the authors' view; the European Commission and their executive agency are not responsible for any use that may be made. We acknowledge the Global Carbon Project, which is responsible for the global carbon budget, and we thank the ocean modelling and $fCO_2$-mapping groups for producing and making available their model and $fCO_2$-product output. The SOCAT is an international effort endorsed by the International Ocean Carbon Coordination Project (IOCCP), the Surface Ocean Lower Atmosphere Study (SOLAS), and the Integrated Marine Biosphere Research (IMBeR) programme to deliver a uniformly quality-controlled surface ocean $CO_2$ database. The many researchers and funding agencies responsible for the collection of data and quality control are thanked for their contributions to SOCAT.

## Author contributions

N.M. and C.L.Q. conceived the study. C.L.Q., E.T.B. and R.M.W. designed and performed the NEMO-PlankTOM12.1 model simulations. D.C.E.B. supervised the collation and use of the SOCAT database. J.H. supervised the use and interpretation of the global carbon budget database. E.T.B. developed the theory of the hybrid approach. N.M. carried out the implementation of the hybrid approach and performed all complementary analysis. N.M. wrote the manuscript with input from all authors. C.L.Q. supervised the study.

## Competing interests

The authors declare no competing interests.
