## [Peer Review File · Nature Communications]

Constraining the trend in the ocean CO₂ sink during 2000-2022REVIEWER COMMENTS

Reviewer #1 (Remarks to the Author):

Authors Nicolas Mayot, Erik T. Buitenhuis, Rebecca M. Wright, Judith Hauck, Dorothee C. E. Bakker, and Corinne Le Quéré

Title Constraining the trend in the ocean CO₂ sink during 2000-2019

Manuscript NCOMMS-23-48601

Key results:

Mayot et al. present a hybrid method fusing processed-based models and observations to constrain the trend in oceanic CO₂ uptake for the period 2000-2019 to 0.30 ± 0.05 Pg C yr⁻¹ decade⁻¹. Critically, this revised trend in the ocean CO₂ sink enables them to infer the terrestrial CO₂ sink to 0.33 ± 0.12 Pg C yr⁻¹ decade⁻¹ over the same 2000-2019 period.

The hybrid model forces the state-of-the-art NEMO-PlankTOM12.1 GOBM with NCEP reanalysis data and observed atmospheric CO₂ concentrations. It provides an annual estimate of the ocean CO₂ sink by constraining the NEMO-PlankTOM12.1 yearly outcome with SOCAT observations

Validity:

The overall analysis appears sound, but there are several discrepancies that require better explanation and justification. The hybrid results for the 1990s and 2000s react significantly to the fCO₂ data, adjusting the NEMO-PlankTOM12.1 results into closer agreement with the fCO₂ data, as expected. However, the 2010s trends not only appears to disregard the data, but drives the NEMO-PlankTOM12.1 trend 50% lower and in the opposite direction of an adjustment towards the observations. This result dominates the 2000-2019 trend analysis as well, where the hybrid model again adjusts the final result away from the data trend. The explanations given for this in the text is not compelling – if the hybrid approach did disregard the fCO₂ observations, then it should have delivered a result very close to the NEMO-PlankTOM12.1 result.

The discussion surrounding the North and South regions and how the hybrid model weighted the observations in these regions is also inconsistent. The authors clearly state that there are more observations in the North, yet the hybrid model does not appear to weight them heavily as the hybrid results in Figs 1b and 3b largely follow the NEMO-PlankTOM12.1 lines. Yet the hybrid results do track the fCO₂ data and not the NEMO-PlankTOM12.1 results in the Southern region (Figs 1d, 3d) despite the fact that these data are sparser and potentially biased due to undersampling (Section 4). This needs to be explained with greater clarity since the high latitude regions dominate the global results (Figs 1a, 3a)

Significance:

The results of this study will have a major impact on the understanding and quantification of the global ocean CO₂ sink and its implications for quantifying the terrestrial CO₂ sink. It should be published after revisions.

Data and Methodology/Analytical Approach:

The approach and data use appear reasonable. Previous reviews have in principle vetted the hybrid approach used here (Refs 23-25).

Suggested improvements

Improvements are suggested below in the detailed line by line comments. In particular, there are several important results and conclusions that the authors should consider adding to the Abstract.

Detailed Comments:

Line 10 The study period is 2000-2019, but the first decade mentioned in the abstract is the 1990s. The report should include the entire period analyzed, unless strong justification is provided for focusing on a particular sub-period.

Lines 25-26, Ref 5 “However, the ocean carbon reservoir is also sensitive to climate variability and climate change”

The outcome reported by McKinley et al. [2020] - that there will be an immediate reduction in ocean carbon uptake as atmospheric pCO₂ responds to cuts in anthropogenic emissions - seems more relevant to the theme of this paragraph than the review paper by Gruber et al since the McKinley study deals explicitly with the timescale and magnitude of the ocean response to changes in anthropogenic CO₂ emissions

McKinley, G.A., Fay, A.R., Eddebar, Y.A., Gloege, L. and Lovenduski, N.S., 2020. External forcing explains recent decadal variability of the ocean carbon sink. *Agu Advances*, 1(2), p.e2019AV000149.

Line 56 “...last two decades...”

Lines 56-59, Table 1 “the two types of estimates diverge over the last two decades (2000-2019) 19–21, with the fCO₂-product ensemble suggesting a decadal rate of growth of the ocean CO₂ sink of 0.68 ± 0.13 Pg C yr⁻¹ decade⁻¹, double the 0.35 ± 0.05 Pg C yr⁻¹ decade⁻¹ simulated by the GOBMs ensemble (Table 1)”

Note: the hybrid model derived trend, 0.30 ± 0.05 Pg C yr⁻¹ decade⁻¹, is very close to the GOBM ensemble value of 0.35 ± 0.05 Pg C yr⁻¹ decade⁻¹, which makes one question how much influence the observations have on the hybrid model result, particularly since one would expect the observation-based trend of 0.68 ± 0.13 Pg C yr⁻¹ decade⁻¹ to drive the hybrid model result higher than the GOBM ensemble result, not lower.

Table 1 The first 2 columns provide information on the 1990-2019 trends, but these are largely ignored in the analysis

Table 1 The -50% adjustment that the Hybrid approach makes to the decadal trend in the 2010s ($0.26 \rightarrow 0.12 \pm 0.12$) makes no sense given the fCO₂ value of 0.79 ± 0.41 . One would expect a positive adjustment of ~0.4 in line with the adjustment that the method made in the 2000s ($0.31 \rightarrow 0.70 \pm 0.12$ with fCO₂ value of 0.81 ± 0.40). This must be justified as the 2010s trend also appears to distort the 2000-2019 trend significantly lower than the NEMO-Plank estimate and the fCO₂ trend

Lines 185-194 do not justify or even explain this result, they only note it.

Line 92-93 "... by constraining the model output fields of fCO₂ against the observed fields"

The authors should quantify the strength of the fCO₂ constraint given the sparse nature of the observed fCO₂ fields (Lines 50-51, Ref 16) and the fact that machine learning has been used to extrapolate the observations. How much of the hybrid result is due to the actual observations vs the extrapolation method?

Lines 93-96 As above, it is unclear how much the ocean CO₂ sink estimate is constrained by the actual (sparse) observations vs the machine learning method used to extrapolate fCO₂ fields

Lines 145-47, Fig 1 "The hybrid approach substantially modified the simulated ocean CO₂ sink in the high latitude regions, with the tropics remaining similar to the original NEMO-PlankTOM12.1 results"

Fig 1d shows that the Southern oceans (Hybrid – NEMO-Plank TOM ~ 0.5 PgC yr⁻¹) drive most of the global difference between the NEMO-Plank TOM and the hybrid results (Fig 1a); the Northern oceans contribute only ~0.1 PgC yr⁻¹ to the model-hybrid difference (See lines 157-158)

Fig1 Suggest plotting Figs 1b-1d with the same y-axis scale (-0.5 to 2.5) to illustrate the magnitude of the fluxes in each domain and their relative importance to the global result (Fig 1a). This will also emphasize the degree of agreement between NEMO-Plank TOM and the Hybrid model in each region

Line 152-53 It looks like the Hybrid approach increased the Northern ocean CO₂ sink by ~0.1 Pg C yr⁻¹, not 0.01 PgC yr⁻¹ (Fig 1b)

Lines 157-158 Suggest adding "the mismatch between the model and the fCO₂ observations in the 35-55°S band is the main cause of the underestimation of the ocean CO₂ sink in the South and globally" to the abstract

Fig 3 As with Fig 1, suggest plotting Fig 1b-d with the same y-axis

Fig 3b,d Why does the hybrid product track the fCO₂ values in the Southern region (Fig 3d) but not the Northern region (Fig 3b), where it tracks the NEMO-Plank TOM results instead? This appears to be a breakdown of the hybrid approach, especially given the high density of observations in the Northern region (Lines 200-202)

Fig 5, Line 208 Arctic Ocean not included in analysis

Lines 247-250 Deficiencies of the GOBMs in the high latitudes are a major result of this analysis and should be emphasized more. These deficiencies also demand increased sampling in the high latitudes to confirm the hybrid results presented here and correct GOBM deficiencies.

Lines 273-74 "Our results suggest that fCO₂-products overestimate the decadal trend of the 2010s in regions where there are few observations"

This suggests that the hybrid method defaulted to the NEMO-Planck TOM model in undersampled high latitudes in both hemispheres; however, there should have been some adjustment made for the actual observations even if they are sparse.

Lines 281-84 The authors appear to assume that the observations in the Southern Ocean are biased based on Refs 31 and 33, but the hybrid model locks onto the measurement data (Fig 3d). If the observations are biased, then why trust the results of the hybrid method in the Southern region? A similar argument about biased observations in the North carries less weight given there are more observations in the 30-90N latitude range than the 30-90 S range. This is a significant inconsistency that is not clearly resolved in the text.

Lines 293-95 "GOBMs represent the correct processes, but either they do not respond sufficiently

to changes in external forcing, or the balance among thermal and non-thermal processes in response to that forcing is imperfect.”

This is a general statement, but should address NEMO-Planck TOM specifically given its central importance in the hybrid method

Lines 320-322 Suggest adding “the stronger trend in the 2000s and the weaker trend in the 2010s estimated with the hybrid approach help to reduce the trend associated with the carbon budget imbalance between 2000 and 2019 from $-0.24 \text{ Pg C yr}^{-1} \text{ decade}^{-1}$ to $-0.003 \text{ Pg C yr}^{-1} \text{ decade}^{-1}$ ” to the Abstract

Line 323 How sensitive is the estimated trend in the terrestrial CO₂ sink to the value of the trend assigned to the ocean CO₂ sink?

Line 331 The authors miss the opportunity to advocate for more and more extensive fCO₂ observations, particularly at the high latitudes. Strongly urge that a final paragraph be added to Section 4 that addresses research needs unresolved by the current study.

Lines 389-90 “Model simulations are too short to fully represent the input of river fluxes and their subsequent outgassing of CO₂ in the open ocean”

This is a potentially serious model deficiency as riverine inputs will significantly alter coastal CO₂ uptake and will have a measurable impact on the global ocean CO₂ sink

Reviewer #2 (Remarks to the Author):

The manuscript addresses an important point regarding the disagreement between different estimates of ocean carbon and shows that data availability may be one of the culprits for the discrepancy.

However, in the current state, the manuscript is focused too much on technical details and lacks broader context, so it may be more suitable for a more methodology-oriented journals, such as Global Biogeochemical Cycles. Below are some of points that would need to be addressed to broaden the context of the manuscript before I can recommend it for publication in Nature Communications.

1. Why was the comparison only done with NEMO-PLANK model? There are other (higher-resolution) ocean circulation models for physics-biogeochemistry coupling, for example, ECCO-DARWIN (Carroll et al, 2020, JAMES). This model is higher resolution than NEMO-PLANK and is data-assimilating. I suggest the authors compare how the global estimates from their models compare with ECCO-DARWIN and whether their hybrid approach performs better than ECCO-DARWIN.

2. As Nature Communications is for a broader audience, the authors should discuss the meaning of fCO₂ and the limitation of this metric in relation to ocean carbon storage in the introduction. Specifically, fCO₂ is an ocean-surface measurement and does not represent the changes in ocean carbon content at depth, importantly below the permanent pycnocline. In this context, the authors should discuss recent studies done for the ocean carbon storage in 3D space, e.g., Gruber et al (2019, Science), Zemskova et al (2022, Nat Comm).

3. The manuscript contains many acronyms and numbers, which deteriorates the readability of the manuscript. For example, are the acronyms A-IV and A-DV really necessary? The authors could just say "interannual" and "decadal" variability in words. I also suggest that the authors comb through the manuscript to limit the number of numbers that they present to those that are crucial to the central points of the paper. These points need to be emphasized, as they are currently lost in the plethora of values presented in text. This trimming would make the paper more suitable for a broader higher-impact journal like Nature Communications rather than a methodology-oriented journal.

Reviewer #3 (Remarks to the Author):

Mayot et al. review, November 2023

The manuscript by Mayot and coauthors evaluates discrepancies between the surface fCO₂ and air-sea flux of carbon in earth system models and observation-based data products. The authors use an optimization approach to identify estimates that are as consistent as possible with both observational and model constraints.

I think this manuscript will be of interest to the readership and appropriate in scope for Nature Communications. It is also timely and well written. I do have concerns about how particular methodological choices and resulting caveats are communicated. In general I think my concerns can be resolved by in text clarifications, though I do recommend inclusion of some diagnostics which may be more time consuming to produce.

My major comments are as follows (in order of my perception of priority):

A. The authors chose to exclude the coastal ocean from their analysis--everything within 3 degrees of the coast and shallower than 1000 m. Given the outsize role of these regions for marine carbon fluxes, I believe the quantitative consequences of that choice should be clarified and visualized for readers.

B. In my opinion, this paper would be more impactful and robust if the authors included the results of sensitivity analyses used to inform their approach. Specifically:

- i. Several possible mechanisms are proposed to drive discrepancies between models and data-products. Varying subsampling and model parameters would help identify the most likely of these mechanisms.
- ii. Presenting the results of the sensitivity analysis used to select and tune key parameters for optimization and perturbation--within the constraints identified in the literature--would increase confidence in the results.

C. I think the model manipulations described here constitute a sufficiently different approach and

parameter set that validation should be presented. e.g., an evaluation of how major variables like temperature and salinity, as well as fCO₂, differ between this and prior PlankTOM configurations and climatological hydrography.

D. Various results are compared, but statistical tests are not presented. Including these would help support interpretations of whether or not values are similar or meaningfully different.

E. I think the nudging/data assimilation approach could be better referenced to illustrate how it draws on or differs from prior literature.

F. The authors appear to use different sources and calculations for water and air fCO₂, instead of the concurrently and self-consistently calculated values from SOCAT. I recommend the authors evaluate whether it is possible to consolidate their data sources in this case.

Detailed comments by line number (because this is a long list, I don't expect responses to all of these).

52. It would be helpful to further distinguish what is meant by fCO₂-products here in contrast to observational data only. Are the authors only referring to gridded, interpolated, and climatological datasets that use different methods but all fundamentally rely on SOCAT? Are there additional substantive differences readers should be aware of?

57. What are the specific fCO₂ products being considered as part of the ensemble? And the specific GOBMs included in the ensemble? I think each product considered should be referenced and included in a table associated with this manuscript. If both ensembles are identical to those used in the Global Carbon Budget 2022 publication, then I think that this point should further be clearly stated in the inline text with an associated reference number.

65.

a. The result in the 2010s trend column--in which the result is significantly lower than the model ensemble, preferred model, or fCO₂-product ensemble and thus seems to contradict either observational or model evidence--strikes me as a red flag that requires clearer explanation in the text. I'm not sure section 3.4 is sufficient to convince me. Any proposed resolution would be more convincing with a model sensitivity study demonstrating a plausible mechanistic basis.

b. The larger interannual variability in the presented approach than the other models and data products is also concerning, though less so given the reported uncertainties.

c. I suggest a concise table title, rather than expanding this into a caption.

85. Given that this approach is so central to the analysis, I think a brief overview here (a couple more sentences), prior to the trailing methods section, would be appropriate.

90. The trends in the preferred model are 20-60% lower than the ensemble in each decade. Is it fair to call these similar trends? Or perhaps it makes sense to restate this as a similar multidecadal

trend, since the decadal trends diverge. I also suggest double checking the overall 2000-2019 NEMO trend--based on Fig. S1 it appears that either the overall trend or detrended decadal component trend should be a bit smaller than that of the ensemble (but I may be mistaken).

96. A caveat to this comparison to is that these prior studies performed sensitivity analyses across a wide range of parameters and included observational constraints (of parameters other than the objective variable) on the the tuned parameters. If this study does either, it was not obvious to me. So I think it is appropriate to include some comment on how those prior studies guided the authors approach, and any limitations.

103. Similar to my comment about line 85, I think the authors should introduce a concise summary of the perturbation and tuning approaches here, beyond stating that they occurred. This context seems important for helping readers understand why state estimates sometimes arise outside the bounds established by the perturbations.

113. This is a matter of personal preference, but now that the authors have defined this variability in terms of amplitude, could the acronyms be shortened to IV and DV? Or, see comment for line 133.

115. "Very close" = indistinguishable? Considering these as a one sided test of difference (with table of tests results in supplement?) might clarify the meaning of statements about similarity ('very close', 'close to', 'marginally overlapping', 'in line with', 'within the range of', etc.). For consistency of interpretation by readers, the authors might consider choose a single phrase to indicate not significantly different comparisons.

124. I think it would be useful to label the perturbed simulation lines with a descriptive label of how they were perturbed (e.g., microbial $K_{1/2} = \dots$). Perhaps this is semantic, but in the final sentence of the caption I think "constrained but outside the bounds of the perturbation experiments" might be more descriptive than 'uncertain constrained'.

133. Wouldn't a moving-average as described here often result in lower variability even without changes in the data distribution? I suggest that smoothing the data in this way is no longer a comparison of interannual variability in the same sense as it was previously introduced. To be very clear, the authors might decide to replace A-DV and A-IV, with a notation that clearly displays the timescale considered. e.g., $V_{\{10y\}}$, $V_{\{3y\}}$, $V_{\{1y\}}$.

156.

a. A mismatch may help diagnose mechanisms, but it is not itself a cause of anything. Nor does the presence of a mismatch--absent a clearly aligned expectation grounded in a particular mechanism--conclusively indicate that the southern mid-latitude CO₂ sink is underestimated in models as opposed to overestimated in sparse observations. I think this statement needs to be revised--perhaps the authors meant instead that the overall regional and global disagreements between data sources reflect these latitudes in particular rather than elsewhere.

b. At face value, this finding seems to contradict that of Fay et al. 2021 (cited by the authors as reference 27) with respect to the geographic specificity of model-data mismatches.

160. Are the uncertainty bounds smaller than the symbols on most of these bands? Does it make sense that they are this small given the large interannual and interdecadal variability previously discussed?

194. See comment for line 65a.

204. If relying on significance for argument, I suggest including test result(s) (null, p, test-statistic).

216. I think that this statement implicitly assumes a constant gas transfer velocity. I'm not familiar enough with global windspeed or momentum flux trends to know whether that is likely but there are observed and modeled changes in these parameters over the last few decades in at least some regions within this latitude band (e.g., Eastern boundary current regions). The assumptions here probably need some reinforcing citations.

219. a. Please include trends and relevant statistical tests to reinforce this conclusion.

b. To me, this distinction is a bit arbitrary (or at least reflect a choice of emphasis). Visually, any given decade seems unlikely to diverge greatly from the overall 1990-2020 trend. i.e., if comparing each decade to null hypothesis of no difference from the secular trend, are any of these decadal variations actually significant?

225. a. I think this point about spatial bias in the interpolated products is valuable and it would be good to start a new paragraph here to visually highlight this idea.

b. This point (bifurcation of the interpolation products into high and low clusters) would be strengthened by a supplemental figure.

242. My perception is that statistical approaches to 'nudging' earth, ocean, and climate model results to incorporate observational constraints has a long history, which should probably be referenced here. Examples and context include: <http://dx.doi.org/10.5065/D60Z716B>, [https://doi.org/10.1175/1520-0493\(2000\)128<3664:ASNTFD>2.0.CO;2](https://doi.org/10.1175/1520-0493(2000)128<3664:ASNTFD>2.0.CO;2), <https://www.myroms.org/forum/viewtopic.php?t=320>, <https://doi.org/10.5194/gmd-16-1857-2023>.

The author's will note that these sources point out that nudging to observations is sometimes considered a suboptimal data assimilation strategy. I recognize that the model optimization used here is more complicated, and may not be well summarized as "nudging".

Other works have used more data assimilation techniques to incorporate marine carbonate system variables in concert with physical parameters: <https://doi.org/10.1016/j.jmarsys.2016.02.011>, <https://agupubs.onlinelibrary.wiley.com/doi/10.1002/2016JC012650>, <https://agupubs.onlinelibrary.wiley.com/doi/full/10.1029/2009GB003531>, <https://www.jstor.org/stable/24860162>, <https://www.jstor.org/stable/24861004>.

Given this context, I think the authors need to not just reference related concepts and works of this nature, but more clearly delineate how this work is novel in approach and focus. I acknowledge that

the length-limitations and format of this journal may have obscured significant advances in the approach used by the authors--if so and I missed these, I take that as further evidence that these need to be clearly and concisely delineated in the text.

258. Please see comment regarding line 156a. Without testing one of the mechanisms proposed, I think that this statement is not fully supported as written.

324. This is unclear to me as written. I suggest including an equation and calculated fluxes (and their data sources) in the supplement.

348. SOCAT reports both the wet and dry atmospheric fCO₂ measured concurrently with the seawater fCO₂ selected by the authors, eliminating the need for this calculation and the potential errors in introduces because of discrepancies in choices of equation and virial coefficients used by different groups, differences in the spatial gridding, etc. I strongly recommend the authors revisit their data source and this calculation considering the alternative SOCAT availability--if this is not possible, please explain why (e.g., are there no comparable interpolated products for the SOCAT air fCO₂?).

363. Randomly selected with what distribution? I ask because the authors previously chose to use tests that are suitable for non-normal data. If the data is not well approximated by a normal distribution, that should be noted here and an appropriate distribution chosen for this Monte Carlo approach.

385. Some additional details about the treatment of major inorganic carbon components would be appropriate here, even if covered in the reference.

398. a. If re-optimized as described, this is a new model variant that really requires independent validation from prior published works. Did this process knock any parameters back out of reasonable ranges? Are basic properties like T & S well described in at least a climatological sense? I acknowledge model validations are a lot of work, but as a reviewer it is difficult for me to have confidence in the results presented here without validation of parameters beyond fCO₂ of this new product presented in the supplement.

b. This optimization scheme is empirical but it isn't clear to me that the optimized parameters are realistically constrained. For example, if I am interpreting the temperature sensitivities correctly (e.g., based on parameter description by Le Quere 2016), that's equivalent to a Q₁₀ values that are lower than typically measured or modeled on either the organismal or ecosystem level (usually 2-3, the values here are equivalent to 1-2). Another example is the use of a constant e-folding scale for light, even though this varies widely in the ocean. Things like this make me question whether appropriate constants are being used and if the model optimization is able to distinguish between compensatory drivers.

c. Also, Table S1 does not include the 47 parameters that the text says were optimized. I think that if other variables were unchanged (I think only changed parameters have been tabulated?), then the text should be updated and the table clear on what is included and why.

403. Please elaborate on why these parameters were chosen, at least in a supplemental text. That makes these choices more transparent, and helps other modeling teams understand which differences are important.

410. I would prefer representative results to back up this claim were included.

422. But Table S1 appears to show changed physical parameters as well...please resolve this (appearance of) inconsistency.

425. Is this different from the A-IV discussed earlier? If so please consider removing the new acronym, as it is only used a couple times in the methods.

456. What is the order of magnitude of observations required to do this robustly?

460. a. This is a choice that I think should be clearly communicated in the main text, and as a mask or outline in the maps in Fig. 5 and Fig. S3. It is unclear from this whether the model outputs and data products are similarly masked out in the same coastal band, so please clarify that these analyses are comparing like to like.

b. Areas of <1000 m depth and within 3 degrees of shore are disproportionately important for marine carbon cycling. Ignoring them in order to characterize more pelagic regions may make it easier to identify trends and present more constrained appearing values, but at the cost of considering real, environmentally relevant variability and significant contributions to the global carbon flux. Reading this, I have diminished confidence in claims about the changed magnitude of the climatological flux (and inferred land flux), trends (which may be compensated or exacerbated by coastal processes), and the relative agreement of variability estimates. I think that this is too important a caveat to be buried in the Methods and not discussed in the Discussion.

c. How do the results change if coastal areas are considered as well, or separately? I would like to see even a quick discussion of this and followup plots in the supplement. It may be that focusing on the pelagic ocean is defensible and interesting, but this would be a more compelling manuscript if readers were able to consider the coasts as well--and if the authors were to point out some potential open questions that arise from those considerations.

467. Is this the most relevant variable to change?

Could the authors present sensitivity analysis supporting the selection of this parameter instead of/without others?

What are the actual values of the perturbations used? This doesn't seem to be reported anywhere, but needs to be.

Is it reasonable to assume a constant value (as shown in Table S1) or are biogeographic variations in this (and other constants) expected?

What are the observational constraints on this parameter and how were the magnitudes of changes selected within those constraints?

I think this choice needs significantly more explanation. There many fit results (in time and space) plotted outside the range of these perturbed models, which leads me to ask if the perturbations really are providing useful bounds or not--and if they are, whether the hybrid approach would

benefit from additional constraints or flexibility in parameters. I understand that this is a poorly constrained parameter, but there does seem to be a finite range that is consistent with measured ocean DOC values, which could help justify bounds on the perturbations.

470. Please provide a representative graphical example of this approach in the supplement (or a couple, a good fit and a bad fit).

477. Related to comment on line 467, I'd like to see some more quantitative assessment of the statement here.

486. There is no Fig. S4 in the supplement provided for review, but this would be welcome and would address my request for line 470 as well.

501. What quadratic function? Is there a mathematical justification for a different calculation? i.e., is this an arbitrarily larger confidence bound?

Fig. S1. The letters above the last two panels are repeated.

Fig. S3. I suggest making these panels larger.

In the data link, the hybrid results have a mislabeled column 1 in the tab for results by latitude.

The code looks plausible (I did not attempt to run it), and additional annotation would be appreciated to help users looking to translate this to an open source programming language for testing.

Dear Editor,

Thank you for inviting a revision of our work for further consideration in your journal. We sincerely thank the reviewers for their invaluable comments on our manuscript. Our revision addresses all the points raised by the reviewers and reduces the use of acronyms and technical language to enhance accessibility.

In particular, following your recommendation and the reviewer's comments, we have made the following important improvements:

- We conducted and detailed a sensitivity analysis of the hybrid approach, which shows that results on decadal trends are robust to the choices of parameters and model set up.
- We have included a validation of our model based on the model evaluation process used in the Global Carbon Budget 2023, which demonstrates that the NEMO-PlankTOM12.1 model is appropriate for estimating the evolution of the oceanic CO₂ sink, and that it is largely in line with the results of other models.
- We have improved the description of the hybrid approach, both in the Methods and in the main text.
- We have used all available fCO₂ observations including those from the coastal zones (which were excluded in the original submission).
- We have clarified the presentation of the results about the North and South regions.
- We have included a new figure that illustrates the low robustness of the 2010s trend in the ocean CO₂ sink estimated with fCO₂ products, giving even more weight to our own conclusions using our hybrid approach.

Furthermore, since the submission of our manuscript, the Global Carbon Budget 2023 (GCB 2023) has been published. We thus updated our analysis to use the latest available estimates for the NEMO-PlankTOM12.1, the Global Ocean Biogeochemistry Models (GOBMs) and the fCO₂-products published in GCB 2023, and expanded the period of analysis to year 2022. Our main focus is now on constraining the ocean CO₂ sink trend during 2000-2022 (instead of 2000-2019 in the original submission). Finally, we limited the use of the hybrid approach to start in 1980 as the number of fCO₂ observations prior to that were extremely low and discontinued.

All these modifications strengthen our results and improve the readability of our manuscript. See below our point-by-point response to each reviewer, with the reviewer's comments in blue, the new text in red and our response in black.

Response to Reviewer #1:

The overall analysis appears sound, but there are several discrepancies that require better explanation and justification. The hybrid results for the 1990s and 2000s react significantly to the fCO₂ data, adjusting the NEMO-PlankTOM12.1 results into closer agreement with the fCO₂ data, as expected. However, the 2010s trends not only appears to disregard the data, but drives the NEMO-PlankTOM12.1 trend 50% lower and in the opposite direction of an

adjustment towards the observations. This result dominates the 2000-2019 trend analysis as well, where the hybrid model again adjusts the final result away from the data trend. The explanations given for this in the text is not compelling – if the hybrid approach did disregard the fCO₂ observations, then it should have delivered a result very close to the NEMO-PlankTOM12.1 result.

For clarification, the hybrid approach uses directly the fCO₂ observations from the SOCAT database, not the interpolated and extrapolated version of these observations proposed by the fCO₂-products. Therefore, there is no assumption that the obtained estimates from the hybrid approach for the ocean CO₂ sink should follow the estimates from the fCO₂-products at global or regional scale. A difference between the hybrid approach estimate and the fCO₂-product estimate could, for example, be linked to imperfect interpolation and extrapolation of fCO₂ observations by fCO₂-products. We also would like to mention that the hybrid approach is applied similarly for each year, meaning that it cannot disregard the fCO₂ observations for a specific period. A graphical interpretation of the hybrid approach method is presented in Fig S4 (this figure was regrettably missing in our first submission).

The discussion surrounding the North and South regions and how the hybrid model weighted the observations in these regions is also inconsistent. The authors clearly state that there are more observations in the North, yet the hybrid model does not appear to weight them heavily as the hybrid results in Figs 1b and 3b largely follow the NEMO-PlankTOM12.1 lines. Yet the hybrid results do track the fCO₂ data and not the NEMO-PlankTOM12.1 results in the Southern region (Figs 1d, 3d) despite the fact that these data are sparser and potentially biased due to undersampling (Section 4). This needs to be explained with greater clarity since the high latitude regions dominate the global results (Figs 1a, 3a)

As mentioned above, we are not expecting that the hybrid approach results should exclusively follow the results from the fCO₂-products. In section 3.4, which has been revised by modifying figure 4, we identified discrepancies in the North between fCO₂-products, GOBM and the hybrid approach, and relate these discrepancies to potential imperfect interpolation and extrapolation of fCO₂ observations by fCO₂-products in under-sampled northern regions. For the South, in section 3.2, we demonstrate the deficiency of NEMO-PlankTOM12.1 in simulating a correct ocean CO₂ sink in this region.

The discussion paragraphs relating to these results (paragraphs 4 and 5 of section 4) have been revised. For the North, our suggested explanations are now clearly stated and we have added information on the potential influence of coastal regions. For the South, a new separate paragraph has been added, in which we clearly recognise the uncertainties associated with the lack of fCO₂ observations (new text in red): *“In the northern latitudes, where the availability of measurements is highest, the fCO₂-product ensemble gives a decadal trend in the 2000s not significantly different from that of the 2010s (Fig. 4). Four fCO₂-products suggest a growing or strong trend during the 2010s, contrary to the fCO₂ observations alone, which is explained here by their strong trends in areas that were undersampled during the 2010s (Fig. 5). Hence, we hypothesise that methodological issues in some fCO₂-products could lead to an unrealistic amplification of the ocean CO₂ sink trend in the 2010s (Bennington et al., 2022a). In addition, the ocean CO₂ sink in the northern region is also more influenced by coastal phenomena than the southern region, which despite their importance remain largely uncertain (Dai et al., 2022; Regnier et al., 2022;*

Laruelle et al., 2018; Mathis et al., 2024; Resplandy, 2024). Consequently, in the northern region, the way in which coastal fCO₂ observations are taken into account by the various fCO₂-products could induce some of the discrepancies among fCO₂-products. This would partly explain the lack of coherence between the GOBMs and the fCO₂-products over this recent decade (Friedlingstein et al., 2023; Gruber et al., 2023; Keppler et al., 2023; Müller et al., 2023).

In the Southern Ocean, our hybrid approach suggests that existing fCO₂ measurements could corroborate a strong and positive decadal trend in this region in the 2010s, and more generally between 2000-2022. However, the paucity of fCO₂ measurements impede our ability to evaluate the decadal trend using observations only (Gloege et al., 2021), but recent studies showed that undersampling could be responsible for strong biases in fCO₂-products in that region (Hauck et al., 2023a)."

And in the second to last paragraph: *"Moreover, it [our analysis] confirms the importance of high-density fCO₂ observations, which are notably lacking in the Southern Ocean, for informing the fCO₂-products and our hybrid approach."*

1) Line 10 The study period is 2000-2019, but the first decade mentioned in the abstract is the 1990s. The report should include the entire period analyzed, unless strong justification is provided for focusing on a particular sub-period.

Our main study period is 2000-2022 (expanded to 2022 to take account of the latest models and fCO₂-products made available in the global carbon budget 2023 update). The justification of this choice is that it is during this period that ocean CO₂ sink estimates differ the most between GOBMs and fCO₂-products: as reported in the global carbon budget analysis 2023 (Friedlingstein et al., 2023 - doi.org/10.5194/essd-15-5301-2023), ocean CO₂ sink trends from GOBMs and fCO₂-products have diverged by a factor of 2 since 2002, and by a factor of 2.5 since 2010. Furthermore, the number of fCO₂ observations considerably increased from the early 2000 (the number of monthly grid cells observed per year more than doubled between 2000 and 2005), making the discrepancy between various estimates more puzzling. Therefore our analysis helps to resolve this divergence.

The justification sentence of our introduction has been expanded to briefly explain the rationale for the foci of the paper, as follows: *"Here, we investigate the discrepancies between fCO₂-products and GOBMs, focusing on the inconsistency, by a factor of two, of the 2000-2022 trend, which occurs despite the growing number of fCO₂ observations."*

2) Lines 26, Ref 5 "However, the ocean carbon reservoir is also sensitive to climate variability and climate change" The outcome reported by McKinley et al. [2020] - that there will be an immediate reduction in ocean carbon uptake as atmospheric pCO₂ responds to cuts in anthropogenic emissions - seems more relevant to the theme of this paragraph than the review paper by Gruber et al since the McKinley study deals explicitly with the timescale and magnitude of the ocean response to changes in anthropogenic CO₂ emissions McKinley, G.A., Fay, A.R., Eddebbar, Y.A., Gloege, L. and Lovenduski, N.S., 2020. External forcing explains recent decadal variability of the ocean carbon sink. *Agu Advances*, 1(2), p.e2019AV000149.

We have added a sentence to include specifically the role played by variability in atmospheric growth rate, as follows: “*This uptake is modulated by the variable growth rate in atmospheric CO₂ (McKinley et al., 2020). Furthermore, the ocean carbon reservoir is also sensitive to climate variability and climate change (Gruber et al., 2023)...*” The Gruber et al. (2023) paper is a review paper that summarises the factors responsible for trends and variability in the ocean CO₂ sink, which includes also those of McKinley et al. (2020).

3) Line 62 “...last two decades...” Done

4) Lines 61, Table 1 “the two types of estimates diverge over the last two decades (2000-2019) 19–21, with the fCO₂-product ensemble suggesting a decadal rate of growth of the ocean CO₂ sink of 0.68 ± 0.13 Pg C yr⁻¹ decade⁻¹, double the 0.35 ± 0.05 Pg C yr⁻¹ decade⁻¹ simulated by the GOBMs ensemble (Table 1)”

Note: the hybrid model derived trend, 0.30 ± 0.05 Pg C yr⁻¹ decade⁻¹, is very close to the GOBM ensemble value of 0.35 ± 0.05 Pg C yr⁻¹ decade⁻¹, which makes one question how much influence the observations have on the hybrid model result, particularly since one would expect the observation-based trend of 0.68 ± 0.13 Pg C yr⁻¹ decade⁻¹ to drive the hybrid model result higher than the GOBM ensemble result, not lower.

The fCO₂-products are not used by the hybrid approach, which instead uses only the fCO₂ observations published by SOCAT. Therefore, we are not expecting that the hybrid approach results should exclusively follow the results from the fCO₂-products. The hybrid approach is constrained by both the SOCAT observations and by the simulated processes affecting the carbon cycle in the ocean (as represented in the NEMO-PlankTOM12.1 model). In contrast, the fCO₂-products are closely linked to fCO₂ observations and are also sensitive to uncertainties in gas exchange parameterization and the scarcity of data. The extrapolation in data-poor regions performed by the fCO₂-products depends on the numerical methods chosen and does not need to respect physical processes.

Likewise, the GOBMs are not constrained by SOCAT observations, contrary to the hybrid approach. The fact that the hybrid approach deviates from the trends of both the NEMO-PlankTOM12.1 model and the GOBM ensemble shows that the hybrid approach can indeed be influenced by the observations and demonstrates that the method works and does not simply reproduce the results of the original NEMO-PlankTOM12.1 model.

Moreover, in this revised submission, values in Table 1 have been updated to take into account the latest information available. The value of the ocean CO₂ sink trend, between 2000 and 2022, obtained from the hybrid approach now lies between the GOBM and fCO₂-product estimates, while still slightly closer to the GOBMs. We explain the slightly larger updated trend by an additional 3 years as well as additional SOCAT data covering the later years, and by the use of an updated version of the NEMO-PlankTOM12.1 model which better fits the SOCAT observations. The closer proximity of the hybrid approach results to the fCO₂-products results stems also from the update of these products within the last three updates of the Global Carbon Budget analysis (see the end of section 3.3 and Figure 6).

5) Table 1 The first 2 columns provide information on the 1990-2019 trends, but these are largely ignored in the analysis

These results are mostly mentioned later in section 3.1, and discussed within two paragraphs of the discussion. Because the first two columns give information about the amplitudes of the interannual and decadal variations, those results are more reliable when using longer time series. We use these results in two ways. First to show that GOBMs likely underestimate variability (mentioned in the discussion). Second as an additional element to support the validity of the hybrid approach, because the hybrid approach reproduces the patterns of variability produced by the NEMO-PlankTOM12.1 model ($r = 0.5$, $p = 0.004$, Pearson's correlation coefficient) but enhances its variability (also in the discussion). We therefore left two columns of the table unchanged, but have reordered the information to be easier to follow.

6) Table 1. The -50% adjustment that the Hybrid approach makes to the decadal trend in the 2010s (0.26 ± 0.12) makes no sense given the fCO_2 value of 0.79 ± 0.41 . One would expect a positive adjustment of ~ 0.4 in line with the adjustment that the method made in the 2000s (0.31 ± 0.12 with fCO_2 value of 0.81 ± 0.40). This must be justified as the 2010s trend also appears to distort the 2000-2019 trend significantly lower than the NEMO-Plank estimate and the fCO_2 trend

We do not expect the hybrid approach to make the same adjustments between different years or decades, nor to follow the results of fCO_2 -products for the reasons mentioned above (the hybrid approach is constrained by both SOCAT observations and physical processes, while the fCO_2 -products are tightly linked to SOCAT observations and numerical extrapolations). While the NEMO-PlankTOM12.1 model may wrongly estimate a trend for one period, it could correctly estimate a trend for another period. This explains why the adjustments are different between different years and decades. It is also important to note that the estimate of the ocean CO_2 sink obtained for one year by the hybrid approach is not used by the hybrid approach to derive the estimate for the following or previous year, which again induces different adjustments for different periods.

7) Lines 182-185 do not justify or even explain this result, they only note it.

We report the most important results within the results section (including those of lines 182-185), and further explain and discuss them in the discussion section. For example, the trends in the 2010s are reported for the various methods in results section 3.3, the discrepancies are reported in detail in section 3.4 and are discussed in the second paragraph of the discussion. Note, we have added additional results concerning the evolution of the fCO_2 -product estimate over the last three Global Carbon Budget analyses for the 2010s trend.

8) Line 87-90 "... by constraining the model output fields of fCO_2 against the observed fields" The authors should quantify the strength of the fCO_2 constraint given the sparse nature of the observed fCO_2 fields (Lines 50-51, Ref 16) and the fact that machine learning has been used to extrapolate the observations. How much of the hybrid result is due to the actual observations vs the extrapolation method?

Lines 91-95 As above, it is unclear how much the ocean CO_2 sink estimate is constrained by the actual (sparse) observations vs the machine learning method used to extrapolate fCO_2 fields

As we use the $f\text{CO}_2$ observations as published in SOCAT, the results of the hybrid approach are not influenced by the use of an interpolation and extrapolation method. The fit minimises the RMSE between the NEMO-PlankTOM12.1 model output and SOCAT observations (that are provided on a 1x1 grid at monthly resolution). The same weight is given to all observations. We have reformulated the sentence above to make this clear. The sentence now reads: “...goes beyond the traditional model evaluation by constraining the model output fields of $f\text{CO}_2$ against the observed $f\text{CO}_2$ data provided by SOCAT”.

We also added a precision on the treatment of weight in the Methods section 5.3: “...average MSEs (between the model and SOCAT observations, with equal weight given to each gridded observational data)”

10) Lines 143-45, Fig 1 “The hybrid approach substantially modified the simulated ocean CO_2 sink in the high latitude regions, with the tropics remaining similar to the original NEMO-PlankTOM12.1 results” Fig 1d shows that the Southern oceans (Hybrid – NEMO-Plank TOM $\sim 0.5 \text{ PgC yr}^{-1}$) drive most of the global difference between the NEMO-Plank TOM and the hybrid results (Fig 1a); the Northern oceans contribute only $\sim 0.1 \text{ PgC yr}^{-1}$ to the model-hybrid difference (See lines 157-158)

We added a mention of the larger importance of the South in this sentence. Although the differences are not on the same order of magnitude between the different regions, it is also important to mention that the variabilities are mostly modified in the high latitudes regions rather than in the tropics. The new sentence now reads: “The hybrid approach substantially modified the simulated ocean CO_2 sink in the high latitude regions, particularly in the South, but with the tropics remaining similar to the original NEMO-PlankTOM12.1 results”

12) Fig1 Suggest plotting Figs 1b-1d with the same y-axis scale (-0.5 to 2.5) to illustrate the magnitude of the fluxes in each domain and their relative importance to the global result (Fig 1a). This will also emphasize the degree of agreement between NEMO-Plank TOM and the Hybrid model in each region.

Fig 3 As with Fig 1, suggest plotting Fig 1b-d with the same y-axis.

We modified Fig 3 as suggested. That figure now clearly shows the variability as well as the relative magnitude of the three regions. For Fig 1, we modified panels 1b-c to have the same y-axis scale, but kept Fig 1d unchanged, as the mean flux value is very different in the South and the presentation of error bars (also larger in the South) required a different and more extended y-axis. The point about the relative magnitude is now made by Fig 3. We also added a mention of the larger influence of the South region on the global results in the text: “...the mismatch between the model and the $f\text{CO}_2$ observations in the 40-60°S band could be the main cause of the underestimation of the ocean CO_2 sink in the South (Fig. 2). Note that it is the Southern region that has the most influence on the global ocean CO_2 sink.”

13) Line 152-53 It looks like the Hybrid approach increased the Northern ocean CO_2 sink by $\sim 0.1 \text{ Pg C yr}^{-1}$, not 0.01 PgC yr^{-1} (Fig 1b)

This part of the sentence has been deleted in the revised version in response to other comments.

14) Lines 154-156 Suggest adding “the mismatch between the model and the $f\text{CO}_2$ observations in the 35-55°S band is the main cause of the underestimation of the ocean CO_2 sink in the South and globally” to the abstract

Our manuscript focuses on explaining the trends in the ocean CO_2 sink. The suggested sentence, although interesting to explain the origin of a model bias, is peripheral to the main message of the manuscript. Because of the length restriction for the abstract for Nature Communication, we did not add the suggested information to the abstract.

Lines 373-375 Suggest adding “the stronger trend in the 2000s and the weaker trend in the 2010s estimated with the hybrid approach help to reduce the trend associated with the carbon budget imbalance between 2000 and 2019 from $-0.24 \text{ Pg C yr}^{-1} \text{ decade}^{-1}$ to $-0.003 \text{ Pg C yr}^{-1} \text{ decade}^{-1}$ ” to the Abstract

The reduction in the global carbon budget imbalance is subject to additional uncertainties that are not related to our analysis, but to the estimations of the other components of the global carbon budget (e.g., land sink). We therefore did not add the suggested information to the abstract. Furthermore, we would like to mention that a manuscript targeting specifically this question is in preparation within our research group. It will be submitted for publication soon.

16) Fig 3b,d Why does the hybrid product track the $f\text{CO}_2$ values in the Southern region (Fig 3d) but not the Northern region (Fig 3b), where it tracks the NEMO-PlankTOM results instead? This appears to be a breakdown of the hybrid approach, especially given the high density of observations in the Northern region (Lines 200-202)

As detailed in our response above, we are not expecting that the hybrid approach results should exclusively follow the results from the $f\text{CO}_2$ -products. For the South, in section 3.2, we demonstrate the deficiency of NEMO-PlankTOM12.1 in simulating a correct ocean CO_2 sink in this region. In section 3.4, which has been revised by modifying figure 4, we identified discrepancies in the North between $f\text{CO}_2$ -products, GOBM and the hybrid approach, and relate these discrepancies to potential imperfect interpolation and extrapolation of $f\text{CO}_2$ observations by $f\text{CO}_2$ -products in under-sampled northern regions.

The discussion paragraphs relating to these results (paragraphs 4 and 5 of section 4) have been revised. For the North, our suggested explanations are now clearly stated and we have added information on the potential influence of coastal regions. For the South, a new separate paragraph has been added, in which we clearly recognise the uncertainties associated with the lack of $f\text{CO}_2$ observations.

17) Fig 5, Line 225 Arctic Ocean not included in analysis

The figure caption was clarified and now reads: “*Decadal trends in $\Delta f\text{CO}_2$ in the North (>30°N, excluding the Arctic Ocean).*”

18) Lines 337-340 Deficiencies of the GOBMs in the high latitudes are a major result of this analysis and should be emphasized more. These deficiencies also demand increased sampling in the high latitudes to confirm the hybrid results presented here and correct GOBM deficiencies.

Indeed, as mentioned in this paragraph of the discussion, the lack of variability in GOBMs in high latitude regions is a major issue and for reasons that remain partially unknown. However it is already well-known, with several publications have specifically addressed this deficiency of the GOBMs (for example, DeVries et al., 2019, 2023; Hauck et al. 2020; Li et al., 2019; Friedlingstein et al. 2023). They are mentioned and discussed, but we added in the discussion that (lines 330-331) “*Additional fCO₂ sampling at high latitudes could help resolve some of these [high-latitude model] issues*”.

In our revised discussion, we also now reaffirm the lack of observations in the high latitudes regions: “*In the Southern Ocean, our hybrid approach suggests that existing fCO₂ measurements could corroborate a strong and positive decadal trend in this region in the 2010s, and more generally between 2000-2022. However, the paucity of fCO₂ measurements impede our ability to evaluate the decadal trend using observations only (Gloege et al., 2021), but recent studies showed that undersampling could be responsible for strong biases in fCO₂-products in that region (Hauck et al., 2023a).*”

“*Moreover, it [our analysis] confirms the importance of high-density fCO₂ observations, which are notably lacking in the Southern Ocean, for informing the fCO₂-products and our hybrid approach.*”

However, we consider that the main objective and major results of our manuscript concern recent trends in ocean CO₂ sink. To better highlight the focus on our main results, we have now rearranged the order of the paragraphs in the discussion to start with our results on trends in ocean CO₂ sink (the first 5 paragraphs and the last 2 paragraphs). The results relating to the variability of the ocean CO₂ sink are discussed afterwards (in 2 paragraphs).

19) Lines 299-300 “Our results suggest that fCO₂-products overestimate the decadal trend of the 2010s in regions where there are few observations” This suggests that the hybrid method defaulted to the NEMO-PlankTOM model in undersampled high latitudes in both hemispheres; however, there should have been some adjustment made for the actual observations even if they are sparse.

Actual observations are fully considered in the hybrid approach. Suggested errors in fCO₂-products had no influence on the hybrid approach, because we did not use them to constrain the ocean CO₂ sink simulated by NEMO-PlankTOM12.1. The hybrid approach constrains the simulated ocean CO₂ sink based on the SOCAT observations, even if observations are sparse (e.g., in the Southern Ocean). Moreover, this sentence was related to our analysis conducted in the North. We clarified our text in the sentence to avoid confusion: “*Our results suggest that some fCO₂-products overestimate the decadal trend of the 2010s in northern regions where there are few observations*”

20) Lines 315-317 The authors appear to assume that the observations in the Southern Ocean are biased based on Refs 31 and 33, but the hybrid model locks onto the

measurement data (Fig 3d). If the observations are biased, then why trust the results of the hybrid method in the Southern region? A similar argument about biased observations in the North carries less weight given there are more observations in the 30-90N latitude range than the 30-90 S range. This is a significant inconsistency that is not clearly resolved in the text.

The reviewer is pointing out that the hybrid approach is following the results from the fCO₂-products in the Southern Ocean, but not in the North. As mentioned earlier, the fCO₂-products results are not used by the hybrid approach to constrain the ocean CO₂ sink simulated by NEMO-PlankTOM12.1, only direct observations are used. Therefore, we do not expect the hybrid approach to produce estimates similar (or dissimilar) to those of fCO₂-products.

We agree that the estimates of the hybrid approach for the Southern Ocean remain to be confirmed due to the lack of measurements in this region. In our revised discussion, we have added a text arguing in favour of greater data collection, mainly in the Southern Ocean: “*In the Southern Ocean, our hybrid approach suggests that existing fCO₂ measurements could corroborate a strong and positive decadal trend in this region in the 2010s, and more generally between 2000-2022. However, the paucity of fCO₂ measurements impede our ability to evaluate the decadal trend using observations only (Gloege et al., 2021), but recent studies showed that undersampling could be responsible for strong biases in fCO₂-products in that region (Hauck et al., 2023a).*”

In contrast, we have more confidence in the hybrid approach estimates for the North, as it follows the temporal patterns visible in the SOCAT data, which are more abundant in this region. In addition, we have suggested deficiencies for some fCO₂-products in their estimates of the trend in the ocean CO₂ sink in the 2010s in the North (as presented in section 3.4, and discussed in section 4)

21) Lines 325-328 “GOBMs represent the correct processes, but either they do not respond sufficiently to changes in external forcing, or the balance among thermal and non-thermal processes in response to that forcing is imperfect.” This is a general statement, but should address NEMO-PlanckTOM specifically given its central importance in the hybrid method

We added the mention of NEMO-PlankTOM12.1 in the sentence, which now reads as follows “...suggest that *NEMO-PlankTOM12.1 and other* GOBMs represent the correct processes, but...”.

NEMO-PlankTOM12.1 exhibits deficiencies in the simulation of the Southern Ocean CO₂ sink similar to those of the majority of GOBMs used in the analysis of the global carbon budget, and we argue that the mention of other GOBMs in this sentence is justified. This was shown in particular by the analysis carried out by Hauck *et al.* (2023, <https://doi.org/10.1029/2023GB007848>). In this published analysis, NEMO-PlankTOM12.1 has been classified within the group of the DIC-weak GOBMs (as most GOBMs), where the strong underestimation of non-thermal processes causes these models to be too strongly temperature driven across the year. We have added this information in the manuscript to strengthen and clarify our justification: “*For example, in the Southern Ocean, ocean surface*

fCO₂ variations over the year in NEMO-PlankTOM12.1, and in most GOBMs, tend to be too strongly influenced by temperature changes (Hauck et al., 2023b)."

23) Line 375 How sensitive is the estimated trend in the terrestrial CO₂ sink to the value of the trend assigned to the ocean CO₂ sink?

The estimated trend in the terrestrial CO₂ sink is directly influenced by the trend assigned to the ocean CO₂ sink when estimating the land sink using the global carbon budget residual. In the global carbon budget analysis, the land CO₂ sink is estimated by an ensemble of Dynamic Global Vegetation Models. This estimate suggests a 2000-2022 trend of 0.34 ± 0.20 Pg C yr⁻¹ decade⁻¹. Based on the other terms of the global carbon budget, it is possible to derive another estimate of the land CO₂ sink and its associated trend using the residual of the other terms. This methodology for obtaining an estimate of the land CO₂ sink (S_{LAND}) with the carbon budget residual has been added in the supplementary information document "Equations of the Global Carbon Budget analysis:"

Within the Global Carbon Budget analysis, the carbon sinks (atmosphere = G_{ATM} , ocean = S_{OCEAN} , and land = S_{LAND}) and emissions (fossil fuel = E_{FOS} , and land use change = E_{LUC}) are estimated,

$$(G_{ATM} + S_{OCEAN} + S_{LAND}) = E_{FOS} + E_{LUC} \quad (1)$$

By using the estimates of G_{ATM} , E_{FOS} , and E_{LUC} from the Global Carbon Budget analysis published in 2023, with our estimate of S_{OCEAN} from the hybrid approach, an estimate of the S_{LAND} term can be obtained:

$$S_{LAND} = E_{FOS} + E_{LUC} - (G_{ATM} + S_{OCEAN}) \quad (2)$$

In the context of the global carbon budget, uncertainties related to land-use change (E_{LUC}) and the ocean sink for CO₂ (S_{OCEAN}) have the greatest influence on the S_{LAND} estimate. Therefore, our estimate target the uncertainty in the S_{LAND} related to S_{OCEAN} as mentioned in the manuscript by comparing our estimate derived from our hybrid approach with that which would be obtained with fCO₂-products: *"Our estimated trend in the land CO₂ sink is **between the 0.34 ± 0.20 Pg C yr⁻¹ decade⁻¹ trend** estimated by the Dynamic Global Vegetation Model ensemble used in the global carbon budget analysis (Friedlingstein et al., 2023) and the **trend of -0.01 ± 0.13 Pg C yr⁻¹ decade⁻¹** that would be obtained with the ocean CO₂ sink estimate from the fCO₂-products **alone.**"*

24) Line 385 The authors miss the opportunity to advocate for more and more extensive fCO₂ observations, particularly at the high latitudes. Strongly urge that a final paragraph be added to Section 4 that addresses research needs unresolved by the current study.

We have modified and strengthened our second to last paragraph of section 4: *"Moreover, it confirms the importance of high-density fCO₂ observations, **which are notably lacking in the Southern Ocean**, for informing the fCO₂-products **and our hybrid approach**. It suggests that fCO₂-products could be further improved by scrutinising the extrapolation of observations in the 2010s, **which are evolving over the different versions released**, in order to understand differences among fCO₂-products, and then help improve them. "*

25) Lines 449-450 “Model simulations are too short to fully represent the input of river fluxes and their subsequent outgassing of CO₂ in the open ocean” This is a potentially serious model deficiency as riverine inputs will significantly alter coastal CO₂ uptake and will have a measurable impact on the global ocean CO₂ sink

This treatment of river fluxes is conformed to the definition of the ocean CO₂ sink and the protocol used by the research teams providing ocean CO₂ sink estimates from GOBMs for the global carbon budget analysis. In the global carbon budget analysis, the river fluxes are derived from specific published estimates. The river flux influences the mean air-sea CO₂ flux, but would not directly influence its variability, which is the focus of the current analysis.

Response to Reviewer #2:

1) Why was the comparison only done with NEMO-PLANK model? There are other (higher-resolution) ocean circulation models for physics-biogeochemistry coupling, for example, ECCO-DARWIN (Carroll et al, 2020, JAMES). This model is higher resolution than NEMO-PLANK and is data-assimilating. I suggest the authors compare how the global estimates from their models compare with ECCO-DARWIN and whether their hybrid approach performs better than ECCO-DARWIN.

In our study, we compare the ocean CO₂ sink estimate of the hybrid approach with that of NEMO-PlankTOM12.1, but also with the sink estimates of the other nine GOBMs and seven fCO₂-products used in the global carbon budget analysis. In the manuscript, we highlight the difference between the hybrid approach and NEMO-PlankTOM12.1, since the hybrid approach constrains the ocean CO₂ sink simulated by NEMO-PlankTOM12.1. It is therefore important to note the changes made to the NEMO-PlankTOM12.1 model output by the hybrid approach.

Below, we made a comparison between ECCO-Darwin and the fCO₂-products, the GOBMs, NEMO-PlankTOM12.1, and the hybrid approach.

Table R1 | Decadal trends of the ocean CO₂ sink between 1995 and 2018 estimated by different methods.

	2000-2009 (PgC/yr/decade)	2010-2018 (PgC/yr/decade)	2000-2018 (PgC/yr/decade)
ECCO-Darwin	0.17	0.96	0.67
fCO ₂ -products	0.71 ± 0.38	0.67 ± 0.20	0.61 ± 0.18
GOBMs	0.40 ± 0.10	0.44 ± 0.12	0.37 ± 0.05
NEMO-PlankTOM12.1	0.27	0.60	0.46
Hybrid approach	0.80 ± 0.21	0.42 ± 0.19	0.51 ± 0.07

For the decadal trend, ECCO-Darwin is different from $f\text{CO}_2$ -products, GOBMs and the hybrid approach, with a very low value for the period 2000-2009 and a high value for the period 2010-2018. Overall, over the period 2000-2018, ECCO-Darwin suggests a high decadal trend, within the range of $f\text{CO}_2$ -products, but with unique temporal patterns that is not coherent with results from model ensembles presented here.

DeVries et al (2023) reported that ECCO-Darwin exhibits significant temporal variability in the ocean CO_2 sink and in the spatial variability of dissolved inorganic carbon (DIC) accumulation rates. They mentioned that ECCO-Darwin results are subject to caveats concerning the possibility of assimilation-induced model drifts affecting their results. For example, DeVries et al (2023) mentioned that changes in biogeochemical model parameters during assimilation can result in DIC gradients that are advected by the mean ocean circulation and could affect air-sea CO_2 fluxes and DIC accumulation. It is plausible that these potential model drifts explain the differences in decadal trends mentioned above because the data coverage evolves through time, but we are not able to make a firm assessment here because we cannot explore the robustness of the decadal trends to changing data coverage and/or to different model choices within ECCO-Darwin. Furthermore, the ECCO-Darwin model results presented here (from the RECCAP2 project) only cover the period 1995-2018, while our analysis covers three decades during the period 1990-2022. For all these reasons, we prefer not to include the results of the ECCO-Darwin model in our analysis and to limit ourselves to comparisons with other types of estimates used in the global carbon budget analysis, for which several different estimates are provided which can be analysed as ensembles.

However, we agree that more estimates from data-assimilation models are needed and can provide additional information to constrain the ocean CO_2 sink. For example, the Global Carbon Budget analysis in 2023 used estimates from four Earth System Models (ESMs) prediction systems, which are based on assimilating physical atmospheric and oceanic data products into the ESMs. These models were used as a new line of evidence in predicting the 2023 ocean and land CO_2 sinks, and the atmospheric CO_2 growth. For this reason, we have now added some references about ongoing research activity on the development of models using data-assimilation: “*Results from models using data-assimilation (Brasseur et al., 2009; Verdy and Mazloff, 2017), also suggest an underestimated decadal variability in GOBMs air-sea CO_2 flux (Carroll et al., 2020; Li et al., 2023).*”

2) As Nature Communications is for a broader audience, the authors should discuss the meaning of $f\text{CO}_2$ and the limitation of this metric in relation to ocean carbon storage in the introduction. Specifically, $f\text{CO}_2$ is an ocean-surface measurement and does not represent the changes in ocean carbon content at depth, importantly below the permanent pycnocline. In this context, the authors should discuss recent studies done for the ocean carbon storage in 3D space, e.g., Gruber et al (2019, Science), Zemskova et al (2022, Nat Comm).

We have added this information in the introduction:

“This has been made possible by the annual release of quality-controlled observations of CO_2 fugacity ($f\text{CO}_2$) at the sea surface — analogous to the partial pressure of CO_2 — compiled within the Surface Ocean CO_2 Atlas (SOCAT)...”

“It should be noted that these fCO₂-products only assess the air-sea CO₂ flux, and not where anthropogenic CO₂ is ultimately stored in the ocean, which would require additional measurements of carbon in the water column, as well as more assumptions (Gruber et al., 2019; Zemskova et al., 2022; Keppler et al., 2023; Müller et al., 2023).”

The discussion related to the ocean carbon content at depth, which provides a constraint mostly on the mean ocean CO₂ sink value than its yearly variations, was already present in the discussion section. We have extended it and included the references mentioned by the reviewer:

“Finally for the mean ocean CO₂ sink, the hybrid approach returns a higher mean CO₂ sink than NEMO-PlankTOM12.1 in the Southern Ocean because it corrects a consistent bias of overestimation of the surface ocean fCO₂. Studies based on emergent constraint properties have also suggested...

[...]

However, this means that our hybrid approach is less robust in estimating the mean ocean CO₂ sink than the variability and trend of this sink, because the mean ocean CO₂ sink also depends on mixing between the surface ocean and the deep ocean (Orr et al., 2001), a process that is weakly constrained when only using surface fCO₂ observations, as is the case in the hybrid approach. Further work, in particular the use and/or assimilation of ocean interior carbon data (Carroll et al., 2020; Li et al., 2023; Zemskova et al., 2022), would be better suited to constrain the mean ocean CO₂ sink. ”

3) The manuscript contains many acronyms and numbers, which deteriorates the readability of the manuscript. For example, are the acronyms A-IV and A-DV really necessary? The authors could just say "interannual" and "decadal" variability in words. I also suggest that the authors comb through the manuscript to limit the number of numbers that they present to those that are crucial to the central points of the paper. These points need to be emphasized, as they are currently lost in the plethora of values presented in text. This trimming would make the paper more suitable for a broader higher-impact journal like Nature Communications rather than a methodology-oriented journal.

We have significantly reduced the use of acronyms and numbers to improve the readability of our manuscript. As suggested, we have removed the acronyms A-IV and A-DV and used the terms 'interannual' and 'decadal' variability throughout the manuscript.

Response to Reviewer #3:

My major comments are as follows (in order of my perception of priority):

A) The authors chose to exclude the coastal ocean from their analysis--everything within 3 degrees of the coast and shallower than 1000 m. Given the outsize role of these regions for marine carbon fluxes, I believe the quantitative consequences of that choice should be clarified and visualized for readers.

In the revised version of our manuscript, we have decided to use all available observations from open-ocean and coastal zones. When doing so, we are using 94% of the SOCAT observations. The remaining 6% of SOCAT observations not being used are located in areas of shallow water not simulated by NEMO-PlankTOM12.1. Below, we have tested the robustness of our results to the inclusion, or not, of coastal observations, by building two different open-ocean masks.

A first mask is made out by using the monthly $f\text{CO}_2$ fields from the $f\text{CO}_2$ -products and selecting $1^\circ \times 1^\circ$ grid cells covered by all $f\text{CO}_2$ -products. In fact, some $f\text{CO}_2$ -products only cover 90% of the ocean surface, because they do not resolve the ocean CO_2 sink in continental shelves and high latitude regions. We therefore compare our results using SOCAT observations in the open-ocean areas common to all $f\text{CO}_2$ -products. Using this mask, we use on average 88% of the SOCAT observations.

A second mask is made out by using the definition of the coastal zone from Laruelle et al. (2018). This mask was used in the original submitted version of our manuscript. By using this mask we are using on average only 69% of SOCAT observations.

Results show (Table R2 and Figure R1) that whether or not coastal observations are included, this has a limited impact on the conclusions of our analysis concerning the decadal trend for the period 2000-2022. Global and regional variations are similar with and without coastal observations, with the decadal trend of the global ocean CO_2 sink for the period 2000-2022 constrained with the hybrid approach remaining unchanged ($0.42 \text{ Pg C / yr / decade}$). The only exception is that the global ocean CO_2 sink is lower between 2005 and 2011 in the results of the hybrid approach that used only open ocean observations as defined in Laruelle et al. (2018). However, using this definition of coastal areas limits the number of SOCAT observations used to 69%, and removes observations that are considered as open ocean observations by the $f\text{CO}_2$ -products. Therefore, to provide a better comparison with $f\text{CO}_2$ -products, and simplify the explanation of our hybrid approach to readers, we decided to use all available observations.

However, we note that the ocean CO_2 sink in coastal regions, and its evolution, remains uncertain and add this information in the discussion:

“In addition, the ocean CO_2 sink in the northern region is also more influenced by coastal processes than the southern region, which despite their importance remain uncertain (Dai et al., 2022; Regnier et al., 2022; Laruelle et al., 2018; Mathis et al., 2024; Resplandy, 2024). Consequently, in the northern region, the way in which coastal $f\text{CO}_2$ observations are taken into account by the various $f\text{CO}_2$ -products could induce some of the discrepancies among $f\text{CO}_2$ -products. This would partly explain the lack of coherence between the GOBMs and the $f\text{CO}_2$ -products over this recent decade (Friedlingstein et al., 2023; Gruber et al., 2023; Keppler et al., 2023; Müller et al., 2023).”

Table R2 | Temporal variations of ocean CO₂ sink from hybrid approach results when including or not coastal observations.

	Interannual variability (PgC/yr)	Decadal variability (PgC/yr)	1990-1999 (PgC/yr/decade)	2000-2009 (PgC/yr/decade)	2010-2019 (PgC/yr/decade)	2000-2022 (PgC/yr/decade)	Percentage of SOCAT observations being used (annual average)
Hybrid approach with all available observations	0.22 ± 0.02	0.13 ± 0.02	-0.19 ± 0.17	0.80 ± 0.21	0.44 ± 0.15	0.42 ± 0.06	94%
Hybrid approach with all observations in the area covered by all fCO ₂ -products	0.23	0.13	-0.17	0.80	0.46	0.42	88%
Hybrid approach without coastal zones, as defined by Laruelle et al (2018)	0.24	0.12	-0.14	0.47	0.45	0.42	69%

Figure R1 | The global and regional hybrid approach results when including or not coastal observations. In red, the hybrid approach results with all available observations. In blue, when using the open-ocean area covered by all fCO₂-products. In green, when using the open-ocean area as defined by Laruelle *et al.* (2018).

B. In my opinion, this paper would be more impactful and robust if the authors included the results of sensitivity analyses used to inform their approach. Specifically:

i. Several possible mechanisms are proposed to drive discrepancies between models and data-products. Varying subsampling and model parameters would help identify the most likely of these mechanisms.

Our main focus and results are about the discrepancy between the GOBMs and fCO₂-products for the decadal trend of the global ocean CO₂ sink between 2000 and 2022. Based on our analysis, we suggest an overestimation of the decadal trend in the 2010s by the fCO₂-products. The proposed mechanism to explain this overestimation is a lack of fCO₂

measurements, and some methodological issues in the interpolation and extrapolation of the available data. In order to support this suggestion, we report the evolution of the fCO₂-product estimates within the last three global carbon budgets (see text below and the new figure 6). This analysis is more relevant than performing a subsampling experiment with our hybrid approach because it directly demonstrates that the trend of the 2010s from data-products has been sensitive to revisions in data and methods, while that from GOBMs has varied very little: *“In addition to the extrapolation problems in the under-sampled northern regions mentioned above, on a global scale, the estimate of the positive trend in the 2010s from the fCO₂-product ensemble has been revised downwards in successive publications of the global carbon budget analysis between 2021 and 2023, while their trends for the 1990s and 2000s have remained relatively similar (Fig. 6). For each of the global carbon budget analyses published between 2021 and 2023, the fCO₂-product ensemble average has always been produced from seven estimates. However, two of the seven fCO₂-products were introduced, replacing previously submitted products that were not updated, and five were slightly updated. Among these five fCO₂-products, on average, the 2010 trend between the 2021 and 2023 publications decreased by -0.05 Pg C yr⁻¹ decade⁻¹. Thus it was mainly the turnover in the last two fCO₂-products between 2021 and 2023 that led to a visible decrease in the ensemble average of -0.25 Pg C yr⁻¹ decade⁻¹ for the 2010 trend. Consequently, the downward revision observed for the data-based estimate was mainly due to a change in two fCO₂-product methodologies and, to a lesser extent, to the annual updates of the SOCAT database and fCO₂-product methods, suggesting that the trend of the 2010s estimated with the data-products is not robust at this stage.”*

“Our results suggest that the estimate of the ocean CO₂ sink trend in the 2010s by the fCO₂-product ensemble is overestimated and sensitive to the availability and distribution of fCO₂ observations. Over the last three annual updates of global carbon budgets (Friedlingstein et al., 2022a, b, 2023), although the individual members of the fCO₂-product ensemble underwent very few changes, the 2010s trend estimated from the ensemble of available products has decreased by 14% each year. This supports our finding of an overestimated trend in the 2010s ocean CO₂ sink from the fCO₂-products ensemble, which is adjusted downwards as new data become available. In addition, the replacement of two members of the fCO₂-product ensemble by a hybrid approach along the same lines as presented here (Gloege et al., 2022; Bennington et al., 2022b) and by a revised fCO₂-product aimed at improving the retrieval of the ocean CO₂ sink trend (Zeng et al., 2022) have led to this downward revision of the 2010s trend in the latest ensemble. Consequently, our analysis demonstrates the importance of regular updates and efforts to collect fCO₂ observations as part of SOCAT (Bakker et al., 2016), as well as evaluations of fCO₂-product methodologies (Hauck et al., 2023a; Gloege et al., 2021).”

Fig. 6 | Changes in fCO₂-product estimates over the last three updates of the global carbon budget analysis. The last update of the fCO₂-product estimate was in 2023 (black line and grey shade). The previous two estimates were in 2022 (orange) and 2021 (blue). The left y-axis represents the global CO₂ flux anomaly (for each time series the long-term mean between 1990 and 2020 was removed). GOBM estimates for the same three global carbon budget updates are also shown in dotted lines. The number of annual observations in the SOCAT database, for each annual version of SOCAT (v2021, v2022 and v2023), is shown in the bottom right-hand corner using the same colour code as for the line. Years prior to 2005 are not shown.

We also report a discrepancy between GOBMs and fCO₂-products for the decadal trend of the ocean CO₂ sink in the northern region. To highlight this discrepancy we subsampled the ΔfCO₂ data from fCO₂-products and GOBMs, and compared them to the SOCAT database. This analysis suggested potential methodological issues within four fCO₂-products. We also suggest that the way in which coastal fCO₂ observations are taken into account by the different methodologies could be important. However, no obvious influence of the coastal observations in the North region was detected with our coastal observation sub-sampling experiment carried out for our response to the previous comment (A).

Finally, we report a discrepancy between GOBMs and fCO₂-products in decadal variability of the ocean CO₂ sink. The proposed mechanisms are those cited in the existing literature. Several sub-sampling experiments have been published to specifically investigate the performance of the methods used by fCO₂-products to reproduce the decadal variability of the fCO₂ fields of GOBMs (Gloege et al., 2021; Denvil-Sommer et al., 2021; Hauck et al., 2023a). Therefore, performing such an analysis with our hybrid approach would not provide additional information on this discrepancy in the decadal variability of the ocean CO₂ sink between GOBM and fCO₂-products, which has already been addressed in several published analyses.

ii. Presenting the results of the sensitivity analysis used to select and tune key parameters for optimization and perturbation--within the constraints identified in the literature--would increase confidence in the results.

Regarding the optimisation of NEMO-PlankTOM12.1, please see our detailed response to comment C below. We explain our choice to remove the paragraphs devoted to model optimisation (which was made in a broader context of model development rather than focused on this specific study) and to develop model validation instead which is directly relevant to this paper.

For the selection of the parameters to perturb NEMO-PlankTOM12.1: we have tested parameters related to the bacteria (as in the original submission and used in the main manuscript) and the phytoplankton functional types (see Table R3 and Figure R2 below). Bacteria parameters were selected for our main results because bacteria are ubiquitous in the ocean, and therefore modifications of their parameters affect carbon fluxes in all ocean regions in a consistent and uniform manner. Phytoplankton are already more patchy than bacteria. Tests done on phytoplankton parameters provide an idea of robustness but would not be the preferred parameters to use because the response is not as uniform as when using the bacteria parameters due to the inhomogeneous distribution of phytoplankton biomass. We have also tested two different model configurations using two different atmospheric forcings: NCEP and ERA5. Regardless of the parameter and forcing used, the key results of our analysis remain unchanged about the variations of the decadal trends in ocean CO₂ sink over the period 1990-2022, with the lowest trend observed in 1990s, and a higher trend in the 2000s than in the 2010s. We now detail the results of the sensitivity analysis in our manuscript and supplementary material (supplementary table 3, and supplementary figures 5 and 6):

“However, both GOBMs and fCO₂-products ensembles suggest similar trends between 2000s and 2010s, while the hybrid approach (including the sensitivity analyses) always led to a higher trend in the 2000s compared to the 2010s.”

“Although our hybrid approach always suggests an underestimation of the decadal variability by GOBMs, the exact value is sensitive to the specific model configuration.”

“Finally, we performed sensitivity analyses to test the robustness of our results to the choice of perturbed model parameters and model configurations. The perturbed simulations were repeated with parameters of phytoplankton respiration, and with a combination of both bacterial half-saturation and phytoplankton respiration. The model configuration was changed by using ERA5 reanalysis as weather forcing data. In total, we thus have applied the hybrid approach to six different set ups, with three choices of perturbation parameters and two choices of forcing configurations (Supplementary Figure 5). Regardless of the parameter and configuration used, the results of our analysis remain unchanged about the variations of the decadal trends in ocean CO₂ sink over the period 1990-2022, with the lowest trend observed in the 1990s, and a higher trend in the 2000s than in the 2010s (Supplementary Figure 6 and Supplementary Table 3). We show here results of the model forced with NCEP, which has a lower RMSE (38.5 μatm) compared to the configuration forced with ERA5 (40.0 μatm), and the perturbation of the half-saturation constant of bacterial remineralisation which produces perturbations that are more uniform across the ocean.”

In the manuscript, we have decided to present the results obtained with the NCEP model configuration and by perturbing the bacteria. Because when the model is forced with NCEP, it has a lower RMSE (38.5 μatm) than when forced with ERA5 (40.0 μatm). As mentioned above, the biomass of bacteria varies less in GOBMs (and in reality) than the biomass of phytoplankton. Finally, the range of perturbations of the bacterial parameter that we used (5-18.10⁻⁶ mol/L) is in agreement with the literature, e.g., Mentges et al. (2019, Scientific Report). Note that Mentges *et al.*, (2019) also mentioned this parameter as “*being the least constrained by published referenced values*”. Therefore, our analysis carried out with perturbed bacteria could be more easily reproduced by other modelling teams. It should be noted that the lower 2000-2022 trends (0.27-0.24 PgC/yr/decade) associated with the perturbed phytoplankton members could be explained by a large number of yearly values constrained but outside the bounds of the perturbation experiments (uncertain values) obtained with these members.

Table R3 | Temporal variations of ocean CO₂ sink from hybrid approach results with different perturbed parameters.

	Interannual variability (PgC/yr)	Decadal variability (PgC/yr)	1990-1999 (PgC/yr/decade)	2000-2009 (PgC/yr/decade)	2010-2019 (PgC/yr/decade)	2000-2022 (PgC/yr/decade)
Hybrid approach NCEP bact	0.22 ± 0.02	0.13 ± 0.02	-0.19 ± 0.17	0.80 ± 0.21	0.44 ± 0.15	0.42 ± 0.06
Hybrid approach NCEP phyto	0.28	0.16	0.17	0.97	0.48	0.27
Hybrid approach NCEP phyto + bact	0.18	0.09	-0.37	0.73	0.37	0.41
Hybrid approach ERA bact	0.23	0.25	-0.93	1.27	0.35	0.36
Hybrid approach ERA phyto	0.19	0.18	-0.60	0.87	0.23	0.24
Hybrid approach ERA phyto + bact	0.17	0.14	-0.43	1.14	0.23	0.36

Figure R2 | Hybrid approach results when using different perturbed parameters. The black line with the grey shading, represents the results and error estimate from the main manuscript (using NEMO-PlankTOM12 forced with NCEP, and perturbed bacteria). The other coloured lines represent hybrid approach results obtained with a different forcing (i.e., ERA), and/or when perturbing phytoplankton and/or bacteria. The white dots represent uncertain results as defined in the manuscript.

C. I think the model manipulations described here constitute a sufficiently different approach and parameter set that validation should be presented. e.g., an evaluation of how major variables like temperature and salinity, as well as fCO_2 , differ between this and prior PlankTOM configurations and climatological hydrography.

As mentioned earlier, in this revised version of the manuscript we are using the NEMO-PlankTOM12.1 model that has been submitted to the Global Carbon Budget 2023. As part of the submission to the Global Carbon Budget 2023, the NEMO-PlankTOM12.1 model and the other GOBMs undergo a standard protocol for model validation, which has been greatly expanded in the Global Carbon Budget 2023 and includes, in addition to a comparison with fCO_2 observations from SOCAT, comparisons with observed :

- mean ocean CO_2 sink in the 1990s
- Atlantic Meridional Overturning Circulation
- Southern Ocean sea surface salinity
- Southern Ocean stratification index
- surface ocean Revelle factor

The physical model setup from the NEMO-PlankTOM12.1 model used here is unchanged since. It was validated within the 2023 publication of the Global Carbon Budget. The model version provided here differs from the 2023 submission to the Global Carbon Budget 2023 through its forcing only (NCEP instead of ERA5). We have reconducted the model validation as performed by the Global Carbon Budget analysis (Figure R3).

This version of NEMO-PlankTOM12.1, forced with NCEP, has a better representation of fCO_2 than the version forced with ERA5. We also show that this model is within the range of Global Carbon Budget models, except for the Southern Ocean stratification for which

NEMO-PlankTOM12.1 is slightly lower than other models. This validation of the model is now fully explained and described within the section 5.2: "Description and validation of the NEMO-PlankTOM12.1 model"

D. Various results are compared, but statistical tests are not presented. Including these would help support interpretations of whether or not values are similar or meaningfully different.

The text makes multiple comparisons of the hybrid approach results with the GOBMs and fCO₂-products results, as for example with the Table 1. Such comparisons take into account the uncertainty around the hybrid approach results, as well as the $\pm 1\sigma$ (standard deviation) range associated with the results for GOBMs and fCO₂-products. As a conservative approach, if the hybrid approach value, with its uncertainty, is within the range of the GOBMs or fCO₂-products, we consider that the hybrid approach is not significantly different from the values provided by the GOBMs or fCO₂-products. Moreover, we have added results from sensitivity analysis and mentioned them whether they strengthen or not our results. Otherwise, when statistical comparisons or tests were performed (for example with Figure 4), we reported the associated p-value in the text.

E. I think the nudging/data assimilation approach could be better referenced to illustrate how it draws on or differs from prior literature.

This comment refers to this specific comment:

*"242. My perception is that statistical approaches to 'nudging' earth, ocean, and climate model results to incorporate observational constraints has a long history, which should probably be referenced here. Examples and context include:
<http://dx.doi.org/10.5065/D60Z716B>,
[https://doi.org/10.1175/1520-0493\(2000\)128<3664:ASNTFD>2.0.CO;2](https://doi.org/10.1175/1520-0493(2000)128<3664:ASNTFD>2.0.CO;2),
<https://www.myroms.org/forum/viewtopic.php?t=320>,
<https://doi.org/10.5194/gmd-16-1857-2023>.*

The author's will note that these sources point out that nudging to observations is sometimes considered a suboptimal data assimilation strategy. I recognize that the model optimization used here is more complicated, and may not be well summarized as "nudging".

*Other works have used more data assimilation techniques to incorporate marine carbonate system variables in concert with physical parameters:
<https://doi.org/10.1016/j.jmarsys.2016.02.011>,
<https://agupubs.onlinelibrary.wiley.com/doi/10.1002/2016JC012650>,
<https://agupubs.onlinelibrary.wiley.com/doi/full/10.1029/2009GB003531>,
<https://www.jstor.org/stable/24860162>, <https://www.jstor.org/stable/24861004>.*

Given this context, I think the authors need to not just reference related concepts and works of this nature, but more clearly delineate how this work is novel in approach and focus. I acknowledge that the length-limitations and format of this journal may have obscured significant advances in the approach used by the authors--if so and I

missed these, I take that as further evidence that these need to be clearly and concisely delineated in the text.”

We realised that the principle of our hybrid approach is not well summarised by the word “nudging”. We therefore removed this expression from the text and rephrased our description (see below).

We agree that more estimates from model assimilation techniques have been published and provide more information about the ocean CO₂ sink. For example, the Global Carbon Budget analysis in 2023 used four Earth System Models (ESMs) prediction systems, which are based on assimilating physical atmospheric and oceanic data products into the ESMs. They were used as a new line of evidence in predicting, for 2023, the ocean and land CO₂ sink, and the atmospheric CO₂ growth. In addition, such assimilation methodology can provide information not only on the air-sea CO₂ flux, but also on the sequestration and long-term storage of the CO₂. Although, there are also some limitations related to the way the assimilation of data is performed, as mentioned in DeVries et al (2023) for the ECCO-Darwin model. Nonetheless, we have now added some references about this ongoing model assimilation effort. We also point out to another recent study that also used an hybrid approach between GOBMs and fCO₂ observations, and mentioned the similarities in our results:

“Results from models using data-assimilation (Brasseur et al., 2009; Verdy and Mazloff, 2017), also suggest an underestimated decadal variability in GOBMs air-sea CO₂ flux (Carroll et al., 2020; Li et al., 2023).”

“A similar hybrid approach was recently published (Gloege et al., 2022), but with a machine learning algorithm used to derive the factors influencing the fCO₂ variability. Here, the mechanism as represented in the NEMO-PlankTOM12.1 model remained unchanged and thus also constrained the results.”

“In addition, the replacement of two members of the fCO₂-product ensemble by a hybrid approach along the same lines as presented here (Gloege et al., 2022; Bennington et al., 2022b) and by a revised fCO₂-product aimed at improving the retrieval of the ocean CO₂ sink trend (Zeng et al., 2022) have led to this downward revision of the 2010s trend in the latest ensemble.”

F. The authors appear to use different sources and calculations for water and air fCO₂, instead of the concurrently and self-consistently calculated values from SOCAT. I recommend the authors evaluate whether it is possible to consolidate their data sources in this case.

This comment refers to this specific comment:

“348. SOCAT reports both the wet and dry atmospheric fCO₂ measured concurrently with the seawater fCO₂ selected by the authors, eliminating the need for this calculation and the potential errors in introduces because of discrepancies in choices of equation and virial coefficients used by different groups, differences in the spatial gridding, etc. I strongly recommend the authors revisit their data source and this calculation considering the alternative SOCAT availability--if this is not possible,

please explain why (e.g., are there no comparable interpolated products for the SOCAT air fCO₂?)."

The SOCAT gridded product does not have xCO₂, nor fCO₂air values. When measurements are submitted to SOCAT, it accepts fCO₂air measurements made in parallel to fCO₂water measurements, and archives these fCO₂air measurements for future use. However, SOCAT does not publish these measurements.

However, SOCAT publishes Sea Surface Temperature (SST) and Sea Surface Salinity (SSS) measurements. So, we have done the subsampled side of figure 4 with the SST and SSS from SOCAT, and the obtained results are identical (see Figure R3). However, for consistency with the non-subsampled side of figure 4, we have decided to keep our methodology.

Fig. R3 | Decadal trends in $\Delta f\text{CO}_2$ in the North ($>30^\circ\text{N}$, excluding the Arctic Ocean). Comparison of the measured decadal trend in $\Delta f\text{CO}_2$ from SOCAT (in green) with that of the $f\text{CO}_2$ -product ensemble (blue boxplots) and of the GOBM ensemble (grey boxplots). The $f\text{CO}_2$ -products and GOBMs were sub-sampled at SOCAT locations. The trends are calculated from median annual values for the nord region. On the left, the SST values are from OISST1.2 and the SSS values are from EN4 (as in the manuscript), while on the right SSS and SST values are from SOCAT.

Detailed comments by line number (because this is a long list, I don't expect responses to all of these):

1) Line 53. It would be helpful to further distinguish what is meant by $f\text{CO}_2$ -products here in contrast to observational data only. Are the authors only referring to gridded, interpolated, and climatological datasets that use different methods but all fundamentally rely on SOCAT? Are there additional substantive differences readers should be aware of?

We are referring to the product used within the Global Carbon Budget analysis (as clearly mentioned in the text and in Table 1), and all rely on SOCAT. We have rewritten the sentence:

“Estimates from these observation-based products (fCO₂-products), that all used the SOCAT database as a starting point, confirmed some aspects of the ocean CO₂ sink...”

2) 61. What are the specific fCO₂ products being considered as part of the ensemble? And the specific GOBMs included in the ensemble? I think each product considered should be referenced and included in a table associated with this manuscript. If both ensembles are identical to those used in the Global Carbon Budget 2022 publication, then I think that this point should further be clearly stated in the inline text with an associated reference number.

The ensembles are exactly the same as the ones used in the Global Carbon Budget analysis 2023. We have added this information in the text,

“Despite recent progress, the two types of estimates within the global carbon budget analysis diverge over the last two decades (2000-2022), with the fCO₂-product ensemble suggesting a decadal rate of growth of the ocean CO₂ sink almost twice as high as that simulated by the GOBMs ensemble (Table 1).”

3) 72. a. The result in the 2010s trend column—in which the result is significantly lower than the model ensemble, preferred model, or fCO₂-product ensemble and thus seems to contradict either observational or model evidence—strikes me as a red flag that requires clearer explanation in the text. I'm not sure section 3.4 is sufficient to convince me. Any proposed resolution would be more convincing with a model sensitivity study demonstrating a plausible mechanistic basis.

b. The larger interannual variability in the presented approach than the other models and data products is also concerning, though less so given the reported uncertainties.

c. I suggest a concise table title, rather than expanding this into a caption.

In the revised manuscript, values within Table 1 have been modified. The value of the ocean CO₂ sink trend, between 2000 and 2022, obtained from the hybrid approach now lies between the GOBM and fCO₂-product estimates, while still slightly closer to the GOBMs. We explain these changes (between the submitted and revised manuscript) by modifications in the fCO₂-products results over the last three updates of the global carbon budget analysis (see section 3.3 and figure 6), and also by a modification of the NEMO-PlankTOM12.1 model. The description of this result is done in sections 3.3 and 3.4, and discussed in section 4.

We have performed sensitivity analysis about the interannual variability in section 3.1 to explain the observed discrepancy.

The table title was changed.

4) 87. Given that this approach is so central to the analysis, I think a brief overview here (a couple more sentences), prior to the trailing methods section, would be appropriate.

We have now modified the last sentence of this paragraph that provide more information on the hybrid approach (see response to comment #6)

5) 90. The trends in the preferred model are 20-60% lower than the ensemble in each decade. Is it fair to call these similar trends? Or perhaps it makes sense to restate this as a similar multidecadal trend, since the decadal trends diverge. I also suggest double checking the overall 2000-2019 NEMO trend--based on Fig. S1 it appears that either the overall trend or detrended decadal component trend should be a bit smaller than that of the ensemble (but I may be mistaken).

We have modified this sentence as requested:

“The global annual values of the ocean CO₂ sink estimate of the current model version approximately matches the model ensemble average used within the latest global carbon budget analysis, and with similar variability and multidecadal trend (Table 1, Supplementary Fig. 2)”

We have also double checked the values.

6) 96. A caveat to this comparison to is that these prior studies performed sensitivity analyses across a wide range of parameters and included observational constraints (of parameters other than the objective variable) on the the tuned parameters. If this study does either, it was not obvious to me. So I think it is appropriate to include some comment on how those prior studies guided the authors approach, and any limitations.

We have now added results from sensitivity analysis in the manuscript (see our response to comment B). We have also rewrite the sentence to be more specific about which part of the previous analysis was used in the current study,

“Note that the hybrid approach used here, which estimates an optimised value of a target variable based on several model simulations and a cost function, has been used in previous studies to constrain global ocean primary production and air-sea fluxes of N₂O and CCl₄.”

7) 103. Similar to my comment about line 85, I think the authors should introduce a concise summary of the perturbation and tuning approaches here, beyond stating that they occurred. This context seems important for helping readers understand why state estimates sometimes arise outside the bounds established by the perturbations.

Done:

“This was obtained by perturbing model parameters. Perturbed simulations provided a range of possible values for the ocean CO₂ sink around the estimate from the standard model simulation. Then, for each year, the optimal CO₂ sink was found within this range of possibilities by optimising the calculated Mean Square Error (MSE) between the simulated fCO₂ and the SOCAT observations. The hybrid approach also provides a quantitative estimate of uncertainty (see methods).”

8) 113. This is a matter of personal preference, but now that the authors have defined this variability in terms of amplitude, could the acronyms be shortened to IV and DV? Or, see comment for line 133.

Those acronyms were removed from the manuscript to improve the readability.

9) 113. "Very close" = indistinguishable? Considering these as a one sided test of difference (with table of tests results in supplement?) might clarify the meaning of statements about similarity ('very close', 'close to', 'marginally overlapping', 'in line with', 'within the range of', etc.). For consistency of interpretation by readers, the authors might consider choose a single phrase to indicate not significantly different comparisons.

As mentioned and explained in our response to comment D, we have decided to replace this statement by "*within the range*".

10) 120. I think it would be useful to label the perturbed simulation lines with a descriptive label of how they were perturbed (e.g., microbial $K_{1/2} = \dots$). Perhaps this is semantic, but in the final sentence of the caption I think "constrained but outside the bounds of the perturbation experiments" might be more descriptive than 'uncertain constrained'.

We have added the range of the perturbed parameter in the figure caption, and modified the final sentence as requested:

"Empty red dots are years with an ocean CO₂ sink value constrained but outside the bounds of the perturbation experiments (i.e., uncertain values, see methods). The perturbed simulations are produced by varying the half-saturation constant of bacterial remineralisation (from $5 \cdot 10^{-6} \text{ mol L}^{-1}$ to $18 \cdot 10^{-6} \text{ mol L}^{-1}$)."

11) 131. Wouldn't a moving-average as described here often result in lower variability even without changes in the data distribution? I suggest that smoothing the data in this way is no longer a comparison of interannual variability in the same sense as it was previously introduced. To be very clear, the authors might decide to replace A-DV and A-IV, with a notation that clearly displays the timescale considered. e.g., $V_{\{10y\}}$, $V_{\{3y\}}$, $V_{\{1y\}}$.

Yes, the moving-average lowers the variability. Similarly, applying the hybrid approach over a three year period also lowers the variability. Therefore, comparing the variability from the hybrid approach applied to 3 years, with the model and observation-based variability after a 3-year moving average smooth is reasonable. We decide to not employ the suggested notation to not surcharge the manuscript with acronyms.

12) 154. a. A mismatch may help diagnose mechanisms, but it is not itself a cause of anything. Nor does the presence of a mismatch--absent a clearly aligned expectation grounded in a particular mechanism--conclusively indicate that the southern mid-latitude CO₂ sink is underestimated in models as opposed to overestimated in sparse observations. I think this statement needs to be revised--perhaps the authors meant instead that the overall regional and global disagreements between data sources reflect these latitudes in particular rather than elsewhere.

b. At face value, this finding seems to contradict that of Fay et al. 2021 (cited by the authors as reference 27) with respect to the geographic specificity of model-data mismatches.

We have rewritten this sentence:

“...shows that the mismatch between the model and the fCO₂ observations in the 40-60°S band could be the main cause of the underestimation of the ocean CO₂ sink in the South (Fig. 2). Note that it is the Southern region that has the most influence on the global ocean CO₂ sink.”

Fay *et al.* (2021) pointed to some divergence in the subtropics (north and south), but also in the southern high latitudes.

13) 159. Are the uncertainty bounds smaller than the symbols on most of these bands? Does it make sense that they are this small given the large interannual and interdecadal variability previously discussed?

Yes, most of them are smaller than the symbol. It makes sense as this analysis is based on climatology and not on a specific year.

14) 194. See comment for line 72a.
See our response to comment #3.

15) 219. If relying on significance for argument, I suggest including test result(s) (null, p, test-statistic).

This section has been modified, and see our response to comment D.

16) 208. I think that this statement implicitly assumes a constant gas transfer velocity. I'm not familiar enough with global windspeed or momentum flux trends to know whether that is likely but there are observed and modeled changes in these parameters over the last few decades in at least some regions within this latitude band (e.g., Eastern boundary current regions). The assumptions here probably need some reinforcing citations.

The surface ocean $\Delta f\text{CO}_2$ is the driving force behind the air-sea CO₂ flux. The CO₂ flux is generally estimated using a bulk formula in which $\Delta f\text{CO}_2$ is multiplied by a gas transfer velocity, the solubility of CO₂ in seawater and the ice cover. The gas transfer velocity is frequently parameterized as a function of wind speed.

If the air-sea flux of CO₂ were driven primarily by variations in gas transfer velocity, all fCO₂ products would have similar decadal trends for the air-sea flux of CO₂. Because all fCO₂-products use similar atmospheric forcing datasets for wind speed. For example, see Figure 5 of Fay *et al.* (2021, doi.org/10.5194/essd-13-4693-2021), who introduced a standardised approach for calculating fluxes from fCO₂ products, and still observed large differences in estimated CO₂ fluxes between different fCO₂ products that are induced by the various fCO₂ mapping techniques.

Therefore the assumption made is that $\Delta f\text{CO}_2$ is the driving force behind air-sea CO₂ flux. We modified the sentence:

*“The strong decadal trend in the ocean CO₂ flux estimated by the fCO₂-products is primarily driven by diverging trends between the CO₂ fugacity at the surface of the ocean compared to that in the atmosphere ($\Delta f\text{CO}_2$) (Fay *et al.*, 2021).”*

17) 219. a. Please include trends and relevant statistical tests to reinforce this conclusion.
b. To me, this distinction is a bit arbitrary (or at least reflect a choice of emphasis). Visually, any given decade seems unlikely to diverge greatly from the overall 1990-2020 trend. i.e., if comparing each decade to null hypothesis of no difference from the secular trend, are any of these decadal variations actually significant?

See our response to comment D.

18) 238. a. I think this point about spatial bias in the interpolated products is valuable and it would be good to start a new paragraph here to visually highlight this idea.
b. This point (bifurcation of the interpolation products into high and low clusters) would be strengthened by a supplemental figure.

We split this paragraph in two. We did not add a supplementary figure as we believe the text is clear enough.

20) 258. Please see comment regarding line 156a. Without testing one of the mechanisms proposed, I think that this statement is not fully supported as written.

This result has changed in the revised manuscript

21) 378. This is unclear to me as written. I suggest including an equation and calculated fluxes (and their data sources) in the supplement.

Done:

“Equations of the Global Carbon Budget analysis:

Within the Global Carbon Budget analysis, the carbon sinks (atmosphere = G_{ATM} , ocean = S_{OCEAN} , and land = S_{LAND}) and emissions (fossil fuel = E_{FOS} , and land use change = E_{LUC}) are estimated,

$$(G_{ATM} + S_{OCEAN} + S_{LAND}) = E_{FOS} + E_{LUC} \quad (1)$$

By using the estimates of G_{ATM} , E_{FOS} , and E_{LUC} from the Global Carbon Budget analysis published in 2023, with our estimate of S_{OCEAN} from the hybrid approach, an estimate of the S_{LAND} term can be obtained:

$$S_{LAND} = E_{FOS} + E_{LUC} - (G_{ATM} + S_{OCEAN}) \quad (2)$$

Note that the difference between the terms in equation 1 is equal to the Budget Imbalance (B_{IM}) of the global carbon budget:

$$BIM = E_{FOS} + E_{LUC} - (G_{ATM} + S_{OCEAN} + S_{LAND}) \quad (3)''$$

23) 417. Randomly selected with what distribution? I ask because the authors previously chose to use tests that are suitable for non-normal data. If the data is not well approximated by a normal distribution, that should be noted here and an appropriate distribution chosen for this Monte Carlo approach.

We have use a uniform distribution, and specify it in the text:

“...that had been randomly selected (from a uniform distribution) within...”

24) 444. Some additional details about the treatment of major inorganic carbon components would be appropriate here, even if covered in the reference.

We have added:

“Simulated dissolved inorganic carbon and alkalinity are influenced by air-sea exchanges of CO₂, calcification (production and dissolution), primary production, and remineralisation of organic matter (grazing by zooplankton and remineralisation by bacteria). The alkalinity is also influenced by denitrification.”

25) 398. a. If re-optimized as described, this is a new model variant that really requires independent validation from prior published works. Did this process knock any parameters back out of reasonable ranges? Are basic properties like T & S well described in at least a climatological sense? I acknowledge model validations are a lot of work, but as a reviewer it is difficult for me to have confidence in the results presented here without validation of parameters beyond fCO₂ of this new product presented in the supplement.

b. This optimization scheme is empirical but it isn't clear to me that the optimized parameters are realistically constrained. For example, if I am interpreting the temperature sensitivities correctly (e.g., based on parameter description by Le Quere 2016), that's equivalent to a Q10 values that are lower than typically measured or modeled on either the organismal or ecosystem level (usually 2-3, the values here are equivalent to 1-2). Another example is the use of a constant e-folding scale for light, even though this varies widely in the ocean. Things like this make me question whether appropriate constants are being used and if the model optimization is able to distinguish between compensatory drivers.

c. Also, Table S1 does not include the 47 parameters that the text says were optimized. I think that if other variables were unchanged (I think only changed parameters have been tabulated?), then the text should be updated and the table clear on what is included and why.

Please, see our response to comment C about the validation of the PlankTOM12.1 model being used within our submitted manuscript. We also would like to mention that a paper is in preparation within our research team that will precisely compare the different versions of PlankTOM12 that our group developed over the past few years. This evaluation is heavy and out of the scope of this manuscript.

Our objective for this manuscript was to use the PlankTOM12 version used within the latest Global Carbon Budget analysis, and that the model performances are within the range of the other GOBMs being used within the global carbon budget. This is now clearly explained in

section 5.2. Therefore, in this section we have removed the optimization paragraphs and elaborate on the model validation instead.

26) 403. Please elaborate on why these parameters were chosen, at least in a supplemental text. That makes these choices more transparent, and helps other modeling teams understand which differences are important.

410. I would prefer representative results to back up this claim were included.

422. But Table S1 appears to show changed physical parameters as well...please resolve this (appearance of) inconsistency.

In the revised manuscript, and in this section, we have removed the optimization paragraphs and elaborate on the model validation instead, for the reasons previously mentioned.

29) 425. Is this different from the A-IV discussed earlier? If so please consider removing the new acronym, as it is only used a couple times in the methods.

Yes, it was different. However, we have modified this section and now used the amplitude of the interannual and decadal variability as described earlier.

30) 483. What is the order of magnitude of observations required to do this robustly?

There is no rule for knowing exactly how many observations per year are needed for the hybrid approach to be robust. However, the number of observations before 1980 was extremely low (below the 25th percentile = 499 observations), which is why in the revised manuscript we decided to use the hybrid approach from 1980 onwards.

31) 460. a. This is a choice that I think should be clearly communicated in the main text, and as a mask or outline in the maps in Fig. 5 and Fig. S3. It is unclear from this whether the model outputs and data products are similarly masked out in the same coastal band, so please clarify that these analyses are comparing like to like.

b. Areas of <1000 m depth and within 3 degrees of shore are disproportionately important for marine carbon cycling. Ignoring them in order to characterize more pelagic regions may make it easier to identify trends and present more constrained appearing values, but at the cost of considering real, environmentally relevant variability and significant contributions to the global carbon flux. Reading this, I have diminished confidence in claims about the changed magnitude of the climatological flux (and inferred land flux), trends (which may be compensated or exacerbated by coastal processes), and the relative agreement of variability estimates. I think that this is too important a caveat to be buried in the Methods and not discussed in the Discussion.

c. How do the results change if coastal areas are considered as well, or separately? I would like to see even a quick discussion of this and followup plots in the supplement. It may be that focusing on the pelagic ocean is defensible and interesting, but this would be a more compelling manuscript if readers were able to consider the coasts as well--and if the authors were to point out some potential open questions that arise from those considerations.

Please see our response to comment A. We have also added the importance and uncertainty of ocean CO₂ fluxes associated with coastal areas to the discussion, and how this might explain some of the discrepancies observed in our study.

32) 494. Is this the most relevant variable to change?

Could the authors present sensitivity analysis supporting the selection of this parameter instead of/without others?

What are the actual values of the perturbations used? This doesn't seem to be reported anywhere, but needs to be.

Is it reasonable to assume a constant value (as shown in Table S1) or are biogeographic variations in this (and other constants) expected?

What are the observational constraints on this parameter and how were the magnitudes of changes selected within those constraints?

I think this choice needs significantly more explanation. There many fit results (in time and space) plotted outside the range of these perturbed models, which leads me to ask if the perturbations really are providing useful bounds or not--and if they are, whether the hybrid approach would benefit from additional constraints or flexibility in parameters. I understand that this is a poorly constrained parameter, but there does seem to be a finite range that is consistent with measured ocean DOC values, which could help justify bounds on the perturbations.

Please see our response to comment B.

33) 498. Please provide a representative graphical example of this approach in the supplement (or a couple, a good fit and a bad fit).

Done

34) 505. Related to comment on line 467, I'd like to see some more quantitative assessment of the statement here.

Line 505, we mentioned years that have constrained values outside the range of perturbation (as understood by the reviewer, see comment #10): *"Years with a constrained ocean CO₂ sink not within the range of the ocean CO₂ sink from the perturbed simulations are kept but considered as uncertain values"*

We believe it is a fair approach, and we do not understand how quantitative assessment from this sentence could be provided.

35) 514. There is no Fig. S4 in the supplement provided for review, but this would be welcome and would address my request for line 470 as well.

Done

36) 528. What quadratic function? Is there a mathematical justification for a different calculation? i.e., is this an arbitrarily larger confidence bound?

We mentioned before that we used a cubic function, while here we decided to use a quadratic function. The justification is that the quadratic function provides a larger confidence interval, which is a reasonable approach considering the larger uncertainty related to a regional analysis.

37) Fig. S1. The letters above the last two panels are repeated.

Corrected

38) Fig. S3. I suggest making these panels larger.

Done

39) In the data link, the hybrid results have a mislabeled column 1 in the tab for results by latitude.

Corrected

40) The code looks plausible (I did not attempt to run it), and additional annotation would be appreciated to help users looking to translate this to an open source programming language for testing.

We have followed the “data and code availability” policy of the journal.

REVIEWER COMMENTS

Reviewer #1 (Remarks to the Author):

Authors Nicolas Mayot, Erik T. Buitenhuis, Rebecca M. Wright, Judith Hauck, Dorothee C. E. Bakker, and Corinne Le Quéré

Title Constraining the trend in the ocean CO₂ sink during 2000-2022

Manuscript NCOMMS-23-48601A (revised)

Key results:

Mayot et al. present a hybrid method fusing processed-based models and observations to constrain the trend in oceanic CO₂ uptake for the period 2000-2022 to 0.42 ± 0.06 Pg C yr⁻¹ decade⁻¹. Critically, this revised trend in the ocean CO₂ sink enables them to infer the terrestrial CO₂ sink to 0.12 ± 0.07 Pg C yr⁻¹ decade⁻¹ over the same 2000-2022 period.

The hybrid model forces the state-of-the-art NEMO-PlankTOM12.1 GOBM with NCEP reanalysis data and observed atmospheric CO₂ concentrations. It provides an annual estimate of the ocean CO₂ sink by constraining the NEMO-PlankTOM12.1 yearly outcome with SOCAT observations. Using this approach, Mayot et al. argue convincingly that “methodological issues in some fCO₂-products could lead to an unrealistic amplification of the ocean CO₂ sink trend in the 2010s”.

Validity:

The revised manuscript addresses the comments raised by the reviewers in detail. The result is a much clearer presentation of the authors’ findings.

Significance:

The results of this study will have a major impact on the understanding and quantification of the global ocean CO₂ sink and its constraints on quantifying the terrestrial CO₂ sink. It should be published.

Data and Methodology/Analytical Approach:

The approach and data use appear reasonable. Previous reviews have in principle vetted the hybrid approach used here.

Suggested improvements

Improvements are suggested below in the detailed line by line comments.

Detailed Comments:

Line 9 “The hybrid approach reproduces the stagnation of the ocean CO₂ sink in the 1990s...”

This statement implies that the analysis begins prior to the 1990s since stagnation must be judged against some previous trend

Lines 14-15 The ocean sink is increasing almost 4x faster than the land sink over the period 2000-2022. This disparity is not discussed further.

Table 1 Should column 4 under decadal trends be relabeled “2000-2022”?

Suggest adding a vertical line separating the Amplitude and Decadal Trends sections of Table 1 for greater clarity

Line 74 “Here, we investigate the discrepancies between fCO₂-products and GOBMs,”

This is a misstatement as only the NEMO-Plank TOM12.1 GOBM is investigated in the hybrid method – the full GOBM ensemble is not evaluated.

Line 95 Suggest deleting “as represented in the model”

Line 96-97 Consider deleting “which estimates an optimised value of a target variable based on several model simulations and a cost function”

Lines 101-109 How much did the perturbed simulations over- or underestimate the ocean CO₂ sink? Why was this range sufficient to capture the full range of interannual variability? Which parameters were perturbed? Could similar ocean CO₂ sink over- or underestimates be generated by perturbing different parameters or parameter combinations? More details are needed. Suggest a pointer to the supplemental information

Lines 146-47 “...North and South regions.”

Fig 2, Lines 152-157 What does this mismatch between data, models, and the hybrid results suggest about our current understanding of the Southern Ocean sink? The hybrid results track the most perturbed simulations

Figures 1,3, Table 1 Prior to 2000, the hybrid solution in Fig 3a shows large interannual variability to the point of questioning whether this is noise due to insufficient numbers of measurements.

Looking back, Fig 1a shows similar behavior

Suggest adding a row to Table 1 giving the number of measurements contributing to the fCO₂-product ensemble during each period

Lines 191-194 “The distinct decadal trend variations between the 2000s and the 2010s suggested by the hybrid approach is robust to different configurations of the original model and to the choice of perturbed parameters (see methods).”

See comment for Lines 101-109 above

Fig 5, Lines 240-246, Lines 307-314 The largest NH CO₂ flux trend anomalies during the 2010s occurs in the North Pacific/Bering Sea (Fig 5a), an area with exceptionally low measurement density (Fig 5b). This is mentioned in Lines 240-246, but the explanation is unsatisfactory especially given the divergence in the fCO₂-products. However, the assessment in lines 307-314 is compelling

Reviewer #3 (Remarks to the Author):

I think this is a thorough revision that makes defensible choices and useful clarifications. I look forward to seeing this work in print, so that I can share it colleagues who will find it interesting and timely.

A final minor comment: On lines 550 and 551 the word perturbation appears twice--I think this half of the sentence could be modified to be more clear (e.g., I'm assuming that the second use is referring to changes in fCO₂ resulting from the K_{1/2} perturbation, but this could be referring to something else).

Reviewer #3 (Remarks on code availability):

I reviewed the code in moderate detail at first submission. While some input details may have changed since, a cursory inspection suggests the substance is similar. It is reproducible and usable in the scripting language submitted (Matlab) based on partial testing (with dummy testing data for speed). I did not confirm that it would work in the open source alternative, Octave.

We sincerely thank the reviewers for their positive appreciation of our revised manuscript and responses to their earlier comments. Our current revision takes into account all the improvements suggested by the three reviewers, and addresses the major concerns raised by reviewer 2. In particular, we clarified the uncertainty throughout the manuscript, and increased the focus on the most robust results. We would like to thank the reviewers again for their valuable comments on our manuscript.

See below our point-by-point response to each reviewer, with the reviewer's comments in blue, the new text in red and our response in black.

Response to Reviewer #1:

Key results:

Mayot et al. present a hybrid method fusing processed-based models and observations to constrain the trend in oceanic CO₂ uptake for the period 2000-2022 to 0.42 ± 0.06 Pg C yr⁻¹ decade⁻¹. Critically, this revised trend in the ocean CO₂ sink enables them to infer the terrestrial CO₂ sink to 0.12 ± 0.07 Pg C yr⁻¹ decade⁻¹ over the same 2000-2022 period. The hybrid model forces the state-of-the-art NEMO-PlankTOM12.1 GOBM with NCEP reanalysis data and observed atmospheric CO₂ concentrations. It provides an annual estimate of the ocean CO₂ sink by constraining the NEMO-PlankTOM12.1 yearly outcome with SOCAT observations. Using this approach, Mayot et al. argue convincingly that “methodological issues in some fCO₂-products could lead to an unrealistic amplification of the ocean CO₂ sink trend in the 2010s”.

Validity:

The revised manuscript addresses the comments raised by the reviewers in detail. The result is a much clearer presentation of the authors' findings.

Significance:

The results of this study will have a major impact on the understanding and quantification of the global ocean CO₂ sink and its constraints on quantifying the terrestrial CO₂ sink. It should be published.

Data and Methodology/Analytical Approach:

The approach and data use appear reasonable. Previous reviews have in principle vetted the hybrid approach used here.

Suggested improvements

Improvements are suggested below in the detailed line by line comments.

1) Line 9 “The hybrid approach reproduces the stagnation of the ocean CO₂ sink in the 1990s...”. This statement implies that the analysis begins prior to the 1990s since stagnation must be judged against some previous trend

The term “stagnation of the ocean CO₂ sink” is the term generally used in the literature to describe the low trend of the ocean CO₂ sink estimated in the 1990s, as opposed to its strengthening since the early 2000s (Le Quéré et al., 2007; Landschützer et al., 2015; DeVries et al., 2017; Hauck et al., 2020; McKinley et al., 2020; Gruber et al., 2023; Friedlingstein et al., 2023), which are discussed in section 4. Since atmospheric CO₂ increased during the 1990s, the ocean CO₂ sink should also have increased, but it did not. We can therefore say that the term stagnation is appropriate, regardless of variations over previous or subsequent decades (as is the case here and in other publications). It should be noted also that a low trend in the 1990s can be described in comparison not only with previous decades, but also with subsequent decades (2000s, 2010s), or in comparison with longer-term trends (e.g., 2000-2022).

2) Lines 14-15 The ocean sink is increasing almost 4x faster than the land sink over the period 2000-2022. This disparity is not discussed further.

To respond to this comment and considering the requests to clarify the uncertainty raised by Reviewer 2, we have focused the final paragraph of our discussion on the implications of our results for the total land CO₂ sink (which includes natural fluxes and emissions from land-use change). We discuss the total land sink instead of the natural land fluxes as in our original submission because the constraint that ocean data provides on their sum is more robust than the constraints on the individual land fluxes. To clarify the text, we also removed the sentence on the improvement in the global carbon budget imbalance (which was only slightly improved by 0.08 PgC/yr by our analysis) to focus only on the land fluxes. This change allows us to broaden the discussion on potential implications of our results, and to raise possible issues not only for natural fluxes estimated by DGVMs, but also for land-use change estimates using bookkeeping methods in the Global Carbon Budget analysis. We did not offer a reason to account for these shortcomings on land because this is not our area of expertise, but we expect that the conclusion of our study will be of relevance to the land carbon cycle community, and fuel discussions particularly within the framework of the annual update of the Global Carbon Budget analysis.

Line 14: “*The hybrid approach constrains the 2000-2022 trend in the ocean CO₂ sink to 0.42 ± 0.06 Pg C yr⁻¹ decade⁻¹, and by inference the **total land CO₂ sink to 0.28 ± 0.13 Pg C yr⁻¹ decade⁻¹.***”

Line 394: “***Within the limits of the hybrid approach, a trend of 0.28 ± 0.13 Pg C yr⁻¹ decade⁻¹ in the total land CO₂ sink (including natural fluxes and emissions from land-use changes) can be inferred based on our estimate of the trend in the ocean CO₂ sink for 2000-2022, corresponding to a growth of 0.6 Pg C yr⁻¹ over those 23 years. This result was obtained by adding to and subtracting from our estimate of the ocean CO₂ sink, global carbon budget estimates for the growth rate of atmospheric CO₂, and CO₂ emissions from fossil fuels (taking into account cement carbonation, detailed in the Supplementary Information document)(Friedlingstein et al., 2023). Our estimated trend in the total land CO₂ sink lies between the 0.43 ± 0.20 Pg C yr⁻¹ decade⁻¹ trend estimated by the global carbon budget analysis(Friedlingstein et al., 2023) and the trend of 0.07 ± 0.14 Pg C yr⁻¹ decade⁻¹ that would be obtained with the ocean CO₂ sink estimate from the fCO₂-products alone. Therefore, the land trend inferred from fCO₂ observations suggest either an overestimation of the increasing trend in the simulated land CO₂ sink by Dynamic Global Vegetation Models***

(DGVMs), an overestimation of the decreasing trend in CO₂ emissions from land-use changes by bookkeeping approaches used in the global carbon budget analysis (which might be increasing), or both (Friedlingstein et al., 2023). The latter includes the possibility that emissions from land-use changes were stable or even increased during 2000-2022, which is plausible given the large uncertainty in land-cover change data and in management processes (Friedlingstein et al. 2023). Our results demonstrate that ocean observations can constrain trends in land CO₂ fluxes, and that results are limited by the availability of fCO₂ observations.”

Note that the difference in trend between the 0.28 Pg C yr⁻¹ decade⁻¹ presented in this revised version, and the 0.12 Pg C yr⁻¹ decade⁻¹ presented in the previous paper version is entirely due to the expanded scope of the land flux presented (from natural land fluxes to total land fluxes), the results have not changed. It should be noted that the uncertainty provided (i.e., ± 0.13 Pg C yr⁻¹ decade⁻¹) includes uncertainties associated with fossil fuel CO₂ emissions, the atmospheric CO₂ growth rate and the ocean CO₂ sink estimated from the hybrid approach.

3) Table 1 - Should column 4 under decadal trends be relabeled “2000-2022”?

Yes, indeed. Corrected.

4) Table 1 - Suggest adding a vertical line separating the Amplitude and Decadal Trends sections of Table 1 for greater clarity

Done

5) Line 74 “Here, we investigate the discrepancies between fCO₂-products and GOBMs,”. This is a misstatement as only the NEMO-Plank TOM12.1 GOBM is investigated in the hybrid method – the full GOBM ensemble is not evaluated.

The discrepancies between fCO₂-products and GOBMs is the specific context of our study. We investigate this discrepancy by exploiting a hybrid approach based on one of the GOBMs. Therefore, we rewrote the beginning of this paragraph as (line 73): “We *focus here on understanding the factor-of-two inconsistency in estimates of the 2000-2022 trend in the ocean CO₂ sink between fCO₂-products and GOBMs, which occurs despite the increasing number of fCO₂ observations (i.e., from around 4,500 gridded observational data points a year in the 1990s, to 10,000 in the 2000s, and 15,000 in the 2010s). We introduce and use a new hybrid approach that uses, as a starting point, the NEMO-Plank TOM12.1 GOBM...*”

6) Line 95 Suggest deleting “as represented in the model”

Done

7) Line 96-97 Consider deleting “which estimates an optimised value of a target variable based on several model simulations and a cost function”

This additional information was added in response to a comment from another reviewer, and we therefore consider it important to retain it. We have shortened it to, line 98: “*Note that the hybrid approach used here, which optimises a target variable using multiple model simulations and a cost function, has been used in previous studies*”

8) Lines 101-109, and Lines 191-194 “The distinct decadal trend variations between the 2000s and the 2010s suggested by the hybrid approach is robust to different configurations of the original model and to the choice of perturbed parameters (see methods).”

- How much did the perturbed simulations over- or underestimate the ocean CO₂ sink? Why was this range sufficient to capture the full range of interannual variability?

Over the 2000-2022 period, on average, the range of the perturbed simulations goes from 1.29 Pg C yr⁻¹ to 3.22 Pg C yr⁻¹. This fully covers the expected range of the global ocean CO₂ sink estimate published by the Global Carbon Budget analysis. We have added this information in Section 5.3 which provides the technical details on the method (line 517): “*These perturbed simulations range from 1.3 Pg C yr⁻¹ to 3.2 Pg C yr⁻¹ on average during 2000-2022, and span the expected range suggested by the global carbon budget analysis (i.e., 2.6 ± 0.4 Pg C yr⁻¹ on average, with individual years ranging from 1.8 Pg C yr⁻¹ to 3.0 Pg C yr⁻¹)*”.

- Which parameters were perturbed?

In the manuscript, we show results obtained with the model forced with NCEP and the perturbation of the half-saturation constant of bacterial remineralisation. This is mentioned and justified in the method section (line 566): “*We show here results of the model forced with NCEP, which has a lower RMSE (38.5 μatm) compared to the configuration forced with ERA5 (40.0 μatm), and the perturbation of the half-saturation constant of bacterial remineralisation which produces perturbations that are more uniform across the ocean*”. We also mentioned the parameter perturbed in the caption of Figure 1: “*The perturbed simulations are produced by varying the half-saturation constant of bacterial remineralisation (from 5.10⁻⁶ mol L⁻¹ to 18.10⁻⁶ mol L⁻¹)*”.

- Could similar ocean CO₂ sink over- or underestimates be generated by perturbing different parameters or parameter combinations?

We also did five sensitivity analyses (line 558): “*The perturbed simulations were repeated with parameters of phytoplankton respiration, and with a combination of both bacterial half-saturation and phytoplankton respiration. The model configuration was changed by using ERA5 reanalysis as weather forcing data*”.

The ranges of ocean CO₂ sink produced with these perturbed simulations from our five sensitivity analyses span the expected range of ocean CO₂ sinks except for two analyses (i.e., NCEP and phytoplankton respiration, and ERA5 and phytoplankton respiration) for which the maximal ocean CO₂ sink could not be raised further and were, on average over the 2000-2022 period, 2.68 Pg C yr⁻¹ and 2.91 Pg C yr⁻¹ instead of the expected 2.96 Pg C yr⁻¹. This limitation relates to the inhomogeneous distribution of phytoplankton, compared to bacteria (used for our main analysis). This explained the higher number of uncertain

constrained values of ocean CO₂ sink obtained with these two sensitive analyses, as visible and mentioned in the caption of the Supplementary Figure S6. It is also the reason why we chose to perturb bacteria parameters in our main analysis.

- More details are needed. Suggest a pointer to the supplemental information

As suggested by the reviewer, we have added more details about our methodology in the results section of the manuscript and a pointer to the supplementary information (line 107): *“This was obtained by perturbing model parameters. Perturbed simulations provided a range of possible values for the ocean CO₂ sink around the estimate from the standard model simulation, and covered the expected range suggested by the global carbon budget analysis. We show here results obtained with the perturbation of the half-saturation constant of bacterial remineralisation, which is more homogenous and therefore more robust (see methods and the supplementary information for details of the sensitivity analyses)”*.

9) Lines 146-47 “...North and South regions.”

Corrected

10) Fig 2, Lines 152-157 What does this mismatch between data, models, and the hybrid results suggest about our current understanding of the Southern Ocean sink? The hybrid results track the most perturbed simulations

These results are discussed in the discussion section (section 4), and we didn't want to discuss them directly in the results section. The mismatch is specifically mentioned in the discussion section, line 343: *“In the Southern Ocean, our hybrid approach suggests that existing fCO₂ measurements could corroborate a strong and positive decadal trend in this region in the 2010s, and more generally between 2000-2022.”*

We argued for the collection of a larger number of observations (line 345): *“However, the paucity of fCO₂ measurements impede our ability to evaluate the decadal trend using observations only (Gloege et al., 2021), but recent studies showed that undersampling could be responsible for strong biases in fCO₂-products in that region (Hauck et al., 2023a)”*

To improve the discussion (line 363) a reference was added, and we refer readers to a recent in-depth assessment of the ability of GOBMs to simulate the Southern Ocean CO₂ sink: *“Finally for the mean ocean CO₂ sink, the hybrid approach returns a higher mean CO₂ sink than NEMO-PlankTOM12.1 in the Southern Ocean because it corrects a consistent bias of overestimation of the surface ocean fCO₂ (Supplementary Fig. 2). Studies based on emergent constraint properties (Bourgeois et al., 2022; Terhaar et al., 2021, 2022), and thorough assessments of the ability of GOBMs to simulate the Southern Ocean CO₂ sink (Hauck et al., 2023b), have also suggested that the current generation of models underestimates the global ocean CO₂ sink due to a deficient representation of ocean circulation in the Southern Ocean.”*

11) Figures 1,3, Table 1 Prior to 2000, the hybrid solution in Fig 3a shows large interannual variability to the point of questioning whether this is noise due to insufficient numbers of measurements. Looking back, Fig 1a shows similar behavior

Figure R4. The number of annual fCO₂ observations in the SOCAT database (v2023)

At global scale, the number of fCO₂ observations in the SOCAT database was relatively small prior to 1990 (see Figure R4). Consequently, some of our constrained values of the global ocean CO₂ sink prior to 1990, and the resulting interannual variability, may be uncertain. This is why we study the trend in the ocean CO₂ sink only from 1990 onwards using the hybrid approach. It should also be noted that in the Global Carbon Budget analysis (Friedlingstein et al., 2023), estimates of the ocean CO₂ sink from fCO₂-products are used from 1990 onwards for the same reason:

- In the Supplements of the Global Carbon Budget 2023 (Friedlingstein et al., 2023): *“We also use eight estimates of the ocean CO₂ sink and its variability based on surface ocean fCO₂ maps obtained by the interpolation of surface ocean fCO₂ measurements from 1990 onwards due to severe restriction in data availability prior to 1990”*

Nevertheless, we provide estimates of the hybrid approach between 1980 and 1990, because on a global scale, the number of observations between 1986 and 1989 is similar to that in 1990 (around 1100 gridded observational data points), and in the North region, the number of observations is similar between the early 1980s and the early 1990s. In this revised version, we also further discuss the data limitations in the 1990s in response to comments from Reviewer 2.

12) Suggest adding a row to Table 1 giving the number of measurements contributing to the fCO₂-product ensemble during each period

Note that the number of fCO₂ observations used by the fCO₂-products could vary from one product to another. However, we have added, in the text, information on the number of observations available at global scale within the SOCAT database, line 74: *“... which occurs despite the increasing number of fCO₂ observations (i.e., from around 4,500 gridded observational data points a year in the 1990s, to 10,000 in the 2000s, and 15,000 in the 2010s).”*

13) Fig 5, Lines 240-246, Lines 307-314 The largest NH CO₂ flux trend anomalies during the 2010s occurs in the North Pacific/Bering Sea (Fig 5a), an area with exceptionally low measurement density (Fig 5b). This is mentioned in Lines 240-246, but the explanation is unsatisfactory especially given the divergence in the fCO₂-products. However, the assessment in lines 307-314 is compelling

The reason for the divergence between fCO₂-products within this region remains unclear and could be associated with multiple factors. This would require an in-depth evaluation of the fCO₂-products methods, as well as of the regional trends within the auxiliary datasets that are used as inputs (which are different among fCO₂-products), which is beyond the scope of this manuscript. However, as the reviewer agrees with us, we suggest (lines 335-342) a methodological problem and argue for a cautious treatment of coastal observations in the regions.

Response to Reviewer #2:

I appreciate the amount of work that the authors put into this study. However, I still find a number of discrepancies in the results as presented, which make me question the methodology and the uncertainties associated with the method and the reported values. The detailed major concerns are summarized below (in no particular order, mostly in order of appearance in text):

Thank you for your review which has helped improve our manuscript and clarify our results. We have provided additional explanations here and in the manuscript to remove any remaining questions on the methodology used, to clarify the reported uncertainty, and to increase the focus on the most robust results. We hope these respond to all of your points.

1) Table 1 compared with Table R1: I find it worrisome that including 2019 as an additional year changes the decadal trend values so much for NEMO and the hybrid approach (in this case by more than the standard deviation.)

There was an error in the heading of the fourth column of table 1, which should have read 2000-2022 instead of 2000-2019 (as also mentioned by reviewer 1, comment #3). This is now corrected and we are sorry about the mistake and confusion that it has caused. Therefore, the numbers in Table 1 included four additional years, not just one, and it is a real signal that the trend is decreasing when the four additional years are included, as can be seen in all types of estimates.

See below (Table R4), including 2019 as an additional year does not change much the decadal trend values for the fCO₂-products, GOBMs, NEMO-PlankTOM12.1 and the hybrid approach.

Table R4 | Decadal trends of the ocean CO₂ sink between 2000-2018, 2000-2019, and 2000-2022, estimated by different methods.

	2000-2018 (PgC/yr/decade)	2000-2019 (PgC/yr/decade)	2000-2022 (PgC/yr/decade)
fCO ₂ -products	0.61 ± 0.18	0.62 ± 0.16	0.54 ± 0.13
GOBMs	0.37 ± 0.05	0.35 ± 0.05	0.28 ± 0.05
NEMO-PlankTOM12.1	0.46	0.42	0.33
Hybrid approach	0.51 ± 0.07	0.51 ± 0.06	0.42 ± 0.06

2) L83-85: I do not agree with this statement because the NEMO decadal averages do not fall even within one standard deviation from the other GOBM ensemble average for 2000s and 2010s. There needs to be more care taken to explain why this particular model was used. I would argue that it being used the longest in the annual updates is not necessarily a virtue if better models have since become available.

We have rewritten L83-85 to clarify why we chose the NEMO-PlankTOM12.1 and why this model is appropriate for the study presented here. We also added references to the Methods section 5.2 and to the Supplementary Information which include further details.

Essentially, the NEMO-PlankTOM12.1 model is an established GOBM that was initially designed for quantification of the variability in CO₂ flux (first published in Le Quéré et al., GBC 2000; with major updates published in Le Quéré et al., 2007 - Science, Le Quéré et al., 2010 - GBC, Le Quéré et al., 2016 - Biogeosciences, and Wright et al., 2021 - Biogeosciences; and minor updates done regularly and documented in all 18 updates of the Global Carbon Budget analysis). The NEMO tripolar grid has an enhanced resolution in the tropics and at high latitudes which helps improve representation of variability in those regions. NEMO-PlankTOM12.1 benefits from a long experience in the characterization and evaluation of its CO₂ variability, including through the use of oceanic and atmospheric CO₂ and also O₂ data to constrain model developments and simulation choices. As a result, the NEMO-PlankTOM12.1 model generally produces slightly more decadal variability than similar models, which is consistent with independent observational constraints (see Fig. 3 from DeVries et al., 2019). This can partly explain why NEMO-PlankTOM12.1 sits outside the one standard deviation from the GOBM ensemble for decadal trend, as pointed out by the reviewer.

We have broadened the sentence, on line 82-86, to clarify that the overall performance of the NEMO-PlankTOM12.1 is comparable to that of other similar models based on the metrics used to assess GOBMs in the Global Carbon Budget, and included further detail on the model (see answer to comment 3 below). We acknowledge, as the reviewer pointed out, that being used the longest by itself is not necessarily a demonstration of quality. We have clarified the meaning in the paper to stress that we are using an established model designed for the study of the ocean CO₂ sink variability.

The paragraph now reads, line 82-86: "*NEMO-PlankTOM12.1 is the latest update of an established GOBM that was used from the onset in the annual updates of the global carbon budget analysis and which was designed for the study of the ocean CO₂ sink variability (Le Quéré et al. 2010). Its overall performance in simulating ocean physics and biogeochemistry*

is comparable to that of other GOBMs in the global carbon budget analysis (see section 5.2, Table 1, Supplementary Fig. 2 and 3; Friedlingstein et al., 2023, DeVries et al., 2019)”.

This paragraph refers to several related sections of the paper which show that NEMO-PlankTOM12.1 is comparable to GOBMs used in the Global Carbon Budget:

- Section 5.2 presents a description of NEMO-PlankTOM12.1 and how it was evaluated;
- Table 1 compares results with those of the Global Carbon Budget;
- Supplementary Fig. S2 shows the spatial mean bias of NEMO-PlankTOM12.1 and the GOBM mean with SOCAT observations;
- Supplementary Fig. S3 shows the four metrics used to assess GOBMs in the Global Carbon Budget annual update, comparing NEMO-PlankTOM12.1 with GOBMs used in the 2023 update.

Finally, we note here that the hybrid approach is designed to correct potential model biases. Thus, the original model used (NEMO-PlankTOM12.1) does not need to be perfect or the best model available, but it should be close enough to reality for the optimization to converge for most years, which is the case.

3) L91: A brief discussion of what is included and not included in the GOBM modeling of NEMO would be useful here. What biogeochemical mechanisms are omitted? From its resolution, what physical mechanisms (submesoscale processes?) are omitted?

We have included here a mention of the key physical and biogeochemical processes represented, but given this is the introduction and is meant to be short and accessible to the broad audience of *Nature Communications*, we refer the reader to the detailed description of NEMO-PlankTOM12.1 that is included in section 5.2. The sentence in the introduction now reads (line 91): *“Here, the mechanism as represented in the NEMO-PlankTOM12.1 model, including the mixed-layer dynamics and the large-scale circulation, the carbonate chemistry, and the organic carbon transfer to depth resulting from biological processes (see section 5.2 for a description of the model) remained unchanged and thus also constrained the results.”*

Although the physical ocean model used, i.e. NEMO, has a spatial resolution of up to 2° and does not resolve sub-mesoscale processes, it includes a parameterisation for subgrid-scale eddy-induced mixing as with all models used in the Global Carbon Budget to our knowledge. Furthermore, the latitudinal resolution is enhanced in the tropics and at high latitudes, enabling better representation of dynamical processes in those regions. More importantly though, our hybrid approach can correct for potential model biases through its approach of optimising towards observations. We expanded the model description in Section 5.2, line 447: *“NEMO-PlankTOM12.1 used the NEMO model in its global configuration on the ORCA tripolar grid, with a longitudinal resolution of 2° and an average latitudinal resolution of 1.5°, the latter being enhanced up to 0.3° in the tropics and at high latitudes, and a temporal resolution of 96 minutes. [...] Subgrid-scale eddy-induced mixing is represented with a parameterisation(Gent and McWilliams, 1990).”*

4) L115-117: There should be some explanation why this is happening. Sure, the authors provide a "robustness check" using a three-year period, but it still does not explain why the hybrid approach produces such high interannual variability, higher (more than double and

outside of one standard deviation range) than both the model and observational values. Instead, the model and observational values are actually mostly in agreement. This result actually makes me question the validity of the hybrid approach as applied in this study.

We had not discussed the results on interannual variability any further because they are not the main focus of the paper (which is on decadal variability and trends). Rather than expanding the text on interannual variability in this revised version, we have refocused the results section 3.1 on the presentation of the decadal variability and its robustness (trends are discussed in section 3.3), and refocused the method section 5.1 on the estimations of the amplitude of the decadal variability and decadal trends. We moved the presentation and discussion of the interannual variability results to the supplementary material, also acknowledging that they are less robust than the decadal variability results for reasons explained below.

The high interannual variability in the hybrid approach comes to a large extent from the first half of the 1990s where there are fewer observations (see Supplementary Fig. 1b, number of available observations in the 1990s are now added to Fig. 6). This high interannual variability in the 1990s is generated when few observations exist because the hybrid approach constrains results using only observations for a given year (without using observations from the following or previous year). Excluding the 1990s, the interannual variability produced by the hybrid approach is $0.14 \text{ Pg C yr}^{-1}$, consistent with the $f\text{CO}_2$ -products. In our sensitivity test when the hybrid approach takes into account observations from three consecutive years, the amplitude of interannual variability is reduced by half but the amplitude of decadal variability remains similar. These results highlight that an uncertainty surrounding the amplitude of interannual variability remains, but supports the robustness of the decadal variability suggested by the hybrid approach.

The revised paragraph on decadal variability in section 3.1 (i.e., “3.1 Constraints on the *decadal variability of the global ocean CO₂ sink*”) now reads, line 117: “*The hybrid approach increases the decadal variability of the ocean CO₂ sink simulated by the NEMO-PlankTOM12.1 process-based ocean model (see Methods, section 5.1, for the definition of decadal variability). Originally, over the period 1990-2022, NEMO-PlankTOM12.1 simulated amplitudes of decadal variability for the ocean CO₂ sink of $0.11 \text{ Pg C yr}^{-1}$. This value is at the high end of the decadal variability simulated by the other GOBMs used in the global carbon budget analysis (Table 1). The hybrid approach further increases this simulated decadal variability by 18% to $0.13 \pm 0.02 \text{ Pg C yr}^{-1}$, to a value close to the decadal variability estimated by the $f\text{CO}_2$ -products ($0.14 \pm 0.06 \text{ Pg C yr}^{-1}$).*”

The additional paragraph presenting the robustness of the decadal variability results in section 3.1 immediately follows the above and reads, line 125: “*We tested the robustness of the decadal variability produced by the hybrid approach with respect to (i) the choice in the selected model's configuration and parameter perturbed, and (ii) the annual availability and distribution of the SOCAT data. To do this, we firstly applied the hybrid approach to a total of six different model set ups (see methods, section 5.3 for more details). This first sensitivity analysis suggested a comparable increase in decadal variability ($0.16 \pm 0.05 \text{ Pg C yr}^{-1}$, Supplementary Table 3). Secondly, the hybrid approach was applied by considering observations from three consecutive years (see Methods, section 5.3 for more details). This second sensitivity analysis also suggested a comparable increase in the decadal variability*

(to $0.14 \pm 0.02 \text{ Pg C yr}^{-1}$, Supplementary Fig. 1). Overall, these two sensitivity analyses confirmed the robustness of the amplitude of the decadal variability suggested by the hybrid approach (Table 1). In contrast, the amplitude of the year-to-year variability was less robust because of insufficient data to constrain the hybrid approach on a yearly basis, especially in the 1990s (see Supplementary material).”

The supplementary information presenting and discussing the interannual variability results reads:

“Constraints on the interannual variability of the annual global ocean CO₂ sink

An estimation of the interannual variability of the global ocean CO₂ sink can be obtained by removing the decadal component from the original detrended time series of the annual ocean CO₂ sink (Supplementary figure S1b). The hybrid approach preserves the patterns of interannual variability from the NEMO-PlankTOM12.1 (Supplementary Fig. 1b; $r = 0.5$, $p = 0.004$, Pearson's correlation coefficient), but double its magnitude to $0.22 \text{ Pg C yr}^{-1}$. Originally, over the period 1990-2022, NEMO-PlankTOM12.1 simulated amplitudes of interannual variability for the ocean CO₂ sink ($0.10 \text{ Pg C yr}^{-1}$) comparable to the interannual variability simulated by the other GOBMs and fCO₂-products used in the global carbon budget analysis ($0.11 \pm 0.02 \text{ Pg C yr}^{-1}$ and $0.11 \pm 0.06 \text{ Pg C yr}^{-1}$, respectively). Note that the hybrid approach also increased the regional interannual variability (Supplementary table 2).

As for the decadal variability, we tested the robustness of this interannual variability estimate with respect to (i) the choice in the selected model's configuration and parameter perturbed, and (ii) the annual availability and distribution of SOCAT data. The interannual variability from the six different model set ups used (see methods, section 5.3 for more details) were comparable ($0.21 \pm 0.04 \text{ Pg C yr}^{-1}$). However, when the hybrid approach was applied by considering observations from three consecutive years, the interannual variability was strongly reduced to $0.11 \pm 0.01 \text{ Pg C yr}^{-1}$ (Supplementary figure S1b). This reduction of the interannual variability was mostly observed in the 1990s, when fewer observations were available. This 3-year interannual variability value was still larger than that estimated by GOBMs ($0.06 \pm 0.01 \text{ Pg C yr}^{-1}$), NEMO-PlankTOM12.1 ($0.06 \text{ Pg C yr}^{-1}$), and fCO₂-products ($0.07 \pm 0.03 \text{ Pg C yr}^{-1}$), when smoothed with a 3-year running mean. Overall, results from the sensitivity analyses suggest that a significant uncertainty surrounding the amplitude of interannual variability remains.

Nonetheless, despite remaining uncertainty on its amplitude, our results confirm the general consensus for the temporal patterns of interannual variability, common among the various approaches, in agreement with other studies (Bennington et al., 2022b; Mayot et al., 2023). The fact that the hybrid approach preserves the patterns of interannual variability from the NEMO-PlankTOM12.1, but increased its magnitude, could suggest that NEMO-PlankTOM12.1 and other GOBMs represent the correct processes, but either they do not respond sufficiently to changes in external forcing, or the balance among thermal and non-thermal processes in response to external forcing is imperfect (Li et al., 2019). For example, in the Southern Ocean, ocean surface fCO₂ variations over the year in NEMO-PlankTOM12.1, and in most GOBMs, tend to be too strongly influenced by temperature changes (Hauck et al., 2023b). Additional fCO₂ sampling, mostly at high latitudes, could help constrain the amplitude of the interannual variability obtained by our hybrid approach and resolve some of the identified issues here and in the literature (Hauck et al., 2023b).”

We hope this response answers the queries raised while keeping the manuscript focused on the most important and robust results.

5) Related to this point above: from Table R3, it seems that are actually quite sensitive to the perturbed parameters, which is important to discuss here. Otherwise, the results are presented here are misleading in terms of describing the actual uncertainties associated with the hybrid approach (in particular, for the interannual variability, which is one of the main points of the study).

We expanded and clarified the discussion of the uncertainty throughout our manuscript in response to this comment.

We have added a paragraph to results section 3.1 detailing the sensitivity of the decadal variability amplitudes estimated by the hybrid approach to the choice of perturbed model parameters and model configurations, and have moved the discussion of the interannual results to the Supplementary material (see response to comment 4 above).

It should be noted that the table R3 mentioned by the reviewer is included in the supplementary information (supplementary table 3). For greater clarity, we added a row to this table to provide the average ($\pm 1\sigma$) of all model set ups, and included the number of years that could be constrained within each test (which provides an indication of the reliability of the results for each model set-up).

We also added a paragraph in the discussion that acknowledges the limits of the uncertainty presented in the method. It reads, line 315: *“The $\pm 1\sigma$ uncertainty provided for the hybrid results reflects the capacity of the hybrid approach to constrain the annual ocean CO₂ sink given the availability and distribution of the fCO₂ observations. The annual uncertainty is then propagated to the decadal trend. The trend for the period 2000-2022 is better constrained than the individual ten-year trends, since the longer period naturally filters out short-term variability. Nevertheless, sensitivity tests suggest that additional uncertainty to the model set up influences the exact value of the trends, but not the overall patterns, and in particular the trend in the 2010s which is systematically lower than the trend in the 2000s in all sensitivity tests performed, and also systematically lower than the the fCO₂-products ensemble for that decade. Our analysis demonstrates the importance of regular updates and efforts to collect fCO₂ observations as part of SOCAT(Bakker et al., 2016), as well as regular evaluations of data-products, including fCO₂-products and new hybrid methodologies(Gloege et al., 2021; Hauck et al., 2023a).”*

In the Method (section 5.3), we highlighted that this sensitivity analysis demonstrated that our results about the variations of the decadal trends are robust, line 563: *“Regardless of the parameter and configuration used, the results consistently produce the lowest trend in the 1990s, and a higher trend in the 2000s than in the 2010s, although the exact trend within each decade varied (Supplementary Fig. 6 and Supplementary Table 3).”*

The robustness of our results is also mentioned in the results section, line 203: *“The distinct decadal trend variations between the 2000s and the 2010s suggested by the hybrid approach is robust to different configurations of the original model and to the choice of perturbed parameters (see methods).”*

We also added a few references to the supplementary table 3:

- Line 301: “However, both GOBMs and $f\text{CO}_2$ -products ensembles suggest similar trends between 2000s and 2010s, while the hybrid approach (including the sensitivity analyses, Supplementary Table 3) consistently produced a higher trend in the 2000s compared to the 2010s.”
- Line 351: “Although our hybrid approach always suggests an underestimation of the decadal variability by GOBMs, the exact value is sensitive to the specific model configuration (Supplementary Table 3).”
- Line 563: “Regardless of the parameter and configuration used, the results of our analysis remain unchanged about the variations of the decadal trends in ocean CO_2 sink over the period 1990-2022, with the lowest trend observed in the 1990s, and a higher trend in the 2000s than in the 2010s (Supplementary Fig. 6 and Supplementary Table 3).”

Finally, we modified the closing paragraph which discusses the implications for the estimation of the total land CO_2 sink, line 394: “Within the limits of the hybrid approach, a trend of...”, and removed the less robust sentence commenting on the reduction of the carbon budget imbalance. In response to this comment on uncertainty and a request of Reviewer 1 to expand the discussion of the implications of our results for the land CO_2 sink, we have focused the final paragraph of our discussion on the implications of our results for the total land CO_2 sink (which includes natural fluxes and emissions from land-use change), which are more robust.

6) L70-71: I find it odd to call regions poleward of 30 degrees as high-latitudes. Mid- and high-latitudes would be more accurate.

Corrected. Line 70: “This lack of consistency between the $f\text{CO}_2$ -products and the GOBMs ensemble originates in the mid- and high-latitude regions of both hemispheres (poleward of 30°N and 30°S)”

Line 155: “The hybrid approach substantially modified the simulated ocean CO_2 sink in the mid- and high latitude regions, particularly in the South,”

Line 209: “The differences in the decadal trends for the 2010s among the hybrid approach, the NEMO-PlankTOM12.1 and the $f\text{CO}_2$ -products were mostly associated with the mid- and high latitude regions (Fig. 3)”

Line 327: “Differences between NEMO-PlankTOM12.1, the hybrid approach, and the $f\text{CO}_2$ -products ensemble for the 2010s decadal trend are mostly visible in the mid- and high-latitude regions of both hemispheres (Fig. 3).”

Line 354: “These deficiencies in the mid- and high-latitude regions have been related to the coarse resolution of the ocean circulation models, the generally poor representation of the seasonality of \$f\text{CO}_2\$ in these regions,...”

7) L220-222: The presented explanation does not explain why observational products have larger trends in 2010s compared with model based products and hybrid approach (it also doesn't follow from Fig. 4 referenced in that paragraph). Rather, it explains why the observational products don't show a difference between 2000s and 2010s trends, but the hybrid approach might. The authors should re-write their explanation more accurately to reflect their statements.

The explanation was clarified, line 236 : *“The differences between the subsampled and not subsampled results suggest that different extrapolation methods outside of data-rich regions could account for the higher decadal trend in ocean CO₂ sink in the North over the 2010s in the fCO₂-products ensemble compared to the GOBMs ensemble and the hybrid approach (Fig. 3b, Supplementary Table 2).”*

We have added information regarding the significant differences between fCO₂-products and models in the 2010s. When there is no subsampling, there is no overlap in the estimated interquartile ranges associated with the distributions of the GOBMs and fCO₂-product estimates of the decadal trends in ΔfCO₂ in the 2010s. Consequently, the decadal trend in ΔfCO₂ in the North during the 2010s is significantly higher in the fCO₂-product ensemble than in the GOBMs ensemble (Kruskal–Wallis test, p-value < 0.01). We have added this information (line 234): *“In addition, the decadal trend in ΔfCO₂ in the North during the 2010s is significantly higher in the fCO₂-product ensemble than in the GOBMs ensemble (Kruskal–Wallis test, p-value < 0.01).”*

The hybrid approach suggests a higher decadal trend in the ocean CO₂ sink in the North in the 2000s (0.33 Pg C yr⁻¹ decade⁻¹) than in the 2010 (−0.02 Pg C yr⁻¹ decade⁻¹), as mentioned in the supplementary table 2. We have added uncertainties associated with the hybrid approach estimates (0.33 ± 0.06 versus −0.02 ± 0.06 Pg C yr⁻¹ decade⁻¹), which confirmed a significant difference (no overlap in the decadal estimates even when considering the uncertainty ranges). We have added a reference to this supplementary table (line 240): *“...(Fig. 3b, Supplementary Table 2).”*

8) Fig. 6: the bar graph part is confusing. Are the blue bars (if no orange or black present) to be assumed that the 2022 and 2023 models have the same number of monthly grid cells as 2021? Also, the authors do not really discuss those values, and they roughly appear to be the same from year to year and v2022 and v2023 do not add that many new data except for in the years they were published. Is that part of the plot necessary?

Yes, it is an overlapping bar chart. Note that fCO₂ observations are always added from one annual version of SOCAT to another. We changed the caption of figure 6: *“The overlapping bar chart in the bottom right-hand corner represents the number of annual observations in the SOCAT database, for each annual version of SOCAT (v2021, v2022 and v2023; using the same colour code as for the line).”*

This bar chart, showing small changes in the number of available fCO₂ observations from one annual version to another (except for the last year), supports our finding that the downward revision observed for the fCO₂-product estimate in successive publications of the global carbon budget analysis between 2021 and 2023 could not be linked to a major

change in the availability of fCO₂ observations. Instead, we argued that it is driven by changes in two fCO₂-products (line 286): “Consequently, the downward revision observed for the observation-based estimate was mainly due to a change in two fCO₂-product methodologies and, to a lesser extent, to the annual updates of the SOCAT database and fCO₂-product methods”. We felt this plot was necessary and we want to keep it.

Response to Reviewer #3:

I think this is a thorough revision that makes defensible choices and useful clarifications. I look forward to seeing this work in print, so that I can share it with colleagues who will find it interesting and timely.

Remarks on code availability:

I reviewed the code in moderate detail at first submission. While some input details may have changed since, a cursory inspection suggests the substance is similar. It is reproducible and usable in the scripting language submitted (Matlab) based on partial testing (with dummy testing data for speed). I did not confirm that it would work in the open source alternative, Octave.

1) A final minor comment: On lines 550 and 551 the word perturbation appears twice--I think this half of the sentence could be modified to be more clear (e.g., I'm assuming that the second use is referring to changes in fCO₂ resulting from the K_{1/2} perturbation, but this could be referring to something else).

Modified, line 569: “...the perturbation of the half-saturation constant of bacterial remineralisation which produces *changes in fCO₂* that are more uniform across the ocean.”

REVIEWERS' COMMENTS

Reviewer #2 (Remarks to the Author):

The authors have put in a substantial amount of work into improving the manuscript, which has made the presentation of their results much clearer. There are a few final points that the authors should address before final publication to strengthen the support of the central claims of the paper.

- 1) Figures 1 and 3: is there any benefit to showing data prior to 1990, which is the beginning of the period considered in this paper? It seems to be unnecessarily confusing to include this data.
- 2) Figure 3d/surrounding discussion: why are the trends in the South for the hybrid approach larger than both the NEMO and fCO₂ products after 2010? Some discussion on that is necessary (perhaps after the comparison of well-sampled/poorly sampled areas in the North) as it most certainly affects the global trends presented in the paper and affects the central point.
- 3) L214-215: wording is somewhat confusing. I suggest rephrasing as "it is possible in this region to compare the trends ... in areas that are generally well-sampled to those in areas that are poorly sampled"
- 4) L237: this point is not exactly defended based on the data presented in that paragraph. Providing p-values for the subsampled results and SOCAT sampling points for comparison here would be crucial to make the point that is central to the paper.

We sincerely thank the reviewers for their positive appreciation of our revised manuscript and responses to their comments. Our current revision takes into account all the improvements suggested by reviewer 2. We would like to thank the reviewers again for their valuable comments on our manuscript.

See below our point-by-point response to reviewer 2, with the reviewer's comments in blue, the new text in red and our response in black.

Response to Reviewer #2:

The authors have put in a substantial amount of work into improving the manuscript, which has made the presentation of their results much clearer. There are a few final points that the authors should address before final publication to strengthen the support of the central claims of the paper.

1) Figures 1 and 3: is there any benefit to showing data prior to 1990, which is the beginning of the period considered in this paper? It seems to be unnecessarily confusing to include this data.

As mentioned earlier in response to comment #11 from reviewer 1, on a global scale, the number of $f\text{CO}_2$ observations in the SOCAT database was relatively small prior to 1990. Consequently, some of our constrained values of the global ocean CO_2 sink prior to 1990, and its resulting interannual variability, may be uncertain. To clear up any confusion, we no longer present the results of our hybrid approach before 1990.

Line 750: *“Figure 1. Ocean CO_2 sink constrained at global scale and by latitude bands between 1990 and 2022.”*

Line 768: *“Figure 3. Anomalies of the ocean CO_2 sink constrained at global scale and by latitude bands between 1990 and 2022.”*

It should also be noted that in the Global Carbon Budget analysis (Friedlingstein et al., 2023), estimates of the ocean CO_2 sink from $f\text{CO}_2$ -products are used from 1990 onwards for the same reason:

- In the Supplements of the Global Carbon Budget 2023 (Friedlingstein et al., 2023):
“We also use eight estimates of the ocean CO_2 sink and its variability based on surface ocean $f\text{CO}_2$ maps obtained by the interpolation of surface ocean $f\text{CO}_2$ measurements from 1990 onwards due to severe restriction in data availability prior to 1990”

2) Figure 3d/surrounding discussion: why are the trends in the South for the hybrid approach larger than both the NEMO and $f\text{CO}_2$ products after 2010? Some discussion on that is necessary (perhaps after the comparison of well-sampled/poorly sampled areas in the North) as it most certainly affects the global trends presented in the paper and affects the central point.

The 2010s Southern Ocean CO₂ sink trend estimated by our hybrid approach is larger than those estimated by NEMO-PlankTOM12.1 and the fCO₂-product ensemble, but has a large uncertainty. For example, two fCO₂-products have 2010s trend estimates for the Southern Ocean in the lower range of our estimate (0.51 ± 0.18 Pg C yr⁻¹ decade⁻¹). Nevertheless, our estimate of the 2000-2022 trend for the Southern Ocean CO₂ sink (0.27 ± 0.07 Pg C yr⁻¹ decade⁻¹) is similar to the fCO₂-product ensemble (0.28 ± 0.10 Pg C yr⁻¹ decade⁻¹).

The larger uncertainty ranges associated with the application of our hybrid approach in the South than in the North or the Tropics (see Supplementary Table 2) are explained by the fact that, for these regional analyses, the number of available SOCAT observations is lower (line 506), particularly in the South. Furthermore, as mentioned in line 266: “*The trend for the period 2000-2022 is better constrained than the individual ten-year trends, since the longer period naturally filters out short-term variability.*”

Therefore, as suggested by the reviewer, we have enhanced our discussion (line 292): “*In the Southern Ocean, our hybrid approach suggests that existing fCO₂ measurements could corroborate a strong and positive decadal trend in this region in the 2010s, and more generally between 2000-2022. But the paucity of fCO₂ measurements in the Southern Ocean impedes our ability to evaluate the decadal trend in this region using observations only⁴¹. Nevertheless, our estimate of the decadal trend of the Southern Ocean CO₂ sink in the period 2000-2022 is within the range of the fCO₂-product ensemble (Supplementary Table 2). But the uncertainties associated with our hybrid approach are the largest in the Southern Ocean. Moreover, recent studies showed that undersampling could be responsible for strong biases in fCO₂-products in that region⁴².*”

3) L214-215: wording is somewhat confusing. I suggest rephrasing as "it is possible in this region to compare the trends ... in areas that are generally well-sampled to those in areas that are poorly sampled"

Done. Line 188: “*Because of the higher density of observations in the North compared with other latitudes, it is possible in this region to compare the trends in ocean CO₂ flux in areas that are generally well-sampled to those in areas that are poorly sampled.*”

4) L237: this point is not exactly defended based on the data presented in that paragraph. Providing p-values for the subsampled results and SOCAT sampling points for comparison here would be crucial to make the point that is central to the paper.

We have added the results of the Kruskal-Wallis tests for the subsampled results. Note that we cannot perform a statistical test to compare the SOCAT sampling point (one value) with the GOBMs and fCO₂-products results (several values), but the comparison is clearly visible in the figure.

Line 197: “*The SOCAT observations converted into ΔfCO_2 in the North region between 2000 and 2019 show a positive and higher trend in the 2000s compared to the 2010s (Fig. 4). Similar temporal patterns were visible in the ΔfCO_2 data from the fCO₂-products subsampled to SOCAT sampling points with a decadal trend in ΔfCO_2 in the 2000s significantly higher than in the 2010s (Kruskal–Wallis test, p -value < 0.01), as expected. However, when not subsampled, the fCO₂-products suggested a decadal trend in ΔfCO_2 in the 2000s that is not*

significantly higher than in the 2010s (Kruskal–Wallis test, p -value = 0.14), with an overlap in the estimated uncertainties in the two decades, explaining the small differences in the CO₂ sink trend between the 2000s and 2010s. This is induced by the fact that three of the seven fCO₂-products suggested a greater trend in the 2010s compared to the 2000s, and a fourth fCO₂-product suggested a strong trend in both decades. In comparison, when **subsampled or** not subsampled, the GOBMs suggested a decadal trend in $\Delta f\text{CO}_2$ in the 2000s significantly higher than in the 2010s (Kruskal–Wallis **tests**, p -value < 0.001). In addition, the decadal trend in $\Delta f\text{CO}_2$ in the North during the 2010s is significantly higher in the fCO₂-product ensemble than in the GOBMs ensemble (Kruskal–Wallis test, p -value < 0.01). The differences between the subsampled and not subsampled results suggest that different extrapolation methods outside of data-rich regions could account for the higher decadal trend in ocean CO₂ sink in the North over the 2010s in the fCO₂-products ensemble compared to the GOBMs ensemble and the hybrid approach (Fig. 3b, Supplementary Table 2).”